# A tissue-intrinsic mechanism sensitizes HIV-1 particles for TLR-triggered innate immune responses

Samy Sid Ahmed[1], Liv Zimmermann [2,3], Andrea Imle [1], Katrin Wuebben[3], Nadine Tibroni[1], Lena Rauch-Wirth [4], Jan Münch [4], Petr Chlanda [2,3,5], Frederik Graw [3,6,7] & Oliver T. Fackler [1,5,8] ✉

In vivo, HIV-1 replicates within tissues, yet the impact of three-dimensional (3D) environments on viral spread remains unclear. Our laboratory previously showed that collagen-rich 3D extracellular matrix (ECM) imposes an Environmental Restriction to cell-free Virus Infectivity (ERVI). Here, we demonstrate that ERVI is mediated by adhesive ECM components assembled into tissue-like scaffolds. Transient interactions with collagen fibers rapidly diminish virion infectivity across diverse primary strains by impairing virus fusogenicity. Notably, collagen-experienced particles also induce a distinct antiviral transcriptional program and strong pro-inflammatory cytokine secretion in monocyte-derived macrophages. Mechanistically, collagen contact induces conformational changes in the viral glycoprotein Env, enhances its interaction with toll-like receptor 2 (TLR2), and promotes trafficking into TLR8-positive endosomes, thereby amplifying innate immune sensing. Thus, ERVI functions through a dual mechanism: reducing virion fusogenicity while increasing innate immune detection. These findings identify the biophysical properties of the ECM as a tissue-intrinsic arm of antiviral innate immunity.

Untreated HIV infection induces a complex pathology characterized by depletion of CD4 T cells, generalized dysfunction of B and CD8 T cells as well as chronic immune activation and inflammation that promotes lymph node fibrosis[1–3]. In particular immune cell dysfunction and chronic inflammation persist even in people living with HIV (PLWH) in which peripheral viral load is suppressed by highly active antiviral therapy[4]. While extensive research over the past decades provided a detailed understanding of the replication strategies and molecular interactions that enable HIV to replicate in individual target cell types ex vivo, much less information is available regarding the

mechanisms of HIV-induced immunopathology in infected tissue. This lack of knowledge on the impact of tissue on HIV spread and pathogenesis reflects the limited availability of culture models to address these questions experimentally. Humanized mouse models provide a valuable system to study virus spread in vivo but fail to recapitulate important immune responses of PLWH and their long-term effects. Organotypic systems such as tonsil or cervix explants allow to study HIV infection in the context of physiological tissue composition but experimental parameters are difficult to control[5,6]. Our laboratory previously established three dimensional (3D) matrices of the major

[1]Department of Infectious Diseases, Integrative Virology, CIID, Heidelberg University, Medical Faculty Heidelberg, Heidelberg, Germany. [2]Schaller Research Groups, Department of Infectious Diseases, Virology, Heidelberg University, Medical Faculty Heidelberg, Heidelberg, Germany. [3]BioQuant-Center for Quantitative Biology, Heidelberg University, Heidelberg, Germany. [4]Institute of Molecular Virology, Ulm University Medical Center, Ulm, Germany. [5]SynthImmune Cluster of Excellence, Heidelberg University, Heidelberg, Germany. [6]Department of Internal Medicine 5, Friedrich-Alexander-Universität Erlangen-Nürnberg and Universitätsklinikum Erlangen, Erlangen, Germany. [7]Interdisciplinary Center for Scientific Computing, Heidelberg University, Heidelberg, Germany. [8]German Centre for Infection Research (DZIF), Partner Site Heidelberg, Heidelberg, Germany. ✉ e-mail: oliver.fackler@med.uni-heidelberg.de

extracellular matrix (ECM) component type I collagen as a minimalistic cell culture model that recapitulates important features of extracellular tissue environments and in which parameters such as cell density and 3D organization can be controlled[5,7]. The initial characterization of this 3D collagen culture model allowed us to address which transmission mode HIV employs in tissue-like environments: HIV can spread from infected donor cells either by the release of cell-free particles into the extracellular space that then diffuse to infect new target cells (cell-free transmission) or via close physical contacts between donor and target cell (cell-associated transmission)[8–10]. Combining computational modeling and subsequent experimental validation identified cell-associated transmission as the predominant mode of virus spread in 3D collagen at conditions of limited cell density where virions either need to diffuse or donor cells have to migrate to new target cells to sustain HIV spread[7]. In contrast, in 3D collagen with very high cell density or in classical suspension cultures, cell-free and cell-associated transmission modes supported HIV-1 spread with comparable efficacy. This shift towards cell-associated virus transmission in tissue-like environments reflected an enhanced efficiency of cell-associated spread, but also a significant reduction in the infectivity of HIV-1 particles. Imaging analysis revealed that HIV-1 particles freely diffuse in the tissue-like environment but undergo short and reversible physical interactions with collagen fibers. These findings suggested that in tissue, the physical contact with ECM limits the infectivity of cell-free HIV-1 particles but the mechanism of this restriction remained elusive.

As an enveloped virus particle, the infectious potential of HIV-1 virions is determined by a large number of parameters. This comprises basic properties such as particle integrity, packaging of the viral genome and essential enzymes, and incorporation of the viral glycoprotein Env, and liquid-order membrane microdomain-like lipid composition of the viral envelope[11–14]. Dependency on this complex set of requirements renders the generation of infectious HIV particle an attractive target of the activity of cell intrinsic antiviral factors (so called restriction factors) including tetherin, SERINC proteins, 90 K, IFTIM proteins, GBP 2 and 5 and PSGL-1 that reduce HIV-1 infectivity via distinct mechanisms[15–22]. In analogy to these intracellular barriers that counteract HIV-1 spread, we designated the decrease of virion infectivity by tissue-like environments as an *Extracellular Restriction of Virion Infectivity* (ERVI).

HIV particles also present a number of pathogen-associated patterns (PAMPs) that can be recognized by pattern recognition receptors (PRRs) to trigger antiviral signaling. HIV associated PAMPs include the incoming single strand RNA genome (gRNA)[23] and products of reverse transcription of the RNA genome into DNA but also post integration replication intermediates[24] and protein structures[25–27]. Whether tissue environments impact the mode and potency of innate immune recognition of HIV-1 particles has not been assessed.

In this study, we set out to define the mechanism and functional consequences of ERVI. We find that ERVI has two mechanistically distinct effects on HIV-1 virions that (i) restrict their infectivity and (ii) sensitize them for innate immune recognition. Mechanistically, the infectivity impairment results from reduced fusogenicity of virions while enhanced innate immune recognition reflects the recognition of the viral glycoprotein by TLR2 and sensing of viral gRNA by endosomal TLR8. These results show that the biophysical properties of tissue-like environments bear an intrinsic antiviral activity that limits viral spread while inducing an inflammatory innate immune response in myeloid cells.

## Results

### ERVI is a rapidly induced and saturable restriction to HIV-1 infectivity

Our previous results had established that tissue-like 3D collagen environments pose the ERVI barrier to markedly reduce the infectivity of HIV-1 particles recovered from the supernatants of 3D collagen matrices in which virus-producing cells or cell-free virus particles had been embedded[7]. Single particle tracking revealed that HIV virions diffuse freely in the 3D matrix but undergo transient physical contacts with collagen fibers. These interactions were in the millisecond range and did not result in coating of the fibers with virus particles[7]. To gain more insight into the nature of ERVI, we compared the impact of culturing HIV-1 NL4.3 particles in suspension (S) or upon embedding in 3D collagen with different densities over time (dense collagen (DC): 3 mg/ml rat tail collagen; loose collagen (LC): 1.6 mg/ml bovine skin collagen) (Fig. 1a). The infectivity of virions that had diffused into the culture supernatant was assessed on TZM-bl reporter cells kept in the absence of collagen, in which the luciferase gene is under control of the HIV-1 promoter and de novo expression of the viral transactivator Tat in productively infected cells triggers luciferase expression[28]. The amount of luciferase expression relative to the amounts of virus used for infection, as determined by quantifying the activity of viral reverse transcriptase by the SG-PERT assay, yielded the relative infectivity of HIV particles. In line with our previous findings, single rounds of infection revealed that culturing HIV-1 in DC or LC for 16 h reduced their infectivity to 10.6% or 14.3% of the particles from parallel suspension cultures (Fig. 1b). Kinetic analysis revealed that virion infectivity in suspension steadily decreased over time, reflecting the established gradual decay of virion infectivity[29]. In contrast, contact with DC and LC reduced infectivity already after the initial 10 min. of collagen polymerization for DC, or within 4 h following 45 min. of collagen polymerization for LC. Based on our previous single particle tracking analyses[7], substantial portions of all particles undergo physical interactions with collagen fibers in these time frames within the 3D collagen and such contacts are likely further increased during diffusion out of the matrix into the cell culture supernatant. In this scenario, the faster kinetics of infectivity impairment in DC likely reflect the higher frequency of contacts between individual virions and collagen fibers in the more densely packed DC as compared to LC. The subsequent reduction of the remaining virion infectivity over time followed comparable kinetics under all culture conditions (Fig. 1c). While assessing the infectivity of virions still contained within the collagen gels was precluded by adverse effects of type I collagenase required to digest the matrix on attachment and viability of the TZM-bl reporter cells, a significant, yet less pronounced, reduction of virion infectivity was also observed when virions were cultured on top of already polymerized collagen matrices (Fig. 1d, see Sup. Fig. 1a for experiment layout: residual infectivity for viruses embedded in collagen for 16 h: 6.6% DC, 32.4% LC or viruses kept on top of collagen: 26.7% DC, 43.5% LC). In contrast, supernatants of collagen matrices polymerized in the absence of virus particles or the buffers in which collagen is polymerized (PA) failed to restrict the infectivity of virions. The reduction of virion infectivity by collagen matrices thus depends on physical contact of virions with the matrix and is not mediated by soluble components.

Physical stress can impair virus infectivity by inducing shedding of the viral glycoprotein from the virion, and the efficacy of glycoprotein incorporation is an important determinant of infectivity[30,31]. However, HIV-1 Env gp120 levels as determined by the gp120 to p24 ratio of particles, which sometimes slightly varied for particles with collagen experience, were overall similar for virions cultured in suspension or in 3D collagen for 16 h (Fig. 1e, f). Since cellular restrictions to virus infection typically act as physical barriers that can be overcome by an excess of virus particles[32], we analyzed if the amounts of virions added to a constant culture volume affected the magnitude of reduction in virion infectivity (Fig. 1g). While the relative infectivity of virus particles kept in suspension for 16 h was unaffected by the concentration of virus, the infectivity reduction by ERVI was significantly less efficient at higher virus concentrations in DC or LC (Fig. 1g; relative infectivity 7- or 2-fold higher at $10^7$ vs. $10^5$ infectious units in DC or LC; $p = 0.022$ and $p < 0.001$ respectively). The magnitude of infectivity restriction by

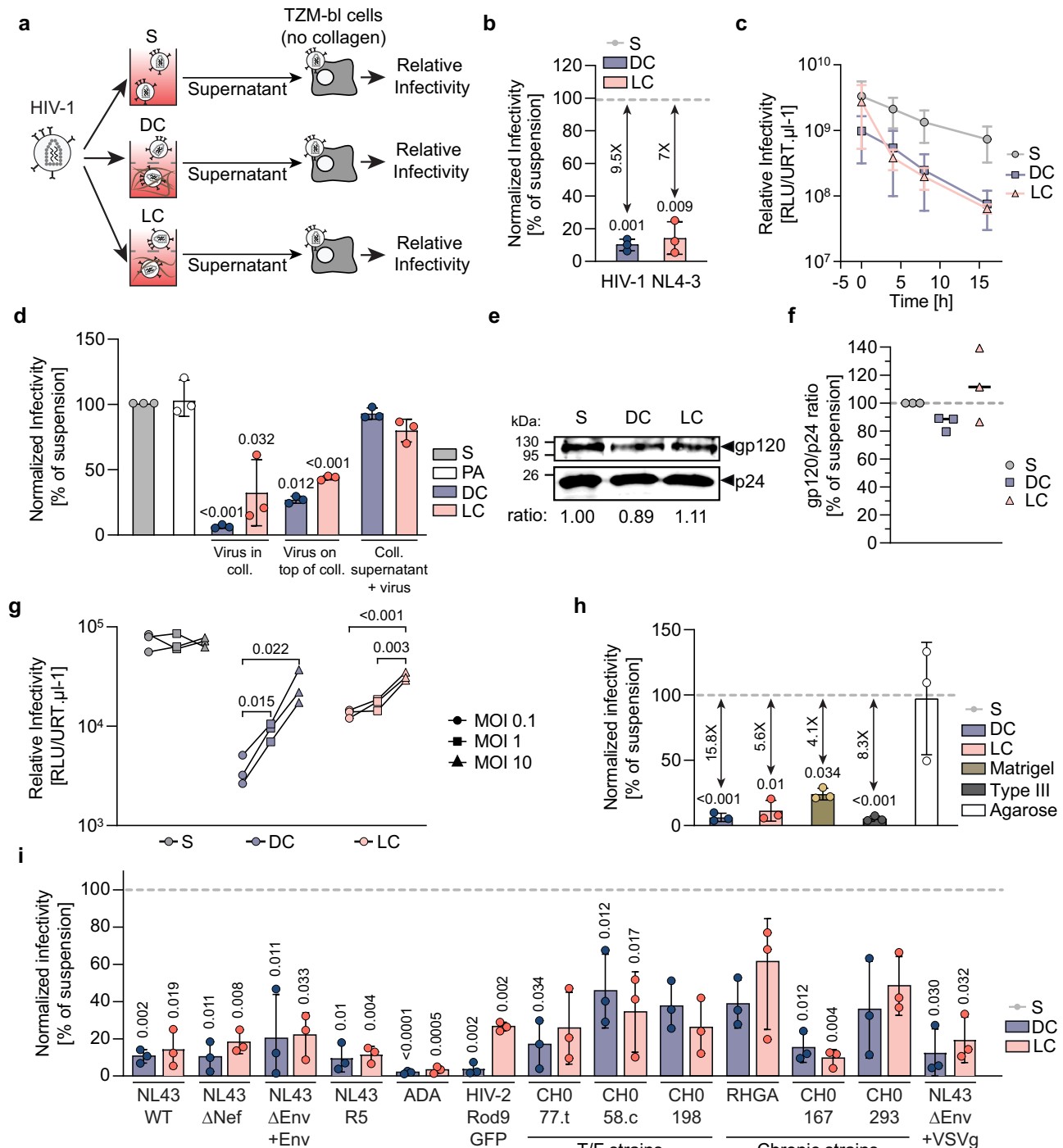

**Fig. 1 | ERVI is rapidly induced, saturable, and conserved across viruses and different types of matrices. a** Experimental workflow. HIV-1 NL4.3 virions were cultured in suspension (S) or embedded in collagen matrices (DC: Dense collagen; LC: Loose collagen). TZM-bl cells were infected and the relative infectivity determined. **b** Normalized infectivity of HIV-1 NL4.3 particles. Virions were cultured in S, DC or LC for 16 h prior to relative infectivity determination. Data is normalized to suspension condition (gray dotted line). **c** Kinetics of ERVI. Culture supernatants were harvested at 0, 4, 8 or 16 h post seeding/embedding, the relative infectivity of virions was determined as in **b**. **d** Normalized infectivity of differentially cultured HIV-1 NL4.3 virions. PA: polymerizing agent. **e**. One representative Western Blot analysis showing the gp120/p24 ratio from HIV-1 NL4.3 virions cultured in S, DC or LC. The gp120/p24 ratios are indicated below the blots. **f** Quantification of western

blot analyses from three independent experiments as in **e**. **g** Saturation of ERVI. Increasing amounts of virions were cultured in S, DC or LC prior to infectivity determination. **h** Normalized infectivity of HIV-1 NL4.3 particles after seeding or embedding in different matrices for 16 h. Data are normalized to suspension condition (gray dotted line). **i** Conservation of sensitivity to ERVI across lentiviruses. The normalized infectivity of a panel of lentiviruses was assessed after ERVI as in **a**. Data are normalized to respective suspension condition (gray dotted line). Results represent the mean ± SD of three independent experiments. Symbols indicate data from individual experiments. Significance is indicated by p-values, and was calculated by one-way ANOVA test; Dunnett's post-test (**b**, **d**, **h**, **i**), or by matched two-way ANOVA test; Tukey's post-test (**g**). Source data are provided as a Source Data file.

ERVI depends on the abundance of virions per matrix volume. This suggested that the capacity of 3D collagen to physically interact with HIV particles may be critical for its antiviral activity. To test this more directly, we assessed if pre-treatment of collagen-coated surfaces (as in Fig. 1d) with HIV-Env containing lentiviral particles affected the antiviral capacity (Sup. Fig. 1b). Indeed, preincubation of with increasing amounts of lentiviral particles slightly reduced the ability of collagen to impair the infectivity of HIV particles in a dose-dependent manner, but did not abrogate the inhibition. Presumably due to competition with infectious HIV particles for binding to the entry receptors, using higher amounts of lentiviral particles led to an overall reduction of infection events on the TZM-bl cells used for infectivity determination and were therefore not informative. These results suggest that specific functional and/or physical interaction sites for HIV-1 Env in collagen are involved in reducing virion infectivity. Whether this mechanism is sufficient to explain the antiviral activity of collagen or it acts in conjunction with additional mechanisms warrants further investigation.

### ERVI is exerted by different collagens and complex ECM and affects a broad range of HIV-1 variants

We next sought to define how the architecture and biophysical properties of the 3D matrix affect its ability to reduce the infectivity of HIV-1 after embedding of virions for 16 h and compared several matrices that (i) can be polymerized without harming per se the infectivity of HIV particles (e.g., high temperature or UV exposure) and (ii) result in pore sizes that allow HIV-1 particles to diffuse within and out of the 3D matrix (Fig. 1h, Sup. Fig. 1c). DC and LC matrices are assembled into fibers polymerized from purified rat and bovine type I collagen proteins, the most abundant collagen type in tissue[33]. Confocal reflection microscopy analysis of these matrices confirmed the different density of both type I collagens after polymerization and revealed that DC was enriched in branched collagen bundles (Sup. Fig. 1c). The architecture of 3D matrices made of human type III collagen, the second most abundant fibrillary collagen in tissue that is synthesized by reticular cells and lines e.g., vasculature[34], resembled that of DC, albeit with shorter and thinner collagen bundles, and type III collagen reduced HIV-1 infectivity with similar efficacy than type I collagen. We also tested Matrigel, a complex extracellular environment of the mouse basal membrane rich in collagen IV, III and I[34] as model for a complex tissue environment. Matrigel assembles into 3D matrices with smaller pore sizes, refractive indexes and higher heterogeneity than purified collagen matrices, which reduces signals from autoreflection microscopy (Sup. Fig. 1c)[35]. Nevertheless, its antiviral activity was comparable to that of DC (24.1 ± 4.4% of suspension or 4-fold reduction). In contrast, agarose hydrogels, which are made of bundles of thin linear filaments without cell adhesion features that do not produce reflection signals[36], did not impair the infectivity of HIV-1 virions. We conclude that ERVI is a conserved feature of adhesive collagen matrices and ECM from different mammals with distinct architecture.

To assess how conserved the sensitivity to ERVI is among primary lentiviruses, we next analyzed the impact of 3D collagen on a panel of lab-adapted and primary HIV-1 strains as well as one HIV-2 strain on TZM-bl reporter cells (Fig. 1i). The results revealed that the infectivity of all HIV strains tested was significantly reduced by ERVI, but to varying magnitude ranging from strong (52.9-fold, HIV-1 ADA, DC) to very mild inhibition (1.6-fold, HIV-1 RHGA, LC). The reduction of infectivity by ERVI was independent of the HIV-1 entry co-receptor preference, but patient-derived transmitted founder and chronic HIV-1 variants tended to be less sensitive to ERVI than lab-adapted HIV-1 (DC: 10.5% residual infectivity for lab-adapted vs 33.1% for primary isolates). Notably, also virions lacking HIV-1 Env but pseudotyped with the glycoprotein of vesicular stomatitis virus (VSVg) were sensitive to ERVI. Identical amounts of infectious units were used for suspension and collagen cultures for all viruses. Importantly, the sensitivity of different

virus strains to ERVI was not correlated to the amount of virus used for infection (Sup. Fig. 1d). Sensitivity to ERVI thus appears to be an intrinsic property of individual HIV variants. To specifically analyze the impact of the viral glycoprotein for sensitivity to ERVI, we pseudotyped GFP-encoding lentiviral particles with glycoproteins of several unrelated viruses which were used to transduce more permissive Huh7.5 cells. The transduction efficiency of Huh7.5 cells using lentiviral particles pseudotyped with the Hepatitis C Virus isolates Con1 or JFH1 or the Vesicular Stomatitis Virus glycoprotein (VSVg) was moderately reduced upon contact with 3D collagen for 16 h (Sup. Fig. 1e, statistically significant reduction for particles pseudotyped with VSVg glycoprotein in DC). Together, these results reveal that a broad range of HIV-1 variants are sensitive to ERVI. This extracellular restriction is implemented rapidly upon contact with the 3D environment, involves the interaction of virus particles with specific sites in collagen, and is exerted by a variety of adhesive extracellular matrices. The results also suggest that the viral glycoprotein is a key determinant for the sensitivity to ECM-mediated infectivity reduction and that the sensitivity to ERVI is dependent on intrinsic properties of glycoproteins that may include their topology.

### ERVI restricts HIV-1 infectivity without affecting structural integrity of virus particles

We next addressed in more detail how the physical contact of HIV particles with the 3D collagen environment affects their infectivity. We considered that following transient contacts of HIV-1 particles with collagen fibers, collagen material may remain attached to the virions and compromise their infectivity. To address this possibility, we generated HIV-1 particles that incorporated a Vpr-integrase.GFP (Vpr.Int.GFP)[37] fusion protein for visualization and validated that they remained sensitive to the infectivity reduction by ERVI after 16 h of contact with collagen (Fig. 2a, 8,6-fold reduction). Embedding these particles in fluorescently labeled LC allowed to assess the presence of fluorescent collagen at their surface by light microscopy (Fig. 2b). These analyses focused on LC using an established protocol[38] since specific labeling of the viscous, highly concentrated DC was technically challenging. Confocal imaging of virions within fluorescent LC matrices allowed to visualize a number of non-tethered particles present between collagen fibers (Fig. 2c). We next assessed the presence of fluorescent collagen fibers at the surface of Vpr.Int.GFP containing virions by TIRF-microscopy (Fig. 2d). While small fragments of free collagen fibers could occasionally be observed, we did not detect any AlexaFluor-647 signals studding the surface of virions (Fig. 2d). To analyze the impact of 3D collagen on virion morphology by cryo-electron tomography, HIV-1 particles kept in suspension or in DC or LC for 16 h were placed on EM grids and processed for cryo-ET analysis. Under all three conditions, enveloped HIV-1 particles with the typical conical core and a diameter ranging from 104 to 154 nm (mean values: S: 134.5 ± 11.5 nm, DC: 135.2 ± 9.6 nm, LC: 132.0 ± 11.6 nm) were observed. Typically, less particles were retrieved from DC as compared to LC and S, likely reflecting the much higher compaction of DC gels that hinder the diffusion of particles into the culture supernatants (Fig. 2e, f). All analyzed particles appeared intact without appreciable membrane rupture or deformation and showed sparsely distributed Env spikes. As size comparison, we also analyzed the morphology of DC and LC fibers, which displayed the characteristic structure of collagen fibrils, with a tight packing of D-periodic polyproline type II helices corresponding to the spacing between individual tropocollagen monomers[39] (Fig. 2g). Consistent with our analysis using fluorescent collagen, we failed to observe virion-associated collagen fibers (Fig. 2e). Since these analyses did not detect morphological aberrations in HIV-1 particle architecture resulting from the interaction with 3D collagen, we next asked if the reduced infectivity of HIV-1 particles subjected to ERVI for 16 h can be rescued by synthetic peptide nanofibrils (PNF) which boost the infectivity of HIV-1 particles by increasing

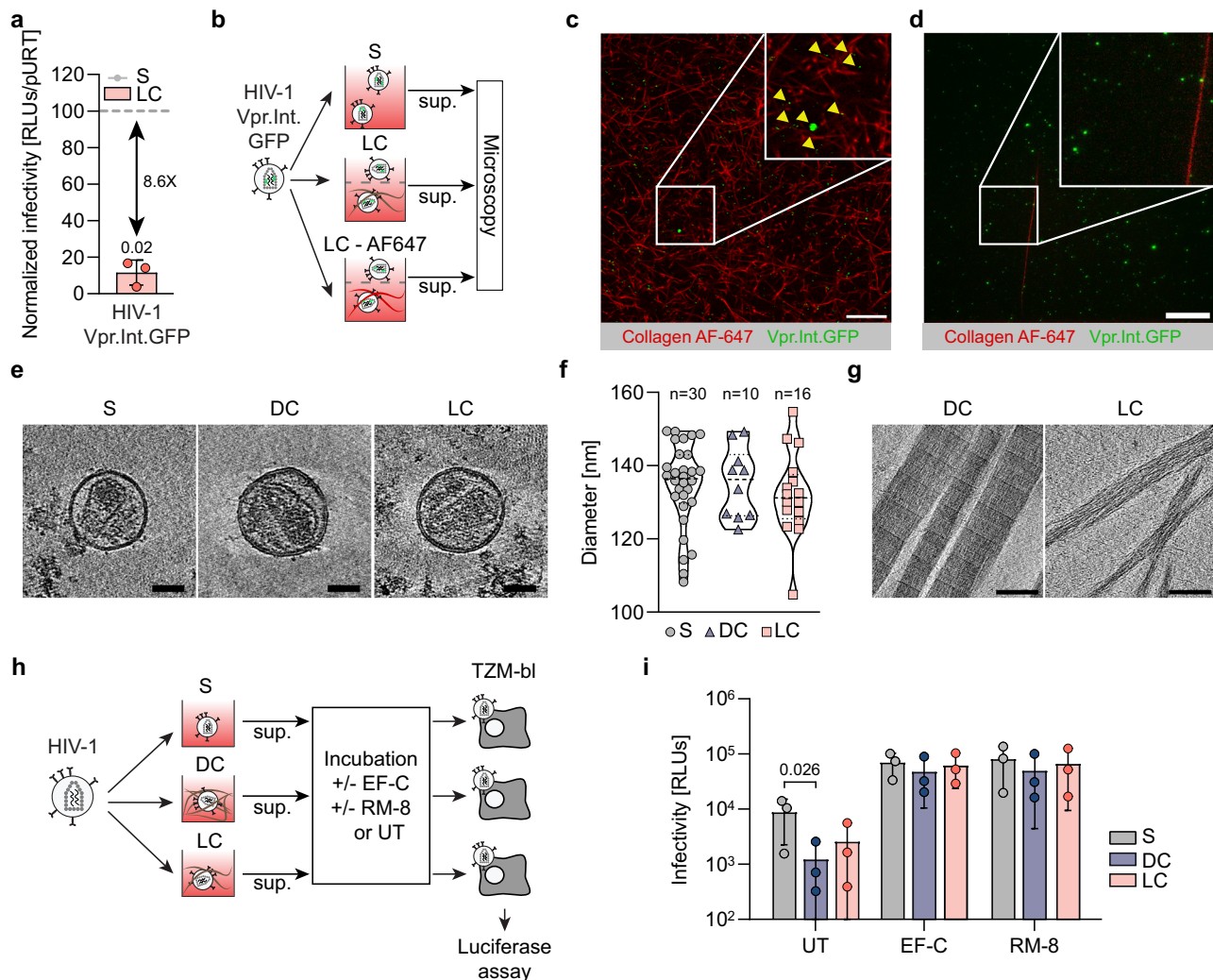

**Fig. 2 | ERVI does not result from collagen deposition or structural damages of embedded virions. a** Normalized infectivity of HIV-1 NL4.3 virus containing Vpr.Int.GFP fusion proteins after exposure to LC. **b** Experimental workflow to assess collagen deposition at the surface of virions. HIV-1 NL4.3 Vpr.Int.GFP virions were collected from S, LC or LCAF647 cultures and analyzed by microscopy. **c** Representative confocal micrographs of HIV1 NL4.3 Vpr.Int.GFP virions within AlexaFluor 647 stained LC matrices. Yellow arrows depict fluorescent virions. Scale bar: 20 μm. **d** Representative TIRF-micrographs of HIV-1 NL4.3 Vpr.Int.GFP virions were harvested from the supernatants of fluorescently labeled collagen matrices as in (**c**). Scale bar: 5 μm. **e** Averaged computational slices of a tomogram showing HIV-1 NL4.3 virions after 16 h of culture in S (left panel), DC (middle panel), or LC (right panel). Scale bar = 50 nm. **f** Quantification of the diameter of virus particles treated as indicated in **d**. Violin plot shows individual data points with corresponding median, 25% and 75% quartiles. **g** Averaged computational slices of a tomogram showing disrupted dense (left panel) or loose (right panel) collagen fibers. Scale bar = 100 nm. **h** Experimental workflow. HIV-1 NL4.3 virions retrieved from suspension or collagen matrices after 16 h of culture were harvested, and equivalent amounts of RT units were incubated with RM-8 or EF-C infectivity-enhancing peptide, nanofibrils (PNFs) or left untreated (UT) prior to infection of TZM-bl reporter cells. **i** Impact of infectivity enhancers on the infectivity of suspension or collagen primed virions. Results represent the mean ± SD of three independent experiments. Symbols indicate individual experiments. Significance is indicated by p-values, and was calculated by one-way ANOVA, Dunnett's post-test (**a**), unpaired t-tests (**d**), or by two-way ANOVA test with Geisser Greenhouse correction; Tukey's post-test (**i**). Source data are provided as a Source Data file.

their interaction with target cells and promoting viral fusogenicity[40–42]. Indeed, incubating HIV-1 particles with EF-C or RM-8 PNFs increased the infectivity of all particles and almost fully overcame the inhibitory effect of ERVI (Fig. 2h, i). Together, these results reveal that ERVI does not result from global disruption of HIV-1 particle architecture and suggest that ERVI does not affect the intrinsic replication potential of HIV particles but rather the efficacy of their functional interaction with target cells.

## ERVI is manifest at the step of virus fusion without affecting virion binding to target cells

The finding that infection enhancers boost the infectivity of HIV-1 particles subjected to ERVI suggested that the restriction acts at the early steps of the viral life cycle. To test if ERVI impairs the ability of

HIV-1 particles to bind to target cells, we generated virions that incorporated Vpr.mRuby2 during virus production for visualization[43] (Fig. 3a). Incorporation of Vpr.mRuby2 did not affect their sensitivity to infectivity reduction upon contact with 3D collagen (Fig. 3b). To visualize their interaction with the surface of TZM-bl target cells by spinning disk microscopy, virus particles retrieved after 16 h culture in suspension or 3D collagen were incubated with the target cells for 2 h. This incubation occurred at 4 °C to avoid particle internalization and to detect individual fluorescent HIV-1 particles attached to the cell surface prior to processing for light microscopy analysis (Fig. 3c). While incubating these particles for 16 h in suspension slightly reduced the number of virions detected at the surface of target cells (Fig. 3d, 2 ± 0.5 bound virus particles/cell for fresh virus vs. 1.1 ± 0.4 bound virus particles/cell for suspension), no additional reduction in binding

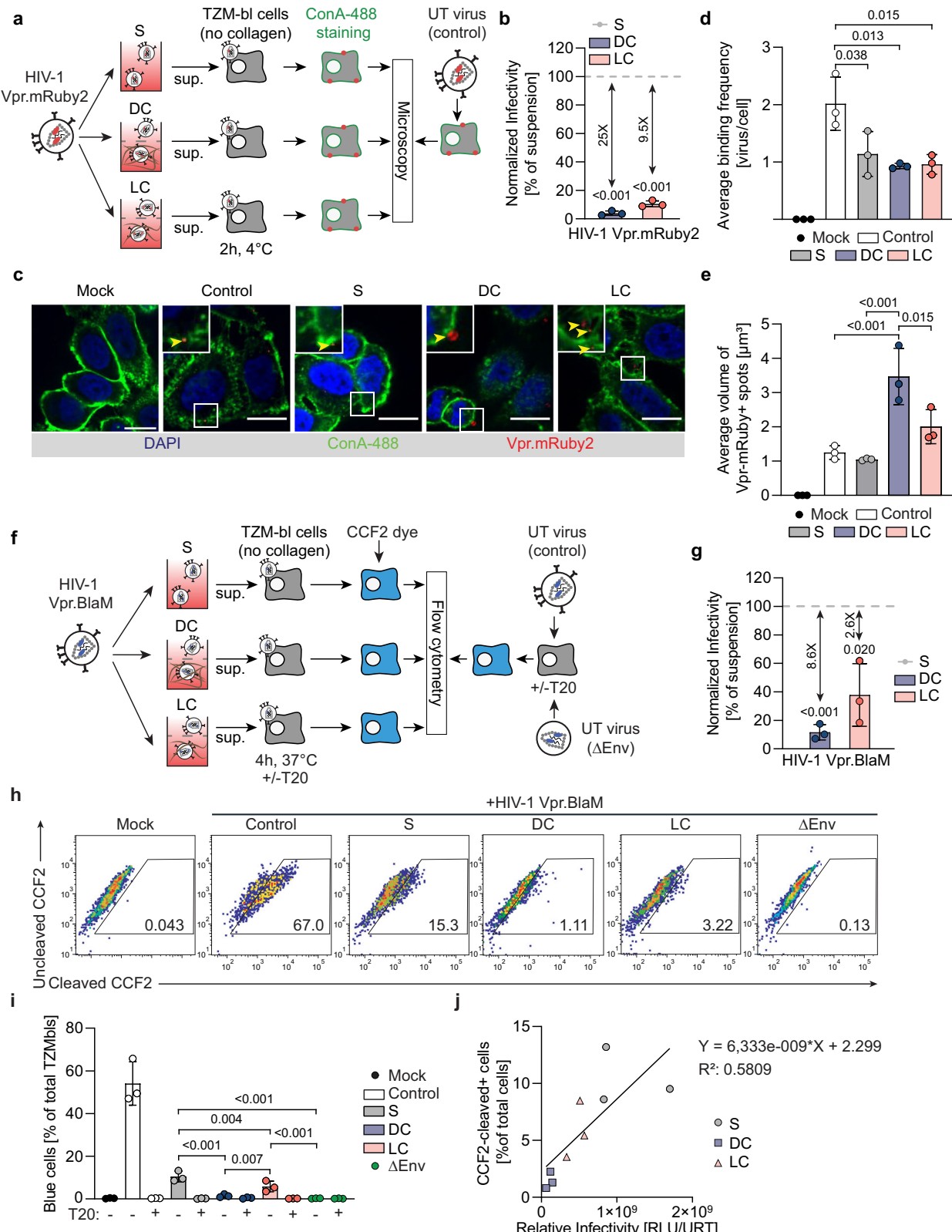

efficacy was observed for virions that had been embedded in DC or LC. Notably however, prior contact with DC fibers resulted in significantly larger aggregates of virus particles being visible at the surface of target cells (suspension: $1.05 \pm 0.03\,\mu m^3$, DC: $3.47 \pm 0.8\,\mu m^3$) (Fig. 3e). Such particle aggregation was not observed upon contact with LC ($2.0 \pm 0.5\,\mu m^3$), indicating that this effect is not essential for the virion infectivity reduction caused by contact with 3D collagen. The

predominant action of ERVI, therefore, is not at the level of virus binding to target cells.

To assess the ability of HIV particles to fuse with target cells, Vpr.BlaM-containing particles were produced and used to quantify the conversion of the ß-lactamase (BlaM) substrate CCF2 in the cytosol of target cells by flow cytometry as a measure for fusion[43,44] (Fig. 3f). The incorporation of Vpr.Blam did not affect the sensitivity of the virions to

**Fig. 3 | ERVI impairs HIV-1 entry but not binding to TZM-bl target cells.**
**a** Experimental workflow. HIV-1 NL4.3 Vpr.mRuby2 virions were cultured in S, DC or LC for 16 h. Equivalent amounts of RT units from culture supernatants or untreated virions were then incubated with TZM-bl cells for 2 h at 4 °C. Cell membranes were then stained with Concanavalin-A AF-488, prior to microscopy processing.
**b** Normalized infectivity of HIV-1 NL4.3 Vpr.mRuby2 virions after 16 h of culture as in **a**. **c** Representative spinning disk micrographs of cells incubated with HIV-1 NL4.3 Vpr.mRuby2 virions. Yellow arrows indicate Vpr.mRuby2+ spots detected at the surface of the target cells. Scale bar: 15 µm. **d** Average binding frequency of HIV-1 NL4.3 Vpr.mRuby2 virions to target cells as shown in **c**. **e** Average volume of Vpr.mRuby2 spots detected as shown in **c**. **f** Experimental workflow. HIV1 NL4.3 Vpr.BlaM virions were as in **a**. Equivalent amounts of RT units from culture supernatants or untreated virions were then incubated with TZM-bl cells for 4 h at 37 °C,

in the presence or absence of T20 fusion inhibitor. Cells were then loaded with the CCF2-AM and processed by flow cytometry. **g** Normalized infectivity of HIV-1 NL4.3 Vpr.BlaM virions after 16h of culture as in **a**. **h** Representative flow cytometry dot plots of CCF2 loaded cells. Gates identify cleaved-CCF2+ cells. See Sup. Figure. 9a for gating strategy. **i** Quantification of the percentage of CCF2-product positive cells measured by flow cytometry. **j** Correlation between Relative Infectivity and levels of CCF2-product positive cells after ERVI by linear regression. Results represent the mean ± SD of three independent experiments. Symbols indicate individual experiments. Significance is indicated by p-values, and was calculated by one-way ANOVA test; Dunnets's post-test (**b**, **g**), or by one-way ANOVA test; Tukey's post-test (**d**, **e**, **i**). Source data are provided as a Source Data file.

ERVI in DC, while the infectivity reduction by LC was slightly less pronounced (Fig. 3g). Analyzing the fusion capacity of these particles revealed efficient cytosolic delivery of Vpr.Blam by HIV-1 particles kept in suspension (10.4 ± 2.4%), which was dependent on the viral glycoprotein Env and could be inhibited by the HIV fusion inhibitor T20 (Fig. 3h, i). This fusion capacity was strongly impaired for particles that had prior contact with DC (1.5 ± 0.7 %). LC (5.4 ± 2.5%) had a less pronounced effect on the fusion capacity of HIV-1 particles and overall, infection rates and fusion capacity under the different conditions analyzed were correlated, albeit with deviations in particular for virions from suspension cultures (Fig. 3j). Consistent with the mapping of the ERVI-induced defect to HIV infection to the fusion step, the presence of fusion-enhancing PNFs during infection increased the infectivity of collagen-experienced virus particles to that of particles without prior collagen experience (Sup. Fig. 1f). We conclude that the interaction of HIV-1 particles with tissue-like environments reduces their infectivity by impairing their ability to fuse with target cells and that ERVI may also affect additional post entry steps.

## ERVI moderately reduces infection of primary CD4 + T cells and MDMs

HeLa-derived TZM-bl cells are a convenient and widely used reporter cell to quantify the infectivity of HIV-1 particles but cannot reflect the differences in entry binding and receptor densities, membrane lipid composition as well as uptake pathways between different primary target cells[45]. We therefore analyzed the relevance of ERVI for infection of primary human CD4[+] T cells and primary human monocyte-derived macrophages (MDMs) with a HIV-1 NL4.3 variant that uses CCR5 as entry coreceptor (HIV-1 NL4.3 R5) due to a minimal set of seven amino acid changes[46] in the Env glycoprotein (Fig. 4a). This co-receptor tropism did not affect the sensitivity of these particles to ERVI when assessed on TZM-bl cells (Fig. 4b), which was again more pronounced in DC than LC. Productive infection of primary target cells was assessed by quantifying the percentage of cells with intracellular p24 capsid by flow cytometry (CD4[+] T cells, Fig. 4c, MDMs, Fig. 4d). The reverse transcriptase inhibitor efavirenz (EFZ) was used to define background detection of input virus. On both, CD4[+] T cells and MDMs, productive infection by particles with prior contact to DC was significantly reduced (3.6-fold reduction to 27.5% of suspension on CD4[+] T cells, 2.9-fold reduction to 34.8% of suspension on MDMs) and LC only mediated a very mild reduction that did not reach statistical significance (1.8-fold on CD4[+] T cells, 1.4-fold on MDMs). These results reveal that ERVI reduces the infectivity of cell-free HIV-1 particles on primary target cells but with lower magnitude than on TZM-bl cells.

With 27.5% infectivity remaining, the impact of ERVI on infection of primary CD4 T cells was less pronounced than on TZM-bl cells, where ERVI reduces virion infectivity to 14% of that of virions in suspension culture. Our computational model of HIV-1 replication in 3D collagen cultures of primary human mononuclear cells, which predicted that cell-associated virus transmission largely dominates over cell-free infection in 3D collagen matrices[7], was based on the value of

14% of cell-free infectivity remaining in 3D. We therefore asked if the fact that the infectivity of these virions is higher on the primary target cells present in these 3D cultures affects this conclusion. To this end, we revisited our previous analyses estimating the efficacy of cell-free and cell-to-cell transmission within suspension and 3D collagen matrices by mathematical modeling[7]. Varying the parameter defining the reduced infectivity of cell-free infection within collagen compared to suspension, we estimate that the contribution of cell-free transmission to viral transmission within collagen continuously increases with increasing infectivity preservation (Sup Fig. 2a). Differences between estimates for LC and DC are partly affected by technical compensations within the fitting procedure due to model constraints, which lead to reduced estimates of cell-to-cell transmission rates within DC and thereby potentially underestimating the contribution of cell-to-cell transmission for this environment (Sup Fig. 2b). Under these conditions, the best estimates predict that an infectivity reduction to 27.5% results in a contribution of cell-cell transmission to overall virus spread of ~60% and ~30% in LC and DC, respectively. We conclude that in our primary cell 3D cultures, cell-associated transmission remains an important driver of HIV-1 spread in 3D collagen even if the infectivity of cell-free particles is less reduced than previously thought.

## Collagen experience sensitizes HIV-1 particles for innate immune recognition by MDMs

Since the virion infectivity reduction by ERVI was not very pronounced upon infection of primary cells such as MDMs, we asked if ERVI has functional implications in MDMs in addition to reducing virion infectivity. Since MDMs can produce cytokines as consequence of innate immune recognition of HIV particles[47–49], we quantified the amounts of a panel of cytokines in the supernatant of MDMs from five independent donors 3 days after challenge with HIV-1 particles with or without 16 h of 3D collagen experience (Fig. 4e). Interestingly, prior encounter of HIV-1 particles with tissue-like 3D environments markedly and broadly altered the cytokine response of MDMs, resulting in increased release of proinflammatory cytokines (Fig. 4e). This effect was particularly pronounced for HIV-1 particles following encounter with DC (Fig. 4e, compare LC and DC for D79 and D80) and the response was similar in breadth but distinct in magnitude and specificity to cytokine production in response to LPS (Fig. 4e, D280-D282). Among the most differentially secreted cytokines were IL-6 (137-fold induction in DC, 154-fold in LC) (Sup. Figure 3a), IL-8 (6.5-fold induction in DC and 5-fold in LC) (Sup. Fig. 3b), TNF (60-fold induction in DC, 38-fold in LC) (Sup. Figure 3c), MIP-1ß (152-fold induction in DC, 2-fold in LC) (Sup. Fig. 3d), IL12p40 (75-fold induction in DC, 16-fold in LC) (Sup. Fig. 3e) or GRO-α (41-fold induction in DC, 12-fold in LC) (Sup. Fig. 3f). Independent validation of production of IL-6, one of the most prominent cytokines induced upon challenge with collagen-experienced HIV-1 particles, with cells from 12 independent donors highlighted that this induction was highly reproducible but that the magnitude of cytokine induction varied between cells from different donors. Induction of IL-6 and IL-8 secretion was also observed with collagen-experienced primary

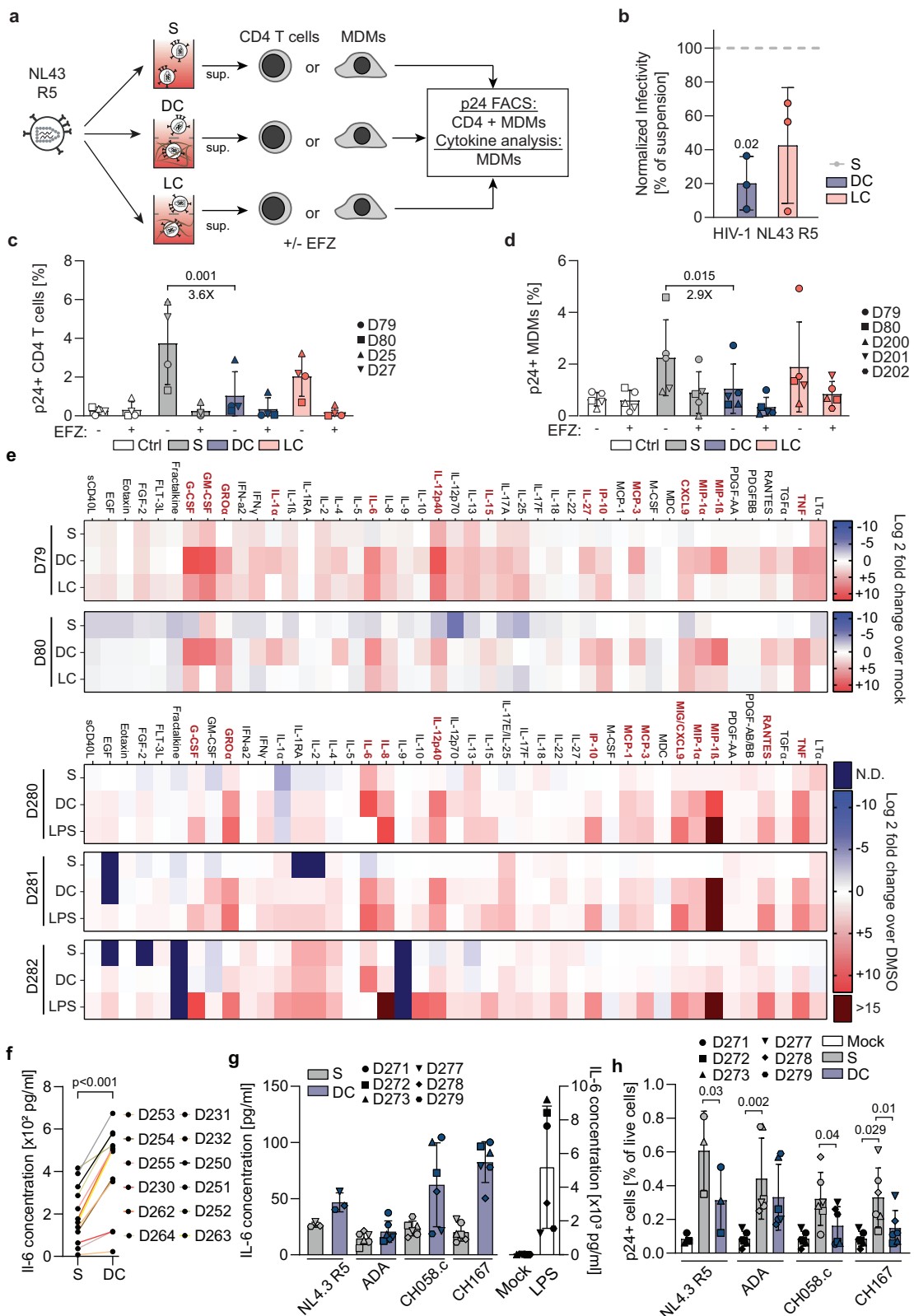

isolates ADA, CHO58.c and CH167, albeit with different efficacy and donor variability (Fig. 4g, Sup. Fig. 3g). Interestingly and in contrast to the results on TZM-bl cells, ADA was almost resistant to the infectivity reduction when the number of productively infected MDMs were assessed by flow cytometry, while the infectivity of CH058.c and CH167 was moderately but significantly reduced (Fig. 4h). Across the HIV-1 strains tested, MDM infection rates and IL-6 production were weakly

correlated (Sup. Fig. 3h, $R^2 = 0.2339$). Importantly, exposure of cells with culture supernatants of DC and LC matrices polymerized in the absence of HIV did not trigger proinflammatory cytokine secretion by MDMs (Sup. Fig. 3i) and the different 3D matrix stock solutions used did not contain endotoxins (Sup. Fig. 3j). ERVI-induced sensing is thus not triggered by soluble matrix components or endotoxin contamination. We conclude that in addition to reducing particle

**Fig. 4 | ERVI moderately restricts the infection of primary cells and sensitizes virus particles for innate immune recognition by MDMs. a** Experimental workflow. HIV-1 NL4.3 R5 virions were cultured in S or DC for 16 h. Equivalent amounts of RT units were incubated with MDMs or activated CD4 + T cells, in the presence or absence of Efavirenz. Infection levels were determined by flow cytometry after p24 staining, and the supernatants from challenged MDMs were profiled for cytokine content. **b** Normalized infectivity of HIV-1 NL4.3 R5 virions after 16h of culture as in **a**. **c** Quantification of the percentage of p24 + CD4 + T cells by flow cytometry (*n* = 4 donors). See Sup. Fig. 9b for gating strategy. **d** Quantification of the percentage of p24+ MDMs by flow cytometry. (*n* = 6 donors). See Sup. Fig. 9b for gating strategy. **e** Cytokine amounts in MDM supernatants 72 h after challenge with virions from S or DC cultures, or treated with LPS (50 ng/ml, see D280-282). Heatmaps indicate Log2 fold change of cytokine amounts in infected conditions as compared to mock-infected conditions for 2 donors. Cytokines that were induced by DC cultured virions >5-fold as compared to suspension conditions for both donors are highlighted in red. **f** Determination of IL-6 concentration from MDM supernatants. MDMs were challenged for 72 h with S or DC HIV-1 NL4.3 R5. Culture supernatants were processed for IL-6 quantification by ELISA (*n* = 12 donors). **g** Determination of IL-6 concentration from MDM supernatants. MDMs were challenged for 72 h with different viral strains harvested from S or DC cultures. Culture supernatants were processed for IL-6 quantification by ELISA (*n* = 6 donors). h. Quantification of the frequency of p24+ MDMs. Cells from **g**. were analyzed by flow cytometry after intracellular staining of p24. Results represent the mean ± SD of three independent experiments or multiple donors. Symbols indicate individual experimental repeats (b) or donors (**c**, **d**, **f**, **g**, **h**). Significance is indicated by *p*-values, and was calculated by one-way ANOVA test; Tukey's post-test (**b**, **c**, **d**), or by two-tailed ratio-paired t-test (**f**, **g**, **h**). Source data are provided as a Source Data file.

infectivity, ERVI sensitizes cell-free HIV-1 particles for specific innate immune recognition by MDMs and that HIV-1 variants differ in the extent of this sensitization.

### ERVI-induced innate immune recognition occurs during non-productive infection and depends on the Env glycoprotein and the viral genome

Multiple HIV-1 components can be recognized by different PRRs in innate immune cells in the post-entry phase of the viral life cycle (Fig. 5a). To identify the viral components of HIV particles that are recognized by cellular sensors after the collagen experience, we challenged MDMs with virions derived from S or DC cultures in the presence of selective inhibitors. These analyses focused on HIV-1 particles derived from DC matrices as they triggered the most pronounced sensing of HIV-1 particles. Interestingly, interfering with virus fusion (entry inhibitor T20) or with reverse transcription (efavirenz, EFZ) did not impair the ERVI-mediated increase in IL-6 secretion (Fig. 5b). Entry and early post-entry steps of the HIV-1 life cycle are thus not required for ERVI-mediated induction of pro-inflammatory cytokine production, suggesting that innate recognition occurs in the context of non-productive uptake of virus particles. To test for a direct involvement of the viral glycoprotein Env, we challenged MDMs from three donors with HIV-1 NL4.3 R5, a non-infectious variant lacking Env (HIV-1 ΔEnv) virions, or HIV-1 ΔEnv pseudotyped with VSVg (see Sup Fig. 4a for infectivity and glycoprotein incorporation of the two independent virus stocks used). In contrast to virions carrying HIV-1 Env, virions pseudotyped with VSVg or lacking Env did not result in the secretion of IL-6 by MDMs after a 16 h collagen experience (Fig. 5c). The Env glycoprotein is thus necessary for the ERVI-induced immune sensitization of HIV-1 particles. To test if HIV-1 gRNA is required for innate sensing of collagen-experienced HIV-1 particles, MDMs were challenged with lentiviral particles with or without gRNAs and without viral glycoprotein or pseudotyped with HIV-1 Env or VSVg following exposure to collagen or medium (see Sup. Figure 4b for verification of adequate packaging of HIV-1 Env and p24 capsid production). Control experiments confirmed that collagen contact reduced the infectivity of lentiviral particles by pseudotyped with HIV-1 Env or VSVg (Sup. Fig. 4c). Importantly, increased secretion of IL-6 after collagen imprint was only triggered by particles containing both HIV-1 gRNA and Env glycoproteins (Fig. 5d, see Sup. Fig. 4d for IL-8 production). This innate immune recognition of HIV-1 particles increased when the local concentration of virions in the 3D environment was increased (MOI 1: 2.7-fold induction for DC virus, MOI 10: 4.7-fold induction by DC) (Fig. 5e). Contrary to the effect of ERVI on virion infectivity, the capacity of the collagen matrix to sensitize HIV particles for innate immune recognition is therefore not saturable but follows a dose response in the concentration range tested. Similar to the observation with TZM-bl cells, DC and LC experience also induced the formation of HIV-1 NL4.3 R5 Vpr.Int.GFP virion aggregates detectable at the surface of MDMs (Fig. 5f,g, untreated). To test if the local concentration of virus particles

is instrumental for the induction of cytokine release, we employed again the synthetic PNFs RM8- or EF-C to induce virion aggregation in the absence of a 3D environment. PNF treatment reversed the infectivity restriction by ERVI upon infection of TZM-bl cells but not MDMs (see Sup. Fig. 4e,f) and also triggered marked virion aggregates for S, DC and LC derived HIV particles, while aggregated S-derived HIV did not elicit increased IL-6 production (Fig. 5f-h). The ability of DC-derived HIV to trigger IL-6 production was slightly reduced upon particle aggregation for MDMs from some but not all donors. Particle aggregation, therefore, is not a prerequisite for the ECM mediated sensitization of HIV particles for innate immune recognition by MDMs. Together, these results identify that the innate immune recognition of HIV-1 particles with prior contact with 3D collagen occurs in the course of non-productive infections and involves the recognition of HIV-1 Env as well as HIV-1 gRNA.

### Sensing of collagen-experienced HIV-1 virions requires TLR2 and TLR8

To assess which PRRs are involved in the recognition of collagen-experienced HIV-1 particles (Fig. 6a), we tested the effect of PRR inhibitors on MDM cytokine production in response to challenge with HIV-1 particles subjected to ECM contact. In line with the finding that fusion and post-entry steps are dispensable for sensing (Fig. 5b), inhibition of the cytoplasmic DNA sensor cGAS had no effect on the ERVI-mediated induction of IL-6 or IL-8 secretion (Fig. 6b). Similarly, TLRs 1/2, 3 and 9 were dispensable for the recognition of collagen-primed HIV particles. In contrast, inhibiting the general TLR signaling adapter MyD88[50], the endosomal TLR-8 that recognizes single stranded RNA[51], or the use of GIT27, which impairs the activity of TLR4 and TLRs2/6 fully abrogated the induction of IL-6 production by ERVI-treated HIV1 (Fig. 6b, Sup. Fig. 5a). Analyzing the full panel of secreted cytokines revealed that inhibition of either TLR8 or TLR4/TLRs2/6 was sufficient to suppress the broad induction of proinflammatory cytokine production with no significant additional inhibition upon simultaneous use of both inhibitors (Sup Fig. 5b-e). Using more selective inhibitors allowed to distinguish between TLRs2/6 and TLR4 and identified a specific requirement for TLR2, which recognizes diacylated lipopeptides[52,53] but also viral proteins[54,55] while TLR4 was dispensable (Fig. 6c). These results suggest that (i) the increased production of proinflammatory cytokines by MDMs challenged with HIV-1 particles sensitized by 3D collagen reflects the recognition of HIV-1 Env by TLR2 and gRNA by TLR8 and (ii) both sensing events occur in one linear pathway.

### Innate immune recognition of collagen-experienced HIV-1 particles triggers a specific antiviral gene expression program

Our secretome analysis (Fig. 4) provided insight into a late and cytokine-specific effector function of innate immune recognition. To assess the broad and immediate-early impact of innate immune sensitization of HIV-1 virions by tissue-like environments, we performed RNASeq transcriptomic profiling of MDMs challenged with virions

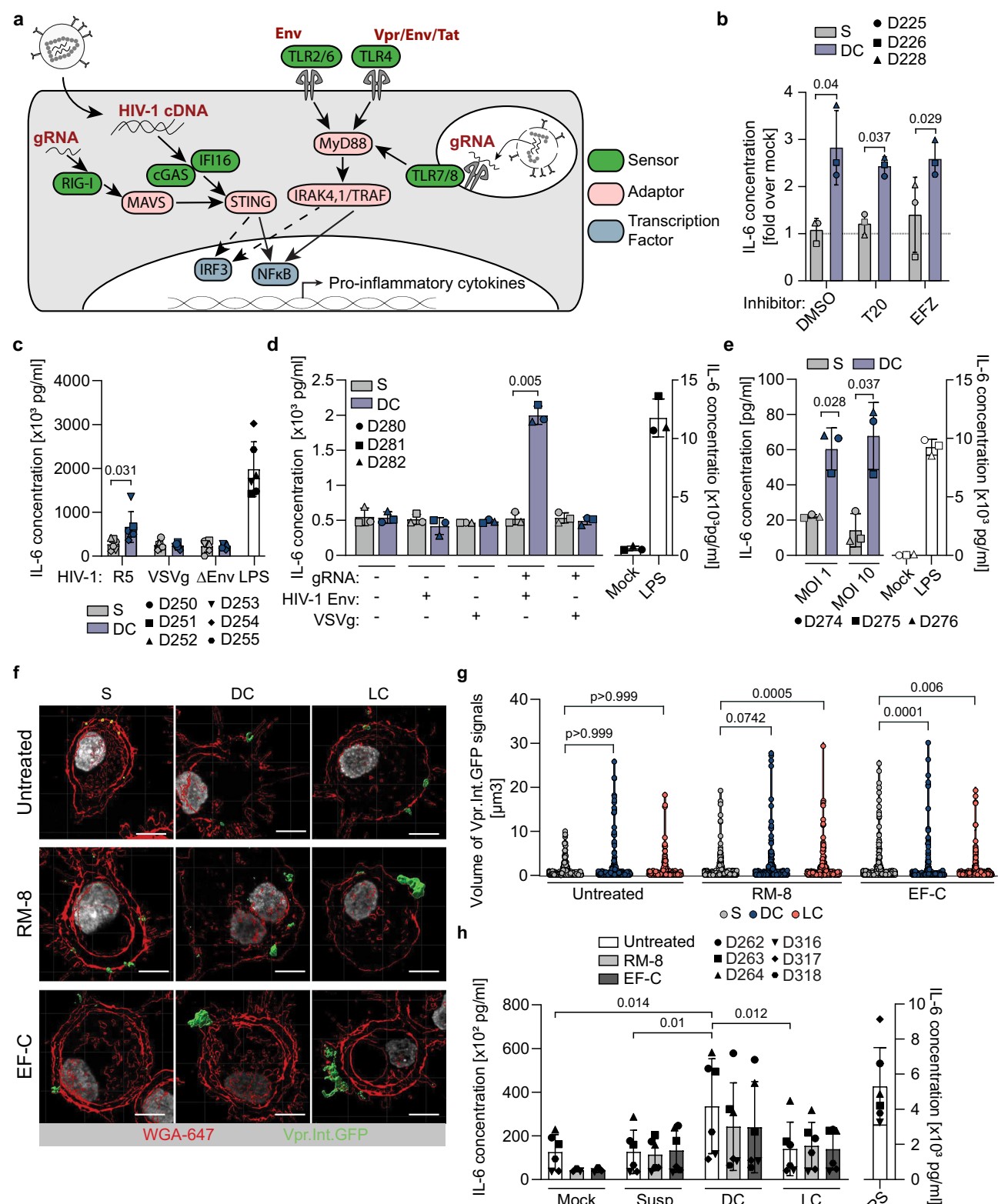

retrieved from from S or DC cultures after 16 h, for 6 h or 24 h in comparison to mock or LPS treated controls (Sup. Fig. 6a). At these timepoints, the induction of cytokine production of MDMs by collagen-primed virions was modest but significant for IL-6 (Sup. Fig. 6b). Principal component analysis (PCA) revealed that the gene expression differences obtained with specific experimental groups were more pronounced than basal donor-specific variation (Sup. Fig. 6c). Expectedly, the LPS positive control induced the most

pronounced transcriptional changes relative to mock treated controls, resulting in the deregulation of 1363 genes at 6 h (Sup. Fig. 6d: 568 downregulated genes; 795 upregulated genes). Challenging MDMs with HIV-1 particles retrieved from S cultures induced only very mild transcriptomic alterations (Sup. Fig. 6e: 10 downregulated genes, 77 upregulated genes). In sharp contrast, exposure of MDMs with HIV particles from DC cultures induced marked changes in expression of 895 genes (Fig. 6d, e: 186 downregulated genes; 709 upregulated

**Fig. 5 | The collagen-induced innate immune sensitization of HIV-1 virions is dependent on the presence of Env and the viral genome and is not induced by aggregation. a** Schematic representation of innate immune sensors involved in the detection of HIV-1-derived PAMPs. PRRs: green, adapter proteins: pink, PAMPs: red, transcription factors: light blue. **b** Determination of IL-6 concentration from MDM supernatants. MDMs were challenged with S or DC HIV-1 NL4.3 R5 in the presence or absence of the T20 or EFZ HIV-1 inhibitors. Data are normalized to mock infected condition (gray dotted line). **c** Measurement of IL-6 release by ELISA from the supernatants of MDMs challenged with NL4.3 R5, HIV-1 ΔEnv VSVg, or HIV-1 ΔEnv after culture in S or DC. **d** Measurement of IL-6 release by ELISA from the supernatants of MDMs challenged with either naked, gRNA-containing and/or VSVg/HIV-1 Env pseudotyped lentiviral particles cultured in S or DC. **e** Measurement of IL-6 release by ELISA from the supernatants of MDMs after challenge with suspension or collagen-experienced virions. Increasing amounts of virions were cultured in similar volumes in S or in DC. Equivalent amounts of RT units were then used to challenge MDMs prior to ELISA analysis. **f** Representative micrographs of MDMs challenged with differentially treated virions. Plasma membrane segmentation: red, virus segmentation: green. Scale bar: 5 μm. **g** Quantification of the volumes of segmented virions as in **f. h** Measurement of the levels of IL6 release by ELISA from the supernatants of MDMs challenged with differentially treated virions. Results represent the mean ± SD of 3 (**b, d, e**) or 6 (**c, h**) independent donors. Symbols indicate individual donors (**b-e; h**) or individual virions (**g**). Significance is indicated by *p*-values, and was calculated by two-tailed paired t-tests or Wilcoxon matched-pairs signed rank test (**b-e**), as well as Kruskal-Wallis test, Dunn's post-test (**g**) or one-way ANOVA, Tukey's post-test (**h**). Source data are provided as a Source Data file.

genes). In comparison to recognition of S-derived HIV particles, the collagen experience resulted in the additional deregulation of 588 genes (Fig. 6f: 81 downregulated genes; 507 upregulated genes). The deregulated genes could be grouped into distinct categories (Fig. 6e,g): (i) known targets of TLR2 and/or TLR8 signaling (e.g., MERTK, CD40, MMP1, MMP10 for TLR2; IL1β, IFI27, IFIT1 for TLR8, see Sup. Data 1,2), (ii) genes involved in innate immune responses but not previously linked to TLR2 or TLR8 (e.g., IL7R, CCL4, EBI3) (Fig. 6g; Sup. Data 3). and (iii) genes not previously associated with innate immune recognition. Pathway analysis demonstrated that downregulated genes were, with low confidence, involved in sterol transport, circadian rhythm, microtubule dynamics and organic acid transport. The more pronounced and significant induction of genes upon challenge with HIV from DC cultures encompassed genes involved in antiviral innate and adaptive immune processes including the GO terms negative regulation of viral genome replication, regulation of viral processes and life cycle, and defense responses characterized by humoral and innate antiviral mechanisms (Sup. Fig. 6f). Specific responses to DC-relative to S-derived HIV-1 included important elements of cytokine receptor signaling (JAK2, SOCS1) (Fig. 6e), proinflammatory cytokines and chemokines (IL1β, IL27, CXCL9, CXCL10, CXCL11, CCL1, CCL8) (Sup. Figure 6g) and regulators of inflammation (ACOD1, CD274/PD-L1) (Sup. Fig. 6g), potent antiviral restriction factors (APOBEC3A/G/B/F, IFITM1/3, MX1/2, TRIM5, GBP2/5, as well as OAS family, ISG15, ISG20) (Sup. Fig. 6h) as well as innate sensors (DDX58, AIM2, ZBP1, cGAS, IFIH1, NLRC5, NOD2, TLR3, TLR7), and adapters/effectors/regulators (MYD88, RIPK2, CASP5, RSAD2, STAT1/2/4, IRFs) (Sup. Fig. 6i). Taken together, our results demonstrate that TLR2 and TLR8- mediated recognition of collagen-experienced HIV-1 virions by MDMs elicits a specialized transcriptional profile that favors the expression of pro-inflammatory cytokines and antiviral genes.

## Collagen interactions induce a conformational change in HIV-1 Env

The above results suggested that physical contact of HIV-1 particles with collagen fibers triggers the recognition of virions by TLR2. HIV-1 Env adopts a complex trimeric structure that persists in a pre-triggered non-fusogenic conformation protected from neutralizing antibody recognition[56]. Subsequent binding to the primary receptor CD4 and co-receptor trigger rapid and pronounced structural rearrangements to facilitate fusion of the viral envelope with target cell membranes[57,58]. Binding sites for antibodies and receptors on HIV-1 are well known (Fig. 7a) and the recognition of conformation-sensitive epitopes can be used to approximate the conformational state of Env (nnAb: 17b; bnAbs: 3BNC117, 10-1074, PG16, NIH45-46). We therefore incubated suspension or DC-derived HIV-1 NL4.3 R5 particles with incorporated Vpr.Int.GFP with such antibodies or soluble CD4 (sCD4), detected these with a secondary red fluorescent reagent, and analyzed them by single particle microscopy (Fig. 7b). Accessibility of epitopes as indicated by colocalization of both fluorescent signals was significantly increased for DC-derived virions

for sCD4 and the 17b and PG16 antibodies (Fig. 7c,d), indicating conformation changes at multiple surfaces of the Env trimer. LC-derived virions, which are not efficiently sensitized for innate recognition, failed to display similar conformation changes in Env (Sup. Fig. 7a,b).

## Collagen-induced conformational changes in HIV-1 Env are associated with increased physical interaction with TLR2 and internalization of virions in TLR8-containing endosomes

To assess functional consequences of the altered Env conformation after collagen experience, we first tested if they impact the physical association of Env with TLR2 by analyzing the binding of recombinant soluble TLR2 (sTLR2) to HIV-1 particles. S or DC-derived HIV-1 NL4.3 R5 or HIV-1 NL4.3 ΔEnv virions were incubated with increasing concentrations of (sTLR2) for 30 min at 37 °C and purified by ultra-centrifugation. Residual pelleting of sTLR2 was already observed in the absence of HIV particles, but significantly higher amounts of sTLR2 were pelleted by DC but not by S-derived HIV and this effect was dependent on the presence of Env in the virus particle (Fig. 7e,f). Quantifying the colocalization of HIV-1 particles following their uptake by MDMs after 3 h of infection revealed that DC-derived HIV-1 NL4.3 R5 particles displayed significantly higher colocalization with intracellular vesicles positive for TLR8 (Fig. 7g,h) or the late endosome marker CD63 (Sup. Fig. 7c,d). A similar increase in uptake into TLR8+ endosomes following collagen encounter was observed for HIV-1 CH167, while the frequency of colocalization with TLR8 remained constant for HIV-1 ADA and CH058.c (Sup. Fig. 7e,f). Collectively, our data suggest that interactions of HIV particles with ECM induce complex conformational changes in Env that affect its fusogenicity and recognition by TLR2. The induction of TLR2 recognition promotes the recognition of HIV-1 gRNA by TLR8. Depending on the virus variant, this can involve increased internalization of particles into TLR8-positive endosomes for more frequent sensing or other mechanisms that e.g enhance the efficacy of recognition of internalized gRNA by TLR8. TLR-mediated recognition of collagen-experienced HIV-1 particles drives a specialized antiviral transcriptional gene expression program and the production of proinflammatory cytokines.

## Discussion

The initial goal of this study was to define how tissue-like environments suppress the infectivity of cell-free virus particles. Investigating the mechanism and relevance of this intrinsic antiviral property of extracellular environments, ERVI, defined that this restriction affects the function of a broad range of HIV-1 strains without affecting glycoprotein incorporation and virion morphology. Since the infectivity restriction can be overcome by using infectivity enhancing PNFs, HIV-1 particles subjected to ERVI are not defective but display reduced infectivity. It will be interesting to study the physiological role of naturally occurring PNFs in tissue in the future. Mapping the effect in the viral life cycle defined virion fusion but not binding of cell-free virus particles with target cells as the key step affected by ERVI.

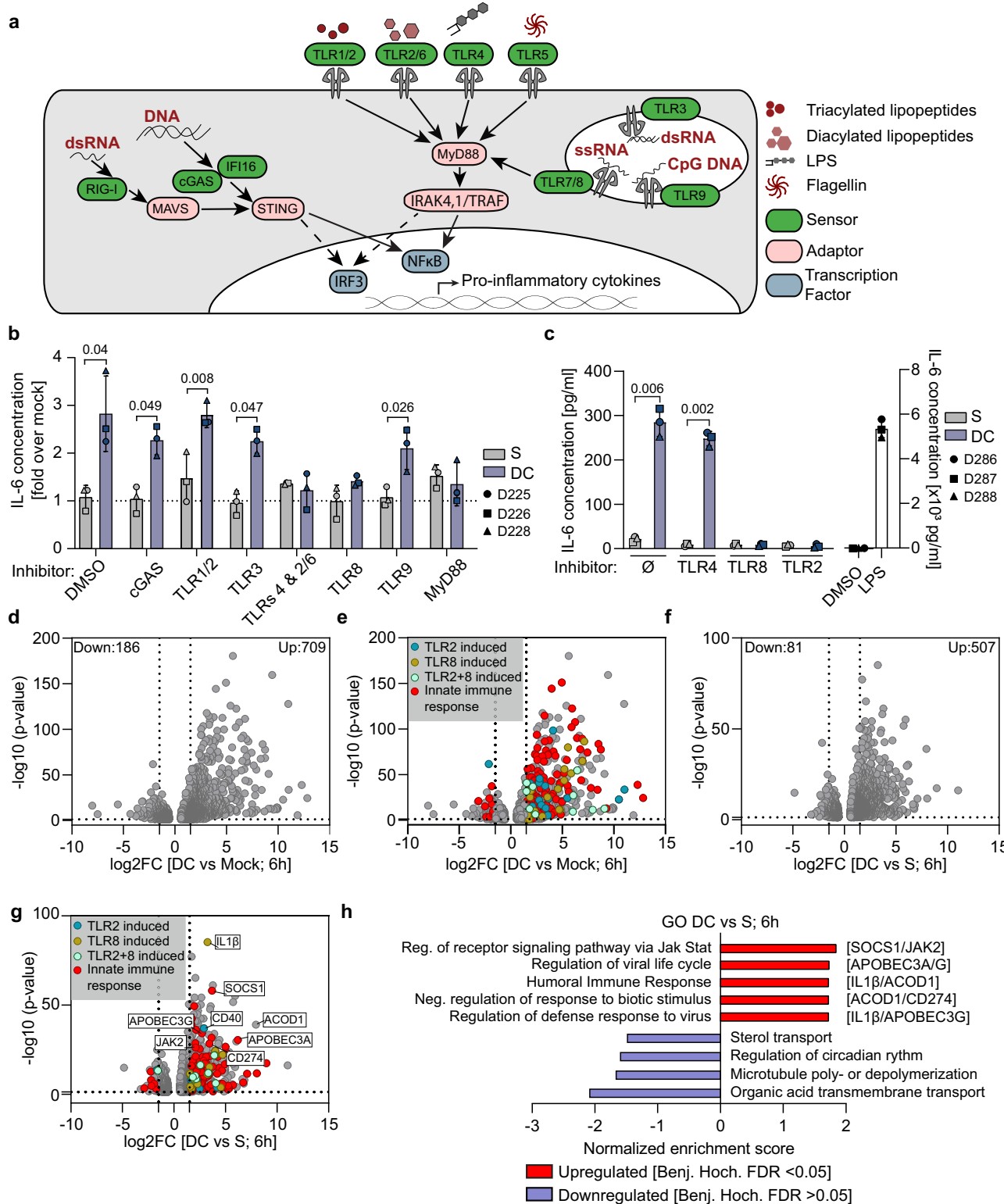

Exposure to 3D collagen induced conformational changes in Env and the extent of these changes was reflected in the magnitude of the infectivity reduction. The transient contacts with collagen fibers thus alter the architecture of Env trimers in virions to reduce their fusogenicity.

In addition to the rat and bovine type I collagen, our study revealed that other adhesive matrices such as human type III collagen or matrigel, a meshwork of basal membrane proteins extracted from mice, also exerted antiviral effects. Together with the high

conservation of mammalian collagen orthologs (e.g., 92.75% amino acid identity between the rat and bovine collagen alpha1(I) protein chains), these results indicate that the antiviral action of polymerized ECM is evolutionary conserved and are in line with the concept that these antiviral effects reflect the biophysical properties of ECM. Future work will be directed at using physiological ECMs derived from healthy and diseased HIV target tissue and encapsulated lymphoid organ-on-a-chip models to gain more insight in the physiological relevance of this tissue-intrinsic immune mechanism[5].

**Fig. 6 | Collagen experienced HIV-1 virions are sensitized for TLR2 and TLR8 recognition in MDMs, reshaping their transcriptional landscape. a** Schematic representation of PRRs and their cognate ligands. PRRs: green, adapter proteins: pink, PAMPs: red, transcription factors: blue. **b** PRR inhibitor screening. MDMs were challenged with S or DC HIV-1 NL4.3 R5 virions in presence or absence of PRR inhibitors. IL-6 release was measured by ELISA from MDM supernatants. Data is normalized to mock infected condition (gray dotted line). **c** Measurement of IL-6 release from the supernatants of MDMs challenged with S or DC HIV-1 virions in presence or absence of the indicated inhibitors. **d** Volcano plot illustrating the differential gene expression between MDMs challenged with DC virus and mock challenged cells for 6 h. The number of down- and upregulated genes are indicated. **e** Volcano plot as in d. with highlighted gene signatures. Genes induced by TLR2 alone (blue), TLR8 alone (brown), co-regulated (light green) or associated with "Innate Immune Response" genes (red). **f** Volcano plot illustrating the differential gene expression between MDMs challenged with suspension or collagen primed

HIV-1 virus for 6 h. The number of down- and upregulated genes are indicated. **g** Volcano plot as in f. with highlighted gene signatures. Genes induced by TLR2 alone (blue), TLR8 alone (brown), co-regulated (light green) or associated with Innate Immune Response genes (red). Top deregulated genes belonging to different gene ontology terms are highlighted. **h** Gene ontology analysis of biological processes repressed or induced in MDMs challenged with collagen primed virions as compared to cells challenged with suspension virus for 6 h. FDR is indicated (only upregulated pathways have a *q*-value < 0.05). Top deregulated genes for each pathway are highlighted. Results represent the mean ± SD of 3 independent donors. Symbols indicate individual donors (**b**, **c**) or individual genes (**d–g**). Significance is indicated by p-values, and was calculated by two-tailed paired t-tests or Wilcoxon matched-pairs signed rank test (**b**, **c**), differential gene expression analysis was performed using DESeq2: genes with adjusted p-value < 0.05 were considered significant (**d–g**), for GSEA analysis, *p*-values were corrected using Benjamini-Hochberg FDR (**h**). Source data are provided as a Source Data file.

Surprisingly, assessing the impact of ERVI on MDM target cells revealed a functional consequence of this extracellular restriction in addition to reducing virion infectivity: prior contact with tissue-like environments triggered innate immune recognition of HIV particles, resulting in the production of proinflammatory cytokines and expression of antiviral genes. While M1 polarized macrophages mount potent innate immune responses against HIV-1[59,60], M0 polarized MDMs only poorly sense and react when challenged with HIV-1[61]. By sensitizing HIV for the detection by M0 macrophages, ECM contacts thus license a cell type for antiviral defense that is otherwise refractory to such responses. Infection rates of collagen primed HIV particles do not correlate with cytokine production, blocking infection does not abrogate sensing, VSVg pseudotypes are reduced in their infectivity but not sensitized for innate immune recognition by ERVI, and infectivity reduction and sensitization for innate recognition differently react to alterations in matrix-particle ratios. Infectivity reduction and sensitization for innate immune recognition are thus mechanistically distinct consequences of ERVI. Our results demonstrate that this innate immune response reflects the sensing of Env by TLR2 and the viral genome by TLR8 with both sensing events being necessary for cytokine induction in the context of non-productive infections (Sup. Fig. 8). Based on our results, we propose the following mechanistic model: in the absence of ERVI, only a minority of particles lead to productive infection while the majority of particles are subject to non-productive endocytic uptake[62–65]. Some of these particles are sorted in TLR8 positive endosomes but their gRNA is not efficiently sensed and particles are not detected by TLRs at the cell surface (Sup. Fig. 8, left panel). Upon encounter of a tissue-like 3D environment, the conformational changes of Env reduce the infectivity of HIV-1 particles but promote recognition of the viral glycoprotein by TLR2, which increases sorting of particles into TLR8-positive endosomes. In dendritic cells, HIV-1 hijacks the dynein motor protein SNAPIN to evade recognition by TLR8[66] and it will be interesting to dissect if TLR2 signaling exerts similar effects in MDMs, possibly involving interactions of TLR-2 with SNAPIN[67]. However, the magnitude of the observed increase in colocalization of virions with TLR8 does not fully explain the substantial and broad induction of cytokine production and gene expression. We therefore propose that recognition of HIV-1 particles by TLR2 also has a qualitative impact on the efficiency by which TLR8 can access viral gRNA in endosomes. Such an effect of TLR2 signaling could affect the nature of the endosome, e.g., by regulation of its pH, to facilitate the access of TLR8 to gRNA. In addition, the combined signaling capacities of TLR2 and TLR8 may be required in this scenario to trigger sufficient activation of MyD88 for marked downstream signaling. TLR8-mediated sensing of HIV-1 gRNA is associated with improved control of HIV replication in patients, the regulation of latency, and can be exploited by HIV to allow the productive infection of plasmacytoid dendritic cells[68–71]. Similarly, TLR2 has been implicated in the efficacy of HIV-1 spread, latency and chronic inflammation[72].

Sensitization for innate recognition by TLR2 and TLR8 by ECM priming likely explains the role of these PRRs in HIV infection. Of note, this innate immune detection upon non-productive uptake of ECM-sensitized HIV particles is distinct from the abortive infection of CD4 T cells that causes pyroptotic cell death following detection of cytoplasmic viral cDNA and the cleavage of CARD8 by the HIV-1 protease[73]. It will be interesting to investigate how this environmental sensitization process can be targeted therapeutically, e.g., by small molecules targeting pathways that are deregulated as result of innate immune recognition of HIV particles.

Our results reveal that the short and transient physical contact of HIV particles with collagen fibers has a significant impact on the fusogenicity of the viral glycoprotein Env as well as its interactions with TLR2. The basal characterization of accessibility of epitopes in Env for antibody recognition suggests that ERVI drives conformational changes in Env with functional consequences. Since our results suggest conformational changes in Env as a key mechanism of ERVI, defining the structure-function relationship of this process will be an important goal of future studies. This will require high resolution structural analyses to define how ERVI-induced conformational changes in Env drive functional consequences in recognition by TLR2 and infectivity reduction. The comparison of Env with differential sensitivity for the two ERVI effector functions will be an asset for such analyses. Interestingly, ADA Env was entirely resistant for sensitization to innate immune recognition and several Env proteins were only mildly impaired in their ability to mediate infection. Env therefore can, in principle, evade the antiviral activities of ERVI. Considering the multiple and complex roles of Env in HIV replication and pathogenesis, such adaptation likely comes with significant fitness cost and defining the balance between the two antiviral effects and virus spread in vivo warrants further investigation. Such studies will also address if the topology of viral fusion proteins plays a role in the sensitivity to ERVI and dissect the precise relationship between ECM-induced alterations in Env conformation and Env function. Increased innate immune recognition of HIV-1 particles is also associated with the reduction of virion infectivity by the host cell factor SERINC5[48]. This restriction involves alterations in Env conformation[74,75] and both effects can be evaded by certain Env variants[76–78]. Regulation by the extracellular and intracellular adaptation of the viral glycoprotein thus emerges as an important parameter for innate immune responses to HIV-1 infection.

The finding that the infectivity reduction by ECM contacts is more pronounced upon infection of TZM-bl cells than on primary human CD4 + T cells or MDMs revealed that the extent of infectivity restriction by ERVI strongly depends on the target cell used for infection. This finding was surprising considering that TZM-bl cells are highly permissive to HIV-1 infection due to high expression levels of entry receptor and co-receptors facilitating virus entry. This result may reflect the use of different detection methods with distinct sensitivities and dynamic range of detection (luciferase assay for TZM-bl cells vs

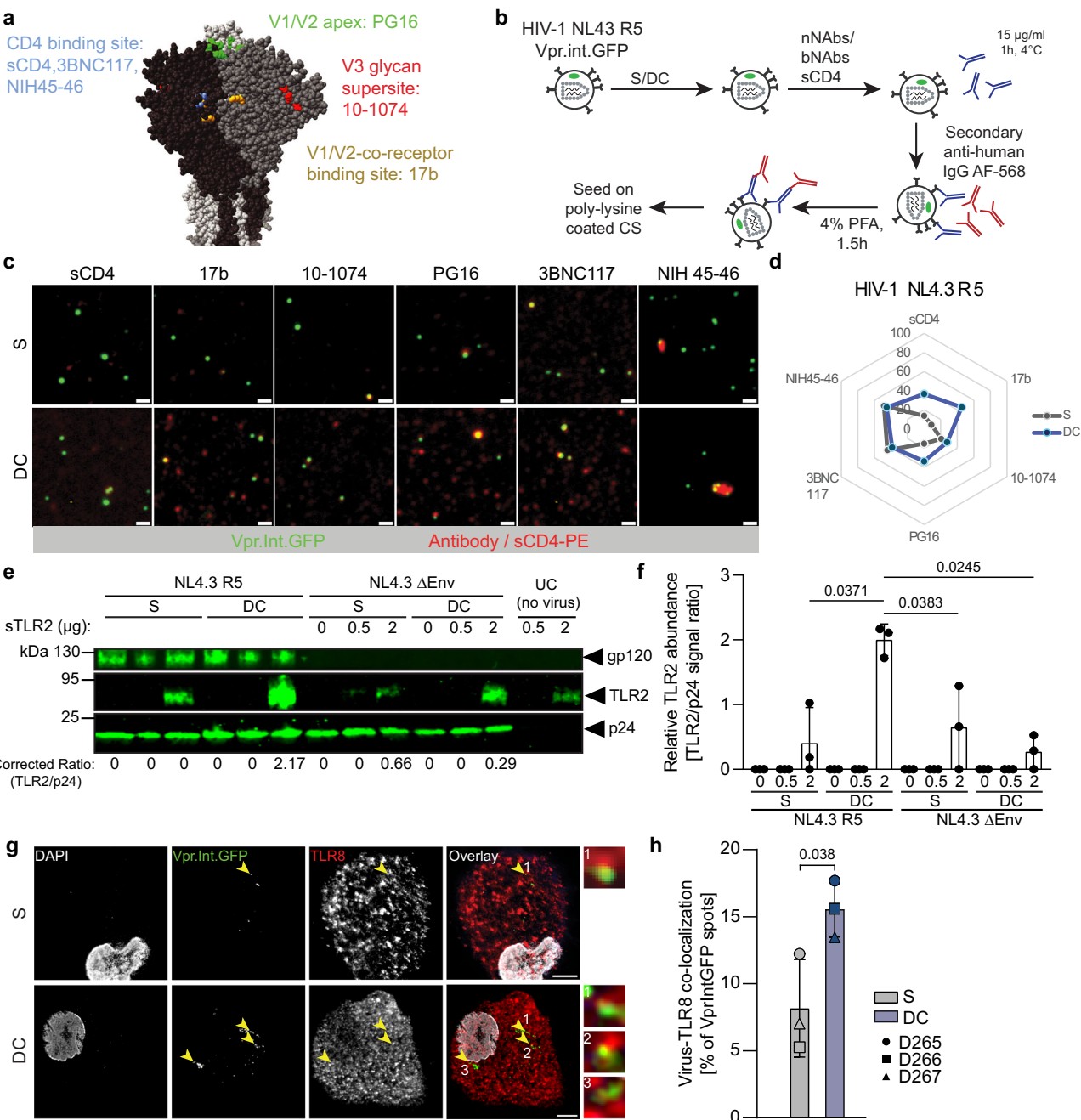

**Fig. 7 | Collagen priming induces conformational changes in HIV-1 Env and promotes association with TLR2. a** Representative structure of an HIV-1 Env trimer. Epitopes targeted by different bNAbs or nNAbs as well as sCD4 are highlighted (based upon a prediction of NL4.3 Env trimer structure using AlphaFold 3). **b** Experimental workflow. HIV-1 NL4.3 R5 Vpr.Int.GFP virions harvested from suspension or collagen cultures were sequentially incubated with primary, epitope-specific anti-gp120 antibodies, then with AF568 coupled secondary antibodies prior to processing for microscopy analysis. **c.** Representative micrographs of stained virions. Vpr.Int.GFP positive spots (green) can be seen bound by different antibodies or sCD4-PE (red). Scale bar: 0.5 μm. **d** Spider plot representing the binding frequency of antibody binding to HIV-1 NL4.3 R5 as in (c). **e.** sTLR2 is retained at the

surface of NL4.3 R5 but not NL4.3 ΔEnv virions after collagen priming. One representative Western Blot analysis showing the ratio from HIV-1 NL4.3 virions cultured in S, DC or LC. The TLR2/p24 ratios are indicated below the blots after correction, after subtraction of the TLR2 background signals from the last two lanes. **f** Quantification of western blot analyses from three independent experiments as in **e**. **g**. Representative micrographs. MDMs challenged with HIV1 NL4.3 R5 Vpr.Int.GFP virions after culture in suspension or in DC. Yellow arrows indicate Vpr.Int.GFP/ TLR8 colocalization. Scale bar: 5 μm. **h** Quantification of Virus/TLR8 colocalization from micrographs as in d. Results represent the mean ± SD of three independent experiments. Significance is indicated by *p*-values, and was calculated by two-tailed paired t-tests (**d**, **f**, **h**). Source data are provided as a Source Data file.

flow cytometry for primary cells) and/or differences in the lipid and protein environment of the entry receptor complex that may impact on the ability of ERVI to restrict virion infectivity. Our previous computational analysis had identified cell-associated infection as the predominant HIV-1 transmission mode in 3D cultures[7]. This conclusion was based on virion infectivity quantifications on model cell lines and

was now questioned by our experimental finding that the effect of ERVI on infection of primary human CD4 T cells is less pronounced. Varying the magnitude of infectivity impairment in the model however revealed that even at these lower levels of cell-free infectivity reduction, cell-associated infection remains an important transmission mode. This relevant role of cell-associated transmission likely reflects

that the long diffusion times required for virions to reach new target cells in 3D matrices are associated with significant reduction in their infectivity. This in turn raises the question why the production of large amounts of cell free virus particles is maintained in lentiviral evolution. The sensitization of non-infectious particles for the induction of pro-inflammatory cytokine responses by ECM identified in this study may be one reason since cytokines such as TNF, IL-2, IL-1 and IL-6 can increase the permissiveness of primary target cells to HIV-1 infection[79–82]. Although the infection rates of local target cells are reduced by ERVI, the induced innate recognition may create a replication prone tissue microenvironment and thereby indirectly facilitate virus spread in tissue. ERVI is exerted by 3D collagen from different species that form fibers of distinct architectures, suggesting that this restriction can be relevant in different HIV target tissues. Notably, chronic immune activation and inflammation observed in HIV patients even under therapy is associated with lymph node fibrosis, which contributes to HIV pathogenesis, e.g., by facilitating depletion of naïve CD4 + T cells[3,83]. Fibrosis reflects the deposition of large amounts of collagen with altered structural and biophysical properties[84], suggesting that the antiviral activity of ERVI is enhanced in fibrotic tissue. Since pro-inflammatory cytokines promote collagen deposition and thus fibrosis[85] and TLR2 signaling drives e.g., pulmonary, renal and cardiac fibrosis[86–88], ERVI may significantly fuel fibrosis and CD4 T cell depletion in HIV-infected lymphoid tissue.

Collectively, our study demonstrates that the biophysical properties of adhesive tissue-like environments exert a two-pronged antiviral mechanism that reduces virus spread by limiting virion infectivity and sensitizing virions for innate immune recognition. We propose that these tissue-intrinsic extracellular mechanisms synergize with the established cell-intrinsic mechanisms to create antiviral innate immunity. Important roles of the ECM have been established role in supporting efficient adaptive immune responses in lymph nodes[89], but also in the support of innate immune responses against bacterial pathogens by e.g., binding or release of PRRs, PAMPs and cytokines[90,91]. The biophysical properties of the ECM that allow the conversion of cell-free HIV-1 particles into potent PAMPs add a layer to its roles in immune regulation that acts as tissue-intrinsic arm of antiviral innate immune defense.

## Materials and methods

### Ethics statement
Our research complies with all relevant ethical regulations in accordance with the ethics committee of the Medical Faculty of Heidelberg University (S024/2022). Human peripheral blood was received from anonymous donors without any information on sex and age. Gender was not determined since sex-specific differences were not the focus of this study.

### Cells
293T cells (ATCC, CRL-3216) and TZM-bl reporter cells (NIH AIDS Reagent Program, courtesy of Dr. John C. Kappes, Dr. Xiaoyun Wu and Tranzyme Inc. (ARP-8129)) were cultured in Dulbecco's Modified Eagle's Medium (DMEM, Gibco) supplemented with 10% heat-inactivated fetal calf serum (FCS, Capricorn) and 1% penicillin/streptomycin (Gibco). Huh 7.5 cells were maintained in complete DMEM medium supplemented with non-essential amino acids.

### Primary CD4 + T cells and MDMs
Human peripheral blood of healthy, HIV-negative donors was obtained from the blood bank HD, according to regulation by local ethics committee (S024/2022). CD4 T cells were isolated from human peripheral blood of healthy, HIV-negative donors using the RosetteSep Human CD4 T cell enrichment kit (15062, StemCell Technologies) according to the manufacturer's protocol. The cells were then used a 3×3 activation as described previously[7] for 72 h and cultured in Roswell

Park Memorial Institute Medium (RPMI, Gibco) supplemented with 10% heat-inactivated FCS, 1% penicillin–streptomycin, and 10 ng/ml interleukin 2 (IL-2, Biomol). For Monocyte-derived macrophages (MDMs), peripheral blood mononuclear cells (PBMCs) were isolated from buffy coats by Biocoll (Merck Biochrom) density gradient centrifugation. CD14+ monocytes were then isolated from PBMCs by positive selection using magnetic beads (CD14 MicroBeads; 130-050201, Miltenyi Biotech) and an AutoMACS Pro Separator (Miltenyi Biotech). $1 \times 10^5$ monocytes per well were then seeded in glass-bottom 96 well plates that were previously coated with fibronectin (2 µg/cm²) and maintained at 37 °C with 5% $CO_2$ in complete RPMI in the presence of 5% human AB serum (H4522, Sigma-Aldrich) for differentiation into MDMs for 10–14 days[48].

### Viruses
Virus stocks of replication competent HIV-1 and HIV-2 strains (pNL4.3 WT, pNL4.3 ΔNef, pNL4.3 R5, ADA, CH077.t, CH058.c, CH0198, RHGA, CH0167, CH0293, HIV-2 Rod9-GFP) were generated by transfecting 293 T cells (sub-confluent 15 cm² dishes) with 25 µg of proviral constructs alongside 75 µl of linear polyethyleneimine (PEI, Sigma Aldrich) in Optimem medium (Gibco). Viruses containing Vpr-BlaM, Vpr.mRuby2 or Vpr.Int.GFP fusion proteins were similarly produced by PEI co-transfection of 293 T cells using 25 µg of pNL4.3 provirus with 7.5 µg of Vpr fusion constructs. For single-round HIV-1 virus production (pNL4.3ΔEnv VSVg, pNL4.3 ΔEnv HIV-1 Env), 22 µg of pNL4.3 ΔEnv were used, complemented with 3 µg of the respective glycoprotein expression vectors[43,48]. Two to three days post-transfection, supernatants were harvested, filtered (0.45 µm) and ultracentrifuged through a 20% (w/v) sucrose cushion. Virus pellets were then resuspended in sterile-filtered PBS 0.1% BSA, aliquoted and stored at −80 °C. All replication-competent virions were handled in a BSL-3 containment laboratory.

### Lentiviral pseudotyping
Lentiviral stocks were prepared by PEI transfection of sub-confluent 293 T cultures. Briefly, 22.5 µg of pWXPL-GFP lentiviral backbone was co-transfected alongside 2.3 µg of pAdvantage, 15 µg packaging vector psPax2 and 8 µg of envelope glycoprotein encoding plasmids (VSVg, pcDNA Con1, pcDNA JFH1, NL4.3 Env) or pcDNA3.1+ for ΔEnv particles. Lentiviral vectors carrying Vpx_mac239 were produced in 293 T cells by co-transfection of pWPI, pcDNA.Vpxmac239, pΔR8.9 NSDP, and VSV-G at a molar ratio of 4:1:3:1[48]. Cell culture supernatants were harvested and filtered 3 days post-transfection, and further concentrated through a 20% (w/v) sucrose cushion ultracentrifugation. Aliquots were stored at -80 °C. Virus titers were determined by SG-PERT analysis[92].

### Plasmids
The proviral plasmids pNL4.3 WT and ΔNef have been previously described[93]. The pNL4.3-R5 WT & ΔNef, containing 7 point mutations in NL4.3 Env were previously described[46,48]. The pNL4.3 ΔEnv, and HIV-2 Rod9-GFP proviruses were previously described[94], provirus encoding the CCR5 tropic ADA provirus were obtained via the NIH AIDS reagent program (ARP-416)[95]. All Transmitted/Founder strains were obtained via the NIH AIDS reagent program (pCH077.t (ARP-11742)[96] pCH058.c (ARP-11856)[96], pCH198[97], and chronic strains (pSTCO[98], pRHGA(ARP-12421)[98], pCH167(ARP-13544), pCH293(ARP13539)[97]). The following expression plasmids were used: pMM311 (Vpr.BlaM)[44], the pmRuby2.Vpr vector was previously described[99] and kindly provided by Dr. Tom Hope. The Vpr.Int.GFP expression plasmid was kindly provided by Anna Cereseto. The pcDNA3.1 Vpx SIVmac239-Myc and the pΔR8.9 packaging vector harboring a Vpx binding motif were previously described[94].

### Reagents and antibodies
The following dyes were used: Concanavalin A-488 (C7642, Sigma Aldrich) was used a concentration of 50 µ/ml to stain TZM-bl cell

membrane for 10 min at RT in the dark, Alexa Fluor 488 NHS Ester (A20000, Invitrogen) was used to fluorescently label loose collagen gels as described below, Fluoromount DAPI (00-4959-52, Invitrogen) was used to mount coverslips and stain cell nuclei, the CCF2-AM dye (K1023, Thermo Scientific) was used to perform HIV-1 entry assays, and WGA-647 (W32466, Invitrogen) was used according to manufacturer protocol to label MDM plasma membranes. The following reagents were used: PEI (Sigma Aldrich), Pur-A-Lyzer™ Mega 3500 kit (PURG35010, Sigma Aldrich), recombinant human TLR2 (2616-TR-050, R&D systems), and RNEasy micro kit (74004, Qiagen).

The following antibodies were used: Zombie Violet dye (Biolegend), anti-p24 KC-57 antibody (Beckman Coulter), rabbit anti-HIV-1 p24 (Kindly provided by Prof. Dr. Barbara Müller), rabbit anti-HIV-1 gp120 (Valerie Bosch, DKFZ, Heidelberg), mouse anti-TLR8 (67317, Proteintech), mouse anti-EEA1 (68065, Proteintech), mouse anti-CD63 (556019, BD), sheep anti-TGN46 (AHP500GT, BioRad), mouse anti-GPP130 (923801, Biolegend), mouse antiLAMP1 (ab25630, Abcam), mouse anti-TfR (13-6800, Invitrogen), anti-HIV-1 gp120 used to probe Env epitope accessibility after collagen priming (PG16: ARP-12150; 3BNC117: ARP-12474; 10-1074: ARP-12477; NIH45-46: ARP-12174; 17b: ARP-4091) and sCD4-PE (CD4HP2H8, ACRO Biosystems). Secondary antibodies used were anti-mouse, anti-sheep or anti-human IgG1 AlexaFluor-568 antibodies (Invitrogen), as well as goat anti-rabbit IRDye700/800 conjugated antibodies (926-32211, Rockland), and goat anti-rabbit IgG-HRP (31460, Invitrogen).

## Virus titer determination

One step Syber Green-based Product Enhanced Reverse Transcriptase assay (SG-PERT) was used to assess HIV-1 virus titers as described previously[100]. Briefly, concentrated virus stocks were first diluted in PBS, or culture supernatants were directly lysed in 2x lysis buffer (50 mM KCL, 100 mM Tris-HCl pH 7.4, 40% glycerol, 0.25% Triton X-100) supplemented with 40 mU/$\mu$l Rnase inhibitor for 10 min at room temperature. Lysed samples were then exported from BSL-3 and further diluted 1:10 in dilution buffer (5 mM $(NH_4)_2SO_4$, 20 mM KCL, 20 mM Tris-HCl pH 8). In parallel, 10 $\mu$l per well of a dilution series of a virus standard (pCHIV, $8.09 \times 10^8$ pURT/$\mu$l) were also lysed for 10 min. All lysed samples were then incubated with 10 $\mu$l of 2x reaction buffer (1x dilution buffer, 10 mM $MgCl_2$, 2x BSA, 400 $\mu$M each dATP, dTTP, dCTP, dGTP, 1pmol of each RT forward and reverse primers, 8 ng MS2 RNA, SYBR Green 1:10,000) supplemented with 0.5U of GoTaq Hot-Start Polymerase. RT-PCR reactions were carried out and read in a real-time PCR detector (CFX 96, Biorad) using the following program: (1) 42 °C for 20 min, (2) 95 °C for 2 min, (3) 95 °C for 5 s, (4) 60 °C for 5 s, (5) 72 °C for 15 s, 80 °C for 7 s, repeat steps 3–6 for 40 cycles with a melting curve read out as a final step. Sequence of the used primers are as follows: fwd RT primer (TCCTGCTCAACTTCCTGTCGAG) and rev RT primer (CACAGGTCAAACCTCCTAGGAATG).

## Virus infectivity determination

To assess the infectivity of virus stocks, TZM-bl reporter cells were infected as previously described in the absence of DEAE dextran[28]. In brief, TZMbl cells stably expressing HIV-1 entry receptors and containing luciferase and β-galactosidase genes under the control of the HIV-1 LTR promoter were infected using dilution series of concentrated virus in triplicates. 72 h post-infection, cells were fixed in 3% PFA, and incubated with a substrate solution (β-Gal supplemented with 200 μg/ml X-Gal) for 3 h at 37 °C. Blue cells were counted to determine infectious virus titers in the form of Blue Cell Units (BCUs). For the determination of the infectivity of virions after culture in suspension or in collagen, the amount of infectious units on TZM-bl cells was determined for each virus stock. The volume containing $10^5$ BCUs was then used for incubation in suspension or embedding in collagen. Supernatants of these cultures were then analyzed by the SG-PERT assay to determine their relative RT activity. Equivalent amounts of RT

units were then used to infect TZM-bl cells for assessment of the impact of the culture condition on the relative infectivity of virus particles. TZM-bl cells were infected in triplicate and lysed 3 days post-infection using 1x lysis buffer (Promega) for 10 min. Lysates were then incubated with Luciferase substrate (Promega), and luciferase activity was measured for 5 s at a Tecan Infinity luminometer.

For saturation experiments, $10^5$, $10^6$, or $10^7$ BCUs were used for seeding or embedding in collagen matrices of equivalent volumes. The culture supernatants were then processed by SGPERT. TZM-bl cells were infected using equivalent amounts of RT units for all conditions.

Virus infectivity after seeding or embedding at different concentrations was then determined by measuring the average luciferase activity from cell lysates 3 days post-infection after subtraction of the background luciferase signal from uninfected cell lysates. In additional experiments, HIV-1 NL4.3 virus was incubated on top of 2D collagen matrices or in suspension in presence or absence of increasing amounts of lentiviral particles carrying Env (Pax2+Env particles) as determined by p24 ELISA for 16 h. TZM-bl cells were then infected with equivalent amounts of RT units for 48 h prior to lysis and determination of infectivity by luciferase assay.

The relative infectivity of the respective supernatants was calculated as the average luciferase activity divided by the average amount of RT units measured from the culture supernatant.

**Virus embedding in 3D matrices.** Type I collagen matrices of different densities were polymerized as previously described[7]. In brief, dense collagen gels (3 mg/ml) were generated by mixing 10X MEM medium with 7.5% NaHCO₃ (both Gibco) and highly concentrated rat tail collagen I (354236, Corning) at a 1:1:8 ratio on ice. Concentrated virus in complete DMEM was then added to the neutralized and chilled collagen solution at a 1:1 ratio (see "Virus infectivity determination" for determination of amount of virus used for embedding). 100 $\mu$l of virus-containing collagen solutions were then distributed in each well in a 96 well-plate and allowed to polymerize within 15 min at 37 °C. Loose collagen gels (1.7 mg/ml) were generated by mixing 10X MEM medium with 7.5% NaHCO₃ (both Gibco) and PureCol bovine skin collagen I (5005, Advanced Biomatrix) at a 1:1:8 ratio on ice. Concentrated virus was then added to the neutralized and chilled collagen solution at a 1:1 ratio and and the mixture was allowed to pre-polymerize for 10 min at 37 °C.

100 $\mu$l per well was then transferred to a 96-well plate. Gels were allowed to polymerize within 45 min at 37 °C.

Human lyophilized placental type III collagen (5019, Advance Biomatrix) was reconstituted using a chilled 2 mM CH₃COOH solution on ice to reach 6 mg/ml. Type III collagen matrices (3 mg/ml) were generated by mixing 10X MEM with 7.5% NaHCO₃ (both Gibco) with the reconstituted type III collagen solution at a 1:1:8 ratio on ice. Concentrated virus in complete DMEM was then added to the neutralized and chilled collagen solution at a 1:1 ratio. Gels were allowed to polymerize within 30 min at 37 °C.

Matrigel Growth Factor Reduced Basement Membrane (354230, Corning) gels (5 mg/ml) were generated according to manufacturer protocol. In brief, concentrated Matrigel aliquots (8.9 mg/ml) were thawed at 4 °C overnight. Thawed Matrigel was then combined with complete DMEM containing concentrated virus at a 3:2 ratio on ice. 100 $\mu$l of mixture was transferred to a 96-well plate, gels were allowed to polymerize within 45 min at 37 °C.

Agarose gels were prepared by preparing a 0.8% agarose solution in PBS. The dissolved agarose solution was combined 1:1 with complete DMEM containing concentrated viruses. 100 $\mu$l of the mixture was transferred to a 96-well plate, gels were allowed to polymerize at 4 °C within 15 min.

All polymerized gels were then overlaid with 100 $\mu$l pre-warmed complete DMEM and incubated at 37 °C. Culture supernatants were harvested at the indicated time points and processed for relative infectivity determination.

## Western blotting

For detection of virion associated gp120, sTLR2 and p24 from supernatants of suspension and collagen cultures, supernatants were concentrated by ultracentrifugation through a 20% sucrose cushion (TLA-100 rotor, 108,000 $g$, 45 min) using an Optima LE-80k ultracentrifuge (Beckman Coulter), and lysed in 2× SDS sample buffer (10% glycerol, 6% SDS, 130 mM Tris Hcl pH 6.8, 10% β-Mercaptoethanol) and boiled for 5 min at 95 °C. Protein samples were resolved with SDS-PAGE electrophoresis and blotted onto nitrocellulose membranes. Membranes were blocked in 4% milk/TBST for 1 h and were incubated with primary antibodies overnight. We used secondary antibodies conjugated to IRDye700/800 (1:20,000, Rockland) for fluorescent detection with Licor (Odyssey).

## Fluorescent collagen gels

Fluorescently labeled LC gels were generated as described previously[38]. In brief, 5 mg of Alexa Fluor 647 NHS Ester dye (A20006, Invitrogen) was dissolved in 0.5 ml DMSO. The dissolved dye was then combined with PureCol bovine skin collagen (3 mg/ml) at a 1:100 ratio on ice and stirred overnight at 4 °C. After labeling, the excess dye was removed by dialysis using a 3.5 kDa molecular weight cut off Pur-A-Lyzer™ Mega 3500 kit (PURG35010, Sigma-Aldrich), placed in 1 L of acetic acid solution (0.02 N, pH 3.9) and stirred for 1 week at 4 °C. The resulting collagen solution was kept at 4 °C until further use. Fluorescently labeled collagen gels were polymerized as described above, by combining labeled and unlabeled collagen solutions at a 1:10 ratio.

## Confocal reflection microscopy

Collagen gels were imaged by confocal reflection microscopy as previously described[7,101]. Briefly, different collagen gels were generated in 15-well angiogenesis µ-slides (Ibidi) as described above. Point laser scanning confocal microscopy was then performed on a Leica SP8 microscope using an HC PL APO CS2 63×/1.4 N.A. oil immersion objective. Images were acquired using PMT detectors in reflection mode with a laser excitation at 567 nm, and a spectral detection window set between 550 nm and 570 nm wavelengths. Fluorescently stained collagen matrices were additionally imaged using a 488 nm laser.

## Widefield imaging

Viruses were embedded in stained or unstained LC matrices as described above. The culture supernatants were then harvested and ultracentrifuged through a 20% (w/v) sucrose cushion. Virus pellets were resuspended in 3% PFA for 90 min. The fixed virus particles were then seeded on 0.01% poly-L-lysine coated coverslips for 30 min and mounted on microscopy slides with Fluoromount DAPI (Invitrogen). Samples were then imaged using an epifluorescence microscope (Olympus IX81 S1F-3) under a 100× oil objective (PlanApo, N.a. 1.40). Quantification of double-positive Vpr-mRuby2 and Alexa Fluor 647 signals was performed using the Spot Detector plugin of the Icy image analysis software.

## TIRF microscopy

To exclude the possibility that collagen fragments remain attached at the surface of virions after contact with collagen fibers, HIV-1 NL4.3 R5 Vpr.Int.GFP virions were embedded in fluorescently labeled collagen for 16 h. The supernatants were then harvested and transferred to 0.01% poly-L-lysine coated µ-Slide 8 Well (80821, Ibidi). TIRF microscopy of the samples was performed on a Zeiss Axio Observer microscopy stand using an alpha PlanAPO 100×/1.46 N.A. oil objective and a Teledyne Prime-BSI sCMOS camera (6.5 µm × 6.µm pixel size). The sample was excited with a 561 nm or a 488 nm Visitron VS-laser control (Visitron Systems GmbH) and emission light was collected using Chroma TIRF Quad Line Beamsplitter with Quad Line Rejectionband 405/488/561/640 or Laser Beamsplitter zt 488 with 525/50 band-pass filter, respectively. The TIRF illumination was realized by a full 360-degree laser beam rotation at the back focal plane using 2D galvo scanners (Ring-TIRF). The evanescent field penetration depth was adjusted to 200 nm. Laser intensities, shutter, and camera were controlled using the VisiView program (Visitron Systems GmbH). The presence of AlexaFluor 647 labeled collagen fibers was assessed at the surface of Vpr.Int.GFP spots using the Icy software.

## Harvesting of collagen fibers for cryo-ET analysis

To assess the morphology of single collagen fibers by cryo ET, DC or LC gels were prepared as previously in 1.5 ml Eppendorf tubes. Polymerized gels were disrupted by sonication on a Sonorex super RK 102 H sonicator for 10 s on ice. Disrupted individual fibers were resuspended in PBS and further processed for cryo-ET.

## Plunge freezing

Collagen fibers and supernatant used for plunge freezing were collected as described above. Holey carbon grids (Cu 200 mesh, R2/1, Quantifoil®) were plasma-cleaned for 10 s in a Gatan Solarus 950 (Gatan). Samples were mixed with 10× concentrated 10 nm protein A gold (Aurion) prior plunge freezing. A total volume of 3 µl was used for plunge freezing into liquid ethane using an automatic plunge freezer EM GP2 (Leica). The ethane temperature was set to −183 °C and the chamber to 24 °C with 80% humidity. Grids were blotted from the back with Whatman™ Type 1 paper for 3 s. Grids were clipped into Auto-Grids™ (Thermo Fisher Scientific).

## Cryo-electron tomography and tomogram reconstruction

Cryo-electron tomography was performed using a Krios cryo-TEM (Thermo Fisher Scientific) operated at 300 keV and equipped with a post-column BioQuantum Gatan Imaging energy filter (Gatan) and K3 direct electron detector (Gatan) with an energy slit set to 15 eV. As a first step, positions on the grid were mapped at 8700× (pixel spacing of 10.64 Å) using a defocus of approximately -65 µm in SerialEM[102] to localize collagen fibers or HIV particles. Tilt series were acquired using a dose symmetric tilting scheme[103] with a nominal tilt range of 60° to −60° with 3° increments with SerialEM. Tilt series were acquired at target focus −4 µm, with an electron dose per record of 3 e⁻/Å² and a magnification of 33,000× (pixel spacing of 2.671 Å). Beam-induced sample motion and drift were corrected using MotionCor2[104]. Tilt series were aligned using AreTomo[105] and tomograms were reconstructed using R-weighted back projection algorithm with dose-weighting filter and SIRT-like filter 5 in IMOD[106]. Tomograms were used to measure the diameter of HIV particles in IMOD. For visualization, tilt series were aligned using protein A gold as fiducials in IMOD. Tomograms in Fig. 2 were reconstructed using R-weighted back projection algorithm with 3DCTF, dose-weighting filter and SIRT-like filter 10. In IMOD, 15 slices of the final tomogram were averaged and Fourier filtered. The diameter of HIV particles was measured from the outer leaflet of the viral membrane in IMOD. The average diameter of two measurements per particle was used in the graph.

## Infectivity enhancement experiments

HIV-1 virions were cultured in suspension or in collagen for 16 h. After performing an SG-PERT analysis from the culture supernatants, equivalent amounts of reverse transcriptase units were then incubated with the specified infectivity enhancers for 20 min at 37 °C prior to infection of TZM-bl cells. Concentrations of infectivity enhancers used: EF-C (15 µg/ml)[40] or RM-8 (15 µg/ml)[41]. The relative infectivity of the treated virions was assessed as previously described.

## Binding assay

NL4.3 Vpr mRuby2 virions were cultured in suspension or in collagen for 16 h. Viral titers in the supernatants were quantified by SG-PERT. Equivalent amounts of RT units were then used to infect TZM-bl cells seeded on glass coverslips in 24-well plates 24 h prior to infection.

After 2 h at 4 °C to prevent virion internalization, the cells were then washed with PBS, and the plasma membrane was stained with Concanavalin A Alexa Fluor-488 (Invitrogen) according to manufacturer's protocol. The samples were then fixed using 3% PFA for 90 min, and mounted on microscopy slides using Fluoromount-DAPI (Invitrogen). Spinning disk confocal microscopy was performed on a PerkinElmer UltraVIEW VoX microscope equipped with a Yokogawa CSU-X1 spinning disk head and a Nikon TiE microscope body. An Apo TIRF 60×/1.49 N.A. oil immersion objective and a Hamamatsu C9100-23B EM-CCD camera were used. Images were acquired using solid state lasers with excitation at 405 nm, 488 nm and 561 nm with matching emission filters. Z-stacks were acquired with a z-spacing of 0.5 μM steps. The percentage of cells with bound virus was then analyzed using the Imaris software (Oxford Instruments), in which Z-stacks were reconstructed in 3D. Cell membranes and virions were segmented as individual surfaces, and statistic values (surface volumes, relative distance of surfaces) were retrieved from the software.

### Vpr.BlaM entry assay

TZM-bl reporter cells were infected with an MOI of 0.1 using HIV-1 NL4.3 viruses containing Vpr-BlaM after treatment in suspension or collagen for 16 h. A concentrated virus that was not seeded in medium prior to infection was used as positive control, virus particles lacking Env were used as negative control. Fusion of HIV-1 particles was allowed to proceed for 4 h at 37 °C. Cells were then washed twice in PBS and stained with 2 mM CCF2-AM dye (Invitrogen) supplemented with 2.5 mM Probenecid in Fluorobrite DMEM 2% FCS for 6 h at 11 °C to prevent particle fusion during the staining process according to manufacturer instructions. The same workflow was used to assess fusion of S or DC virions after incubation with or without EF-C as previously described. Where indicated, T20 was added to block fusion. Cells were then trypsinized fixed in 3% PFA in PBS at 4 °C overnight and analyzed by FACS (see Sup. Figure. 9a for gating strategy).

### Primary CD4 + T cell infection

Activated primary CD4 + T cells were infected in 96-well plates in triplicate with suspension and collagen culture supernatants containing NL4.3 R5 as previously described[7]. Briefly, equivalent RT units were used between the different conditions, as quantified by SG-PERT. The amounts of RT units used between experiments were defined as previously, and amounted to $3.25 \times 10^{10}$ +/- $1.8 \times 10^{10}$ pURT per well. The cells were spininfected at 600 g for 90 min in the presence or absence of 3 μg/ml reverse transcriptase inhibitor Efavirenz (EFZ), then cultured at 37 °C for 3 days. The samples were then stained with a fixable Zombie Violet dye (Biolegend), fixed in 3% PFA for 90 min, then stained with an anti-p24 KC57 FITC antibody (Beckman Coulter) in 0.1% Triton 100-X for 30 min at 4 °C according to manufacturer protocol. Samples were then measured by flow cytometry (see Sup. Fig. 9b for gating strategy).

### Primary MDM infection

Differentiated macrophages were spin transduced at 37 °C in a preheated centrifuge with lentiviral vectors containing Vpx_mac239 for 1 h at 20 g. Within 16 h post-transduction, cells were infected in triplicates with equivalent amounts of RT units as measured by SG-PERT from suspension and collagen culture supernatants in a 96-well plate format, in the presence or absence of EFZ. Virus input for infection was determined as for CD4 + T cells. MDMs were also treated with 50 ng/ml LPS as positive control. Infected MDM culture supernatants were harvested 3 days post-infection and further processed for cytokine analysis.

5 days post-infection, cells were harvested by trypsinization, stained with a fixable Zombie

Violet dye (Biolegend), fixed in 3% PFA for 90 min, then stained with an anti-p24 KC-57 FITC antibody (Beckman Coulter) in 0.1% Triton

100-X for 30 min at 4 °C according to manufacturer protocol. Infection rates were then determined by flow cytometry (see Sup. Fig. 9b for gating strategy).

### Cytokine quantification

The amounts of cytokines and chemokines present in cell culture supernatants were determined by Eve Technologies Corporation using the Discovery Assay®: Human Cytokine Array/Chemokine Array 48-Plex. Samples were exported out of BSL-3 containment after treatment with 0.5% Triton 100-X for 30 min prior to shipping. Results are in pg/ml of cytokines/chemokines according to the company protein standard. Cell-free supernatants were also analyzed for levels of Il-6, Il-8 and TNF by enzyme-linked immunosorbent assay (OptEIA ELISA kits; BD Biosciences) according to manufacturer's instructions.

### Endotoxin level determination

To exclude potential endotoxin contamination of our agarose, collagen and matrigel stocks (batches from the same lot were used throughout the study), supernatants harvested from agarose, type I or III collagens, as well as matrigel matrices (polymerized in the absence of virus) were used for endotoxin testing. Endotoxin levels were quantified using a Pierce Chromogenic Endotoxin Quant Kit (ThermoScientific, A39553) according to manufacturer protocol using the low standard procedure.

### Modeling

Mathematical modeling has been used extensively to reveal and quantify the dynamics of viral infections by connecting experimental data and mechanistic descriptions of the assumed processes. The mathematical model that we use here has been developed previously to estimate the contribution of cell-free and cell-to-cell transmission to HIV-1 spread given different environmental conditions and has been explained in detail within[7]. In brief, the model describes the turnover and dynamics of (un-)infected CD4 + T cells, CD8 + T cells and the viral load within different culture conditions by systems of ordinary differential equations.

Uninfected CD4 + T cells proliferate and die with specific proliferation, $\lambda$, and death rates, $\delta$, respectively, with their proliferation capacity affected by the simultaneously proliferation of CD8 + T cells accounting for resource competition. To account for the various transmission modes, CD4 + T cells can get infected by either cell-free transmission, proportional to the viral load and regulated by the cell-free transmission rate $\beta_f$, or direct cell-to-cell transmission, proportional to the number of neighboring infected cells and dependent on the cell-free transmission rate $\beta_c$. The complete set of mathematical equations and detailed descriptions, as well as pre-defined and obtained parameter values used within the analyses, are given within[7]. In the original publication, the model was fitted simultaneously to the data of co-transfer experiments of infected and uninfected CD4 + T cells into 2D suspension, and 3D loose and dense collagen environments, to estimate the parameters controlling viral infection, i.e., $\beta_f$ and $\beta_c$, and infer the relative contribution of both transmission modes to overall infection. Thereby, a single transmission rate for cell-free infection was assumed across all environments, with environmental restriction reducing the infectivity of cell-free virions within collagen, i.e., the transmission parameter $\beta_f$, to only 14% of the effectivity considered within suspension, i.e., $\beta_{f,loose} = \beta_{f,dense} = \eta \beta_{f,sus}$ with $\eta$ = 0.14. In this study, we re-performed the analysis done within[7] by varying the factor accounting for reduced infectivity in collagen $\eta$ between 0 and 1 within steps of 0.1, and also considering $\eta$ = 0.275, as experimentally determined for primary target cells. Fitting was performed as described within[7] using the *optim*-function within the *R* language of statistical computing. Posterior distributions of parameter estimates were obtained by performing ensemble fits for each value of $\eta$ based on different starting values for the unknown parameters. By

using a stepwise approach incorporating subsequent filtering steps that excluded unreasonable model predictions in terms of CD4 + T cell counts and viral loads (see Sup. Fig. 2b, c), we ensured convergence of parameter estimates, with posterior distributions for parameter estimates relying on ~110-145 successful fits for each value of $\eta$.

## PRR inhibitor treatments

To determine the sensing pathway involved in the collagen-mediated sensitization of virus particles for innate immune recognition, MDMs were pretreated with the different inhibitors prior to infection. 5 μM of cGAS inhibitor (G140, Invivogen), 8 μM of TLR 1/2 inhibitor (Cu-CPT22, Selleckchem), 49 μM of TLRs 4& 2/6 inhibitor (GIT27, Tocris), 2 μM of TLR4 inhibitor (CLI-095, Invivogen), 100 μM of TLR2 inhibitor (TL2-C29, Invivogen) or 10 μM of TLR8 inhibitor (Cu-CPT9a, Invivogen) were incubated with MDMs 3 h prior to infection. The MyD88 inhibitor (Pepinh-MYD, Invivogen) was used at 20 μM and incubated with MDMs 4 h prior to infection. MDMs were also pretreated for 1 h with a TLR3/dsRNA complex inhibitor (Merck Millipore) 5 μM final concentration. MDMs were also treated with 100 μM of T20 HIV-1 fusion inhibitor (Roche) or 3 μg/ml Efavirenz (SML0536, Sigma). The different inhibitors were supplemented again during virus inoculation. The cells were then cultured for 3 days at 37 °C, and supernatants were harvested for cytokine analysis.

## RNA-seq of MDMs challenged with suspension or collagen-primed HIV-1 virions

MDMs from three separate donors were challenged with equivalent RT units of HIV-1 NL4.3 R5 virions harvested from suspension or collagen cultures incubated for 16 h at 37 °C for 6 h or 24 h prior to RNA isolation (RNeasy Micro kit, Qiagen). As controls, untreated cells, and cells treated with LPS (50 ng/ml) were cultured for 24 h at 37 °C prior to RNA isolation. RNA samples were sent to BMKgene (Beijing, China). After verification of RNA purity, concentration and integrity using Nano-Drop, Qubit 2.0 and Agilent 2100, qualified RNA was processed for library construction. The library was then sequenced by Illumina NovaSeq X high throughput sequencing. HISAT2 software was used to map clean reads to the reference genome (Homo_sapiens.GRCh38_release95.genome.fa.). StringTie contrast pairs were then used to assemble the mapped reads for subsequent analysis. Differential gene expression between samples was analyzed by DESeq2 using thresholding criteria of fold changes (FC) ≥ 1.5 and a p-value < 0.05. For pathway analysis, a Gene Set Enrichment Analysis (GSEA) was performed using the GSEA software[107]. In brief, a ranked list of differentially expressed genes for each comparison was used to perform a GSEA analysis using the MSigDB, with a specific analysis using the GO Biological Processes (C5) database[92]. From this analysis, normalized enrichment scores were calculated for each GO term, as well as the Benjamin-Hochberg false discovery rate (FDR). Raw data can be accessed from the Gene Expression Omnibus (GEO) database under the accession number GSE307977.

## HIV-1 colocalization with TLR and sub-cellular compartments

MDMs were challenged with NL4.3 R5 Vpr.Int.GFP virions for 4 h at 37 °C. The plasma membrane of MDMs was then stained using WGA-647 (Invitrogen) for 10 min at 4 °C to avoid internalization of the dye, and samples were then fixed using 3% PFA for 90 min. After export out of BSL3 containment, the cells were permeabilized using 0.1% Triton 100X for 10 min at room temperature, incubated for 30 min with blocking buffer (PBS, 5% BSA) and then with primary antibodies for 45 min at room temperature. After incubation with appropriate secondary antibodies for 30 min at room temperature, the coverslips were mounted onto microscopy slides using Fluoromount-DAPI (Invitrogen). The samples were then imaged using a Leica SP8 point laser scanning confocal microscope using an HC PL APO CS2 63×/1.4 N.A. oil immersion objective. Multichannel images were acquired

sequentially using HyD detectors or PMTs in the lightning mode. Z-stacks were acquired with 500 nm steps. Deconvoluted images were then used to segment virions and TLR/compartments using Imaris (Oxford instruments) while enabling object-object statistics. From the resulting data, the percentage of virus segments overlapping with TLR/compartment segments were calculated, using a threshold of 0.01 μm³ overlap to define colocalization.

## Probing HIV-1 Env epitope accessibility

To assess the overall conformation of Env at the surface of virions, we adapted a workflow from Staropoli et al. (2019). In brief, HIV-1 virions from various strain containing Vpr.Int.GFP fusion proteins were cultured in suspension or in collagen matrices of different densities for 16 h. The supernatants were then harvested, and the viral titers were determined by SG-PERT analysis. Equivalent amounts of pURT were then purified by ultracentrifugation over a sucrose cushion using a tabletop ultracentrifuge (TL100, Beckman Coulter). The purified virus pellet was then resuspended in filtered PBS (0.1 μm) and distributed to a 96 well-plate at $1 \times 10^{11}$ pURT/well. The virions were then incubated with the different antibodies (anti-HIV-1 gp120 used to probe Env epitope accessibility after collagen priming (PG16: ARP-12150; 3BNC117: ARP-12474; 10-1074: ARP-12477; NIH45-46: ARP12174; 17b: ARP-4091)) (final concentration: 15 μg/ml) or sCD4-PE (1:50 dilution as per manufacturer instructions) for 1 h at 4 °C in staining buffer (RMPI with 20% FCS filtered through a 0.1 μm filter). After the first incubation with broadly or non-neutralizing antibodies, the samples were then incubated with a secondary, anti-human IgG1(H + L) AlexaFluor 568 antibody (1:200 dilution; Life Technologies, in staining buffer) for 1 h at 4 °C. The samples were then fixed using 32% PFA (Thermo Sci) to reach a final concentration of 4%. After export out of BSL-3 containment, the samples were then seeded on poly-L-lysine-coated coverslips for 1 h, then mounted on microscopy slides. Samples were then imaged using an epifluorescence microscope (Olympus IX81 S1F-3) under a 100X oil objective (PlanApo, N.a. 1.40). Quantification of double-positive Vpr-mRuby2 and Alexa Fluor 568 signals was performed using the Spot Detector plugin of the Icy image analysis software. For each primary/secondary antibody pair or sCD4-PE, appropriate signal thresholds were arbitrarily defined to exclude background signals. Background fluorescence thresholds were set using the secondary antibody-only condition as a setpoint for unspecific signals. The same thresholds were used to define true positive signals when comparing virions retrieved from suspension or collagen cultures for each antibody.

## sTLR2 binding to collagen-primed virions

The association of sTLR2 with HIV-1 NL4.3 R5 or HIV-1 NL4.3 ΔEnv virions retrieved from suspension or collagen cultures was assessed by performing a western blot analysis. Virus-containing supernatants from suspension or collagen cultures were harvested after 16 h of incubation at 37 °C, and the viral titers determined by SGPERT analysis. For each virus, $10^{11}$ pURT were then either incubated alone, or with increasing concentrations (0.5 or 2 μg/ml) of sTLR2 in complete RPMI medium for 30 min at 37 °C. The samples were then overlaid on top of a 20% (w/v) sucrose cushion and ultracentrifuged using a tabletop ultracentrifuge (TL100, Beckman Coulter) for 45 min at 108,000 g at 4 °C. To determine the background pelleting of sTLR2, the recombinant protein alone was incubated in the absence of virus at 0.5 or 2 μg/ml prior to ultracentrifugation. After removal of the supernatants, viral and/or protein pellets were lysed in 2x SDS sample buffer (10% glycerol, 6% SDS, 130 mM Tris Hcl pH 6.8, 10% β-Mercaptoethanol) and boiled for 5 min at 95 °C, prior to export out of BSL-3 containment. The samples were processed for SDS-PAGE and the membranes were blotted for p24, TLR2 and gp120 detection using a Licor Odyssey system. The ratio of TLR2/p24 was calculated using Fiji, after subtraction of the background TLR2 signal obtained from the sTLR2 only condition to obtain a corrected ratio.

## Flow cytometry

Samples were measured by flow cytometry in BD FACS Celesta with BD FACS Diva Software. Compensation controls were added for each experiment. Gating was performed using FlowJo software 10.4.2 and data were processed in GraphPad Prism 8.4.3 software.

## Statistical analysis

Statistical analysis of datasets was carried out using Prism version 8.4.3 (GraphPad). For each dataset, normal distribution was tested using the Shapiro-Wilk test, and the appropriate test was used accordingly as indicated in the figure legends. Unless otherwise specified, all statistical tests were two-tailed. Statistical significance was calculated using one- or two-way ANOVA tests, as well as paired or unpaired t-tests, and Wilcoxon matched paired tests. Correction for multiple comparisons are indicated in figure legends. Significant differences are indicated with the corresponding $p$-values. Absence of $p$-values indicates a lack of statistical significance.

## Reporting summary

Further information on research design is available in the Nature Portfolio Reporting Summary linked to this article.

## Data availability

Raw and processed data from Illumina RNA-seq were uploaded to GEO under accession number GSE307977. All other data are available in the article and its Supplementary files or from the corresponding author upon request. Source data are provided with this paper.

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

## Acknowledgements

We are grateful to Katharina Morath, Swetha Ananth, Christine Selhuber-Unkel and Ada Cavalcanti-Adam for advice and discussion, to Volker Lohmann, Frank Kirchhoff and Tom Hope for sharing reagents, and to Kathrin Bajak for help with preparing the manuscript figures. The following reagents were obtained through the NIH HIV Reagent Program, Division of AIDS, NIAID, NIH: monoclonal anti-Human Immunodeficiency Virus Type 1 (HIV-1) gp120 Protein, Clone 17b (produced in vitro), ARP-409, contributed by Dr. James E. Robinson; anti-Human Immunodeficiency Virus (HIV)-1 gp120 Monoclonal Antibody (3BNC117), ARP-12474 and anti-Human Immunodeficiency Virus (HIV)-1 gp120 Monoclonal Antibody (101074), ARP-12477 contributed by Dr. Michel Nussenzweig; anti-Human Immunodeficiency Virus (HIV)-1 gp120 Monoclonal Antibody (NIH45-46 G54W), ARP-12174; anti-Human Immunodeficiency Virus (HIV)-1 gp120 Monoclonal Antibody (PG16), ARP-12150, contributed by International Aids Vaccine Initiative. This research was funded by the Deutsche Forschungsgemeinschaft (DFG, German Research Foundation) within project numbers 240245660 – SFB 1129 (project 8 to OTF, project 19 to P.C.), 316249678—SFB 1279 (project A03 to J.M.) and 533587280 - Cluster of Excellence "SynthImmune" (EXC3018/1, O.T.F. and P.C.). FG was supported by the Chica- and Heinz Schaller Foundation and the German Federal Ministry of Education and Research (BMBF, grant 031L0293A/E). O.T.F. acknowledges support by the Ministry of Science, Research and Culture Baden-Württemberg. This work was also supported by research grants from the Chica and Heinz Schaller Foundation (Schaller Research Group Leader Programme) and the DFG Heisenberg Programm (project 537227910) to P.C. We thank the Infectious Diseases Imaging Platform (IDIP) at the CIID at Heidelberg University. We would like to acknowledge access to the infrastructure and support provided by the Cryo-EM Network at Heidelberg University (HDcryoNet), which is funded and supported by the DFG, the Federal Ministry of Science, Research and Culture of Baden-Württemberg, among others, within the framework of the Excellence Strategy of the Federal and State Governments of Germany. The authors gratefully acknowledge the data

storage service SDS@hd supported by the Ministry of Science, Research, and the Arts Baden-Württemberg (MWK), the German Research Foundation (DFG) through grant INST 35/1314-846 1 FUGG and INST 35/1503-1 FUGG.

## Author contributions

Conceptualization, O.T.F.; Methodology, S.S.A., F.G., Investigation, S.S.A., L.Z., A.I., N.T., K.W.; Reagents sharing, L.R.-W.; Data analysis, S.S.A., L.Z.; Mathematical analysis, K.W., F.G.; Writing—Original Draft, O.T.F., S.S.A., L.Z. and F.G.; Writing—Review & Editing, O.T.F, S.S.A., F.G., P.C., J.M.; Funding Acquisition, O.T.F.; Resources, L.R., J.M.; Supervision, O.T.F, P.C., and F.G.

## Funding

## Competing interests

The authors declare no competing interests.
