## [Transparent Peer Review file · Nature Communications]

A tissue-intrinsic mechanism sensitizes HIV-1 particles for TLR-triggered innate immune responses

Corresponding Author: Professor Oliver Fackler

Version 1:

Reviewer comments:

Reviewer #1

(Remarks to the Author)

In this manuscript, the authors investigate how dense collagen (DC, rat tail collagen I) and loose collagen (LC, PureCol bovine skin collagen I) inhibit HIV infection. Authors showed that HIV in a collagen-rich 3D extracellular matrix (ECM) triggers IL-6 production in primary macrophages via TLR2 or TLR8. The main concern is that the physiological relevance of this 3D collagen model is questionable, as it does not recapitulate the complex environment of mucosal tissues. Moreover, it is unclear whether the observed HIV restriction is due to the collagen structure itself or an artifact, such as the shedding of HIV glycoproteins during ECM preparation at 4°C and during experimental manipulation. There are also concerns regarding the experimental design and data interpretation. Specific comments are as follows:

1. Lack of justification for model relevance: The manuscript does not adequately explain the rationale for using these two specific 3D collagen matrices or how they mimic tissue environments lacking key mucosal components, such as antimicrobial peptides and mucin. There are many different types of collagens. Would the study be specific for these two products?
2. It remains unclear whether collagen in a non-3D (soluble or 2D) form would also inhibit HIV, or whether the 3D structure itself is critical for the observed effects.
3. In the fusion study (Fig 3), the authors do not clarify whether equal amounts of virus (normalized by p24 content) were used across conditions (supernatant vs. LC or DC). Additionally, supporting evidence using electron microscopy to visualize virus localization would strengthen the conclusions. The identity of the experimental controls is also not clearly defined.
4. In Fig 4, HIV is known to induce cytokine responses in primary macrophages; however, the authors report minimal cytokine induction from viruses in supernatants. This discrepancy is not addressed.
5. Cytokines were measured on day 3 post-exposure, which may miss early innate immune responses. Measurement at earlier time points (e.g., 3–6 hours or 24 hours) would be more appropriate for capturing initial signaling events.
6. References #44 and #85 do not provide information relevant to the use of HIV NL4.3 R5 viruses and should be revised or replaced with appropriate citations to state the identity of R5.

Reviewer #2

(Remarks to the Author)

SUMMARY

This manuscript presents a compelling and well executed study that explores the biophysical and functional consequences of cell-free HIV-1 virion interaction with the three-dimensional extracellular matrix (ECM) modeling tissue-like environments, on HIV-1 pathophysiology.

The authors convincingly demonstrate that ECM components assembled into 3D scaffolds of dense collagen (DC) or low-density collagen (LC) rapidly reduce virion infectivity. The effect is mediated by transient interactions of freely moving HIV-1 glycoprotein-bearing virions with collagen fibrils. Through a combination of elegant imaging approaches and appropriate controls they show that fibrils do not "stick" to virions or change their morphology. Rather, DC- and LC-experienced virions

were found to form somewhat larger aggregates on target cells (TZM-bl model), and showed lower fusogenicity, and significantly reduced infectivity overall. The authors further demonstrated that transient interactions with collagen fibrils impacted HIV-1 Env glycoprotein conformation as evidenced by altered binding of epitope-specific anti-gp120 antibodies. Interestingly, the degree to which infectivity was reduced by virion interactions in 3D collagen matrices differed among a small panel of viruses bearing Env glycoproteins from different primary virus strains. Of note, the extent of infectivity impairment was less in primary CD4T cells in monocyte derived macrophages (MDM) for the model Env, NL4-3.R5. The authors extended their study beyond direct effect on infectivity to address whether the “priming” of virions by collagen fibril interactions had functional consequences in MDM. They found that DC additionally rendered HIV-1 particles more visible to innate immune sensing via TLR2 and/or routing to the endoplasmic TLR8. The findings are novel and significant, as they reveal a model for which “non-infectious” cell free virions play an indirect role in promoting HIV-1 infection through sensitization of the innate immune system and chronic inflammation. This is especially impactful considering that the field’s focal point has largely been on the compared efficiencies between cell-free and cell-cell infection, and not so much on the immune modulation imposed by cell free virus as a consequence of interactions with the ECM. Through robust mechanistic approaches, the study identifies collagen mediated conformational changes in Env as a trigger for TLR recognition and endosomal routing. The dual effect of limiting viral spread while inducing pro-inflammatory response, is highly relevant to tissue-level HIV pathogenesis and mucosal immunity. The study is further strengthened by the inclusion of infectious molecular clones (IMCs) of primary HIV-1 strains and the appropriate choice of reporters and molecular approaches for the varied technical methods used. The relevance of the study is further strengthened by the extension to primary cells, i.e monocyte derived macrophages, which led to novel mechanistic insights about pathogen sensing that, while designed as an antiviral mechanism, ultimately may contribute to a pro-inflammatory state that enhances susceptibility to HIV-1 overall.

Major Strengths

- The study addresses a clear and biologically important question regarding HIV-host interactions in a tissue-like microenvironment based on the previous report of extracellular restriction of viral infectivity (ERVI)
 - The experimental design is robust, with appropriate controls, biological replicates (except cytokine analyses, stated under areas of consideration below), and functional assays to support the conclusions.
 - The manuscript is clearly written, presents detailed information on the methods used, which facilitates data interpretation, .
 - Use of primary human cells (MDMs) enhances the physiological relevance.
 - The finding that abortive infections can elicit innate immune responses via Env and gRNA is both novel and mechanistically insightful.
 - The study is likely to be of broad interest to a wide array of researchers including virologist, innate immunologists, and those in tissue immunobiology.
-

CONCLUSION

This is a well-conceived and technically rigorous study that contributes meaningfully to our understanding of how the extracellular matrix can modulate viral immunogenicity. The results are of substantial interest and the manuscript is appropriate for publication following minor clarifications or elaborations as noted above.

Areas for Consideration / Minor Suggestions

The comments below do not take away from the overall assessment of the manuscript. However, addressing them would improve clarity in some areas of these very dense, very extensive data sets.

- The authors do not clarify if their TZM-bl infection protocol follows all details as described in Wei et al - was DEAE dextran included in the experiments conducted here? It can be assumed that it was not included as it may mask differences in virion infectivity, e.g. between different strain; nevertheless, a clarification would be helpful, and if DEAE dextran was included, its potential influence should be discussed.
- It would appear to be outside of the scope of the study, however ex vivo validation (e.g., tissue explants, lymphoid organ models), and/or a model that includes target cells of interest in the collagen matrices would be of interest and enhance the translational significance of the findings.
- The authors show the distinct effects ERVI has on the relative infectivity of the various primary HIV-1 env strains tested in TZM-bl cells. It would be informative to also show this for CD4+ T cells and MDM. Currently it is only shown for NL4-3.R5 that the extent of the relative infectivity loss is less in MDM (Figure4b) than TZMbl; and that the primary env strains are differentially “primed” by contact with DC to induce IL-6 release from MDM (Figure 7e).
- Use of the engineered NL4-3.R5 strain for experimentation with primary cells provides a well- defined model and is understandable given the complexity of the experiments (i.e. doing everything with several primary Envs isn’t feasible). However, it would be helpful to include some discussion of the rationale and if the use of NL4-3.R5 may have any drawbacks.

- The authors used rat tail derived and bovine skin derived collagen to generate the dense collagen (DC) and low-density collagen (LC) gels, and in Supp.Figure1b also compared it to human Type III collagen. However, they do not describe whether the rat and bovine derived collagens are essentially the same as tissue specific collagens in humans. I.e., is it of concern that the effect of non-human derive collagen on HIV-1 is measured, and not that of human-derived? The authors should discuss this to address potential concerns that the significance of their findings is lessened by the choice of collagen source. The brief mention (Line 509) doesn't adequately address this.
- In addition, Line 194-196, given n=1 each, it seems a bit of a stretch to conclude that the difference of topology of tested viral glycoproteins (Type I, II, and III) is the basis for the distinct sensitivity to EVRI – could it not just be a difference in aa sequence, irrespective of topology of viral glycoproteins?

Since MOST of the suggestions starting below with Figure1 are minor, the reviewer does not expect extensive responses but would appreciate acknowledgement that the suggestions/ corrections were considered.

Figure 1:

- on Line 129 (Results) /Line 1241 (Figure legends), it is confusing that the legend to Fig 1c refers to relative infectivity “as determined in Fig 1b”, because in 1b, relative infectivity (TZMbl Firefly activity measured in RLU per RT activity units) is further normalized to % of supernatant control, while in 1c, the actual relative infectivity values are plotted over time (the latter makes sense since it shows that the relative infectivity for the supernatant control)
- Line 130: It is surprising that adding virions to DC and immediately removing them again (time t=0) leads to an immediate loss of relative infectivity. The same is not true for LC at time t=0. Do the authors see this as relevant?
- Figure1e and 1f legends would be more clear if they read as: Line 1246: “One representative Western Blot analysis showing [...]”; Line 1248: “Quantification of Western blot analyses of n=3 independent experiments”.
- Line 150: p value p=0.009 given in the text is omitted in Figure1g, and the relative infectivity unit label does not match that given in Figure1c.
- Figure 1i: the T/F virus strain name should be corrected to CH077.t.

Suppl. Fig 1:

- Supp Figure1c; Why were the pURT inputs not normalized at time of infection? And in legend (Line 1380), why is this graph described as showing a “correlation” between pURT and residual infectivity after virions exposure to DC? The x-axis isn't truly numerical, but categorical, and thus, it is hard to draw conclusions as to whether the extent of infectivity reduction is pURT input and/or env strain dependent. It would be more informative to show a true correlation plot and or use normalized pURT for the TZM-bl assay. Similarly, Line 182 only partially addresses this question. It is not clear to the reviewer if the relative sensitivity to ERVI of different lentivirus strains can really be compared given the different pURT inputs used, and the notion that Figure1g and Supp Figure1c suggest that the infectivity moi matters for the degree of reduction of infectivity.
- Line 165 – word “Matrigel” missing?

Figure 2:

- The authors elegantly investigate if fibrils stuck to virions directly or via morphological changes affect infectivity It is , however, not explained why they conduct this experiment with just fluorescently labelled LC (Line 210), and not (also) with DC. This would be helpful.
- While change in sensitivity can be extrapolated from the graph Figure2a, it would be easier to follow the text if reported as fold-change (Line 209).
- Contrast between fluorescently labelled collagen and embedded GFP labelled virions is clear in Fig 2c/d, however, “pCHIV-GFP” is not described in the Legend, (Lines 1264-1266) or in the Results Section.
- Figure 2e convincingly shows that the lack of association between virions and collagen fibers, as well as lack of morphological aberrations to virions in suspension and the two models, however, there seems to be less virion recovery from the DC model (Figure2f) that was not addressed anywhere in the manuscript (considered insignificant?).
- It would be helpful to mention infectivity enhancing peptide names in the Legend (Lines1274-75). Does UT refer to “Untreated”? (Figure2i) Without stating the PNFs in the Figure Legend, UT could be mistakenly interpretable as one. It would also help to spell out here that PNF stands for peptide nano fibrils.
- Figure2c / Line 1265 of the legend states that yellow arrows depict virions. However, in 2c only one arrow seems associated with a GFP signal, making the shown image unconvincing without at least further discussion.
- Line 1275 / Figure2i: the y-axis label says “infectivity [RLUs]” while the legends uses “Relative infectivity of PNF treated virions” as sub-heading; it could be more clear if a subtitle like “ effect of infectivity enhancers on virion infectivity” was chosen since relative infectivity doesn't seem to have been calculated here, but rather actual RLU are plotted.

Figure 3:

- It is not clearly stated in the Results if the image on Figure3c was acquired after the 2 hr incubation or after 16 hrs. (Lines 243-248). Maybe change to: “Prior incubation of virus particles for 16 h in suspension...”?
- Figure3j: a decimal comma is used rather than a decimal point as in other figures.
- Line 255: the authors state that “The predominant action of ERVI therefore is not at the level of virus binding to target cells.” However, isn't the difference in size of virion aggregates a difference in the way how virions are bound? Could it be possible that aggregate in itself had a negative impact on fusion?

Figure 4:

- Figure 4a/b: A clear rationale is given for using primary cells (CD4+ T cells and MDMs) and an R5 tropic virus and adds strength and relevance to the manuscript. Model is clear, but for reproducibility it may be helpful to include the number of RT units used to establish infection in the Legend (Lines 1305-06) or on the Results/Methods sections. Lines 807 and 817 state only “equivalent RT units” were used.

- In Figure 4e, more PBMC donors for cytokine analyses would be more convincing; the number analyzed may be due to resource limitation. Could the sample number be discussed to comment on how representative this is among donors?

Figure 6:

- In Figure 6b, it would be interesting in future (if that's the direction the work's taking), to see if there's also gene enrichment of the purported pathway and if small molecules could augment these pathways to desirable responses?

Figure 7

- Figure 7f: the figure legend should state clearly that/if here HIV-1 CH167 virions were generated in the presence of VprInt-GFP; it would be helpful to also point out WGA-647 as an MDM membrane dye in the figure legend, not just in the Methods.
- Figure 7g, there's not enough information on the Legends about the x-axis. The reader assumes 2A and 7A to be other viruses since plotted next to ADA, but no other information to work with, also not explained in the Results Section. Could be made clearer.

Supp. Figure 2

- The legend to SuppFigure2a should clearly indicate what cells are estimated to be infected (primary cells?). Also, the overlay of "predicted" and "measured" percent infection is hard to discern, especially for the DC plot where colors are nearly indistinguishable. Please edit for clarity. (beyond that this reviewer isn't well versed in the computational methods to judge the soundness of the conclusions drawn from the computational methods)

Supp. Figure 8 – very nice!

- On Line 122-124 (Introduction), the reference for TZM-bl cells should be added (currently listed as Ref. 92, Wei et al.)

Notes on Methods Section:

- On Line 533, the authors should acknowledge the contributors of the TZM-bl cells, not just list ARP as a source.
- On Line 556, the names of two of the T/F viruses are misspelled: "CH077" should be referred to by its complete name as "CH077.t", here and throughout (e.g. Figure 1i); there is an accidental extra space in "CH0 58.c" – should be "CH058.c"
- On Line 584 in the Plasmids section, the correct plasmid names are pCH077.t and pCH058.c as per table 1 in Ref. 88; the cloning vector is not part of the plasmid name; we suggest the authors use the established nomenclature. Also, it should be pCH077.t (there is also pCH077.c, which has a single CD8 escape mutation).
- Similarly, the plasmids for the other TF and chronic IMC aren't usually listed with their cloning vectors as part of the plasmid name
- Given that age and sex may influence HIV-1 pathology and/or innate immune response, it may be worth adding this demographic data for the peripheral blood donors used to derive primary cells, if at all accessible (Line 540).
- Isolated CD4+ T cells were stimulated using the CD3/CD28 beads or 3x3 activation (CD2/CD3/CD28). It's not stated why two different stimulation protocols were used, or the context in which each was applied (Lines 542-543). Were the data comparable from these two protocols?
- It would be beneficial to be more specific about the infection conditions of the TZM-bl cells. On Line 632 is is stated that infection of TZMbl cells was "as described " by Wei et al –however, the standard TZM-bl assay includes DEAE Dextran and it isn't stated anywhere if it was include in the authors' assays; this may affect the data interpretation.
- Although later stated in Figure 7c, it may be helpful to include the antibodies listed in the Methods "reagents & antibodies" Section under the RESULTS: Probing for HIV-1 Env epitope accessibility section (Lines 900-902), and/or add a note that the listed anti HIV gp120 antibodies (Line 606) are the bnAbs and nnAbs used to probe for Env conformation
- The mathematical modelling of the extent of ERVI effects is described in detail (Line 842fwd), but is rather complex for the reader unfamiliar with such approaches
- There's an extra "and" on Line 688

Reviewer #3

(Remarks to the Author)

Here, Ahmed et al. investigate how polymerized type I and type III collagen reduce HIV infectivity in the context of model 3-D tissue extracellular matrices (ECM). The project builds on prior studies from this research team that identified what they called "environmental restrictions to virus infection" (ERVI) during 3-D cell culture studies of cell-free HIV spreading infection.

Noteworthy results include mechanistic insights into ERVI based on data demonstrating that interactions with either loose or dense collagen matrices cause reductions to the efficiency of virion-cell fusion in TZM-bl cells using the established Vpr.BlaM assay. Incubation with collagen matrices did not affect virion integrity but impacted viral glycoprotein epitope exposure, consistent with a model wherein collagen disrupts Envelope trimer conformations in ways that impact fusogenicity. In primary CD4+ T cells or macrophages, collagen effects on infectivity were less pronounced than for TZM-bl but, interestingly, the authors discovered that collagen exposure led the virus to induce inflammatory cytokine production in these cell types, tracked to pathogen receptor pathways regulated by TLR2 and TLR8 and dependent on viral Envelope and

delivery of intact viral RNA.

Taken together, the study suggests at least two highly novel effects of collagen on HIV of potential significance to the field, impacting both virion infectivity and innate immune signaling. Both findings are of potential relevance to in vivo modes of HIV spreading infection, where the role of the ECM in viral infection remains poorly understood. On the other hand, that the collagen effects were reversible by other ECM-relevant peptides and that the fusion and signaling effects in primary immune cells were fairly minor (2 to 3-fold) may make the in vivo significance of these effects hard to address in downstream studies.

Regarding rigor, overall this is an interesting but complex study with a wealth of orthogonal experimental approaches that largely support the authors' conclusions, with the following critiques and suggestions relevant to additional evidence or analysis:

1. Figures 1 vs. Figure 4. Based on the author's model for Envelope inactivation by collagen, it was surprising to see that the restrictive effects were stronger in T2M-bl cells relative to primary cells where receptor levels would be lower and immune signaling more relevant. Did not make sense to me- can these differences be better rationalized?
2. Figures 1c, 1g, 2i. Related to point #1, some of data from the more biophysical experiments was confusing. First, if collagen-Env interactions underpin the defect, then why is the rate of infectivity decay similar with or without collagen? This result did not seem consistent with an inactivating restriction (wherein rate of inactivation should be higher independent of time). Related, the authors propose a hit-and-run mechanism- it should be saturable (1g) but only at low levels of collagen and if hit-and-run is correct this would be expected to be highly time-dependent- can this be more rigorously tested? Finally, That PNF fully rescues the defect (2i) seemed to counter the argument that ERVI is hit-and-run or that collagen effects would necessarily be persistent in a more complex ECM model. However, this important observation was not further addressed/explained.
3. Figures 5 and 6. It is really interesting that ERVI immune activation requires both native Envelope and viral genome (5d) but is not blocked by either T20 (fusion inhibitor) or EFV (RT inhibitor). The authors referred to this as a need for "abortive infection" but I would suggest caution here because I thought a traditional "abortive infection" was an RT-dependent post-fusion restriction (e.g., Doitsch et al. 2014). In this case genomes must be being detected in endosomes, but more details are needed here as to the the source of the signal and what is in these gRNA-minus particles. Related, in Figure 6, the two-step mechanism for immune activation is compelling. However, there is only a two-fold difference in TLR8 compartmentalization for ERVI virions that correlates to a 2 to 3-fold difference in activation. These effects seem minor (e.g., relative to LPS) so can they be more convincingly defined in terms of dose-dependence? I also wondered if these effects would be circumvented by PNF? If so, would this challenge in vivo relevance?

A few minor comments:

1. Note that the experimental flow diagrams are often misleading, e.g., 1A, 2A, 3A, 4A the diagram makes it look like cells or viruses are being pooled in the final steps, could be improved for the reader.
2. Writing- overall good but there are several dense paragraphs- consider breaking up themes for easier readability.
3. Effects are largely convincing but need to be tempered

Reviewer #4

(Remarks to the Author)

Version 2:

Reviewer comments:

Reviewer #1

(Remarks to the Author)

My previous concern regarding the relevance of this model remains. The ex vivo models are available to validate this model. Importantly, authors minimally addressed most of my comments.

Reviewer #2

(Remarks to the Author)

REVISED MANUSCRIPT EVALUATION REPORT: REVIEWER 2 (co-reviewed with Early Career Researchers)

Summary of Revisions:

The authors have submitted a scientifically sound rebuttal in response to prior reviewer feedback. All major and minor concerns have been comprehensively addressed resulting in a well-integrated, methodically rigorous and novel study. The additional experimental data, improved figure presentation, expanded methodological transparency and thoughtful discussion of scope and limitations have substantially elevated the manuscript. The newly included evidence of collagen-mediated enhancement of virion association with TLR2 in an Env-dependent manner (New Fig 7e-f), adds important mechanistic depth to the work.

Major Revisions Addressed:

i. Clarification of Infection Conditions.

The authors confirmed that DEAE-Dextran was not used in the TZM-bl assay and clarified this in the revised Materials and Methods thereby strengthening the interpretation of strain-specific infectivity differences.

ii. Inclusion of Additional Experimental Data.

New infection data in monocyte-derived macrophages (MDMs) using multiple primary Env strains now substantiate the earlier conclusions drawn from NL4-3.R5, with expanded donor validation.

iii. Mechanistic Expansion-TLR2 Association.

Newly added analyses demonstrating that collagen experienced virions show enhanced physical association with TLR2 confirm a direct mechanistic link between ECM 'priming' and innate immune recognition.

iv. Clarification of Model Virus Rationale.

The authors provided a clear rationale for using NL4-3.R5, addressing concerns regarding model relevance.

v. Collagen Source Discussion and Cross-Species Relevance.

The revised text now discusses sequence homology across mammalian collagens and shows consistent restriction effects from human and murine collagens.

vi. Enhanced Discussion and Limitations.

The authors now articulate the challenges of adapting organotypic systems for ECM studies under BSL-3 conditions, appropriately framing these as future directions in response to reviewer's inquisition on the relevance and translation of the study to in vivo events.

Evaluation of Added Data and Revisions:

i. Scientific Soundness

The new data sets are internally consistent, well-controlled, and directly address prior gaps.

ii. Data Presentation

Figure legends, axis labels, and nomenclature have been uniformly corrected.

iii. Textual Accuracy and Style

The manuscript reads clearly, with precise methodological detail and appropriate citations.

iv. Integration

v. Newly added data and clarifications blend seamlessly into the main narrative, strengthening logical flow.

Minor Editorial Points:

All previously noted typographical, labeling, and reference issues have been corrected. Figure legends now clearly describe reagents, strains, and analytical methods. The Materials and Methods section was updated to include all relevant antibody details, plasmid nomenclature and normalization procedures.

Scientific Significance and Impact.

This work provides a novel mechanistic framework linking extracellular matrix interactions with viral structural and immunological outcomes. The demonstration that transient engagement of HIV-1 virions with collagen fibrils simultaneously restricts infectivity and promotes innate immune activation is conceptually innovative and experimentally convincing. These findings expand understanding of how tissue microenvironments shape viral pathogenesis and immune sensing.

Reviewer #3

(Remarks to the Author)

This is a revised version of a manuscript by Fackler and colleagues characterizing the effects of the extracellular matrix (ECM) on HIV infection. Incubation of virus in polymerized collagen was shown to negatively impact infectivity, and to influence how the virus causes cellular activation of innate immune signaling, detected by TLRs.

The authors were responsive to the prior review and have improved the manuscript and I only have a few remaining comments and questions, mostly regarding the effects of collagen on virion infectivity.

1. Fusion mechanism- The data are convincing that there is a detrimental effect on virion infectivity but I still find the proposed mechanism puzzling and model underdeveloped. For example, Lines 132-135 discuss the "kinetics of infectivity impairment" and say the effects are "immediate" for LC (data in 1C). A rapid or "immediate" effect still doesn't make sense to me for a saturable restriction.

2. Also, more be said about the "saturable" results (1G, lines 150-157)- first, the effects are not fully saturated. Second, how can a physical restriction work in this fashion? E.g., if you were to pretreat the collagen with virus or soluble Envelope, wash, and then add virions- would the effect go away? Still need a more reasonable explanation here.

3. I'm still puzzled why PNF was not included as a control for Figure 3's assays on Vpr-blam fusion considering that it fully rescued the defect.

Very minor:

1. Line 69- this may be a remaining instance of overstatement- "...can be controlled to complete the portfolio of experimental systems to study HIV spread.:

2. Line 132- advert = adverse?

3. Line 149- gp120 levels may have been similar but they do seem reduced for DC and LC- should be fair here in the text.

4. Line 194- not sure I understand what is being said here- shouldn't RT be correlated with infectivity if that is what is being used to determine virion infectivity in earlier experiments?

5. Throughout- please specify time points in the results section for the reader- would help to interpret comparisons.

6. Line 207- statement not necessarily correct unless dose titrations were done for all viruses?

7. Line 330: just a suggestion but this transition feels awkward.

8. Line 408 may be a better sentence for the discussion.

Reviewer #4

(Remarks to the Author)

Version 3:

Reviewer comments:

Reviewer #1

(Remarks to the Author)

My primary concern is that the authors remain resistant to establishing the relevance and validity of their model. Their 3D collagen model itself has been published in Nature Communications previously, and in this manuscript the authors incorporate innate immune response components. However, I respectfully disagree with their position regarding the challenges of validating the model. A relatively simple air-liquid culture system was described in 2009 (Nature Protocols, PMID: 19197269) and has been widely adopted in the field, with more than 130 citations in PubMed. Our laboratory has also obtained highly reproducible data from multiple donors using this approach.

Reviewer #3

(Remarks to the Author)

In this second revision, the authors have addressed my remaining comments.

Rebuttal letter:

Reference to lines refers to manuscript version with tracked changes

Reviewer #1:

In this manuscript, the authors investigate how dense collagen (DC, rat tail collagen I) and loose collagen (LC, PureCol bovine skin collagen I) inhibit HIV infection. Authors showed that HIV in a collagen-rich 3D extracellular matrix (ECM) triggers IL-6 production in primary macrophages via TLR2 or TLR8. The main concern is that the physiological relevance of this 3D collagen model is questionable, as it does not recapitulate the complex environment of mucosal tissues. Moreover, it is unclear whether the observed HIV restriction is due to the collagen structure itself or an artifact, such as the shedding of HIV glycoproteins during ECM preparation at 4°C and during experimental manipulation. There are also concerns regarding the experimental design and data interpretation.

Reply: We thank the reviewer for her/his assessment of our work. Please see below our answers to specific points raised in individual comments and how these are addressed in the revised manuscript. Since this point was not raised again in the specific points below, we reply here to the concern that the effects of 3D collagen on HIV infectivity result from shedding of HIV glycoproteins during the collagen embedding procedure. Based on our prior work that had identified that HIV particles undergo transient physical contact with collagen fibers in the 3D matrices (Imle et al., 2019), this was indeed the first hypothesis we tested once we had obtained the result that virion infectivity is impaired by the 3D collagen experience. The results from experiments testing this hypothesis are presented as Fig. 1e, f. Since the ratio of gp120:p24 for virions harvested from suspension or collagen (DC, LC) cultures following purification through a 20% (w/v) sucrose cushion is similar, ERVI is not caused by shedding of the glycoprotein. Instead, the data presented in Fig. 7 reveal that the collagen experience induces conformational changes in the viral glycoprotein associated with increased uptake into TLR8 positive endosomes. New data added to Fig. 7 (see panels e,f) now also define that the collagen-experience potentiates the interaction of the viral glycoprotein Env with TLR2. Collectively, we conclude that the biophysical properties of collagen matrices restrict HIV infection and trigger innate immune recognition by altering Env conformation.

1. Lack of justification for model relevance: The manuscript does not adequately explain the rationale for using these two specific 3D collagen matrices or how they mimic tissue environments lacking key mucosal components, such as antimicrobial peptides and mucin. There are many different types of collagens. Would the study be specific for these two products?

Reply: We fully agree with the assessment that the 3D collagen matrices we use do not recapitulate the complexity of a physiological extracellular matrix. This is in fact by design: For HIV infection studies, the complex tissue models that are available are very useful but do not allow to control and mechanistically dissect the relevance of individual parameters. Furthermore, it is impossible to link the functional characterization of a bulk population of virus particles from a cell culture supernatant to the individual contacts of virions with the tissue environment. Inspired by the seminal studies by Lämmermann and Sixt on immune cell motility, we therefore established 3D collagen cultures as a reductionist model composed of minimal components to study the biophysical impact of a 3D environment (Imle et al., 2019). This rationale is now explained in more detail in the revised manuscript (lines 69-70). Our current manuscript describes that such environments impair infectivity and enhance innate immunogenicity of HIV particles. It is of course correct that these 3D collagen matrices are not equivalent to native tissue and lack additional ECM components, which is why we refer to them as “tissue-like”. Importantly however, we observed similar effects with several types of

collagen as well as a complex ECM (human type III collagen and matrigel, see Fig. 1h). The effects we describe are thus conserved among a range of different collagens and complex ECM, and the biophysical properties of such ECM networks alone are sufficient to exert these antiviral effects. These conclusions are now emphasized more in the revised manuscript (lines 574-580). This does not preclude that additional components of ECM at anatomical sites with relevance to HIV infection exert additional effects, which will be an interesting topic of future studies and is now discussed in more detail (lines 580-582).

2. It remains unclear whether collagen in a non-3D (soluble or 2D) form would also inhibit HIV, or whether the 3D structure itself is critical for the observed effects. Reply: 2D collagen can indeed impair the infectivity of HIV particles, these results are shown as the “on top of collagen condition” in Fig. 1d. This effect is less pronounced than in 3D, where adhesive matrices of very different architecture exert potent antiviral effects (Fig.1h, Sup. Fig. 1b). We also attempted to incubate cell-free virions with soluble collagen, however this reagent is provided dissolved in 0.02 N acetic acid, whose cytotoxicity precluded this type of analysis. Importantly, this solvent is no longer present in our experiments with 3D matrices and the supernatant of these 3D cultures does not contain detectable antiviral activity (Fig. 1d (see “collagen supernatant + virus” condition)). We also digested already polymerized matrices using type I collagenase and incubated the digested matrices with virions. However, the remaining collagenase in these mixtures detached the reporter cells used for infectivity determination and thus precluded the assessment of virion infectivity. This is now explained in the revised text (lines 142-144).

3. In the fusion study (Fig 3), the authors do not clarify whether equal amounts of virus (normalized by p24 content) were used across conditions (supernatant vs. LC or DC). Additionally, supporting evidence using electron microscopy to visualize virus localization would strengthen the conclusions. The identity of the experimental controls is also not clearly defined.

Reply: As presented in the material and methods section as well as the figure legend, equivalent amounts of virions as quantified for reverse transcriptase activity by SG-PERT assay were used for suspension and collagen conditions (DC, LC). Experimental controls are now better described in the revised text. Use of the Vpr-Blam entry assay, the gold standard assay for quantitative analysis of virion fusion, unambiguously identified that prior collagen experience impairs the fusogenicity of virions. We do not feel that additional EM characterization is required to further support this conclusion.

4. In Fig 4, HIV is known to induce cytokine responses in primary macrophages; however, the authors report minimal cytokine induction from viruses in supernatants. This discrepancy is not addressed.

Reply: The reviewer is correct, cytokine responses of MDMs exposed to “suspension HIV particles” is often more pronounced in previous reports than in our case. This likely reflects differences in the differentiation protocols employed for MDM generation. While strong cytokine responses to HIV challenge are typically observed with M1 MDMs differentiated by IFN- γ +/- LPS or GM-CSF, the differentiation by human AB serum we employed results in M0 polarization that is associated with poor pro-inflammatory cytokines responses after challenge with native HIV (see new references 59 & 60). Importantly, TLR activation by HIV-1 in such MDMs is particularly low (see new ref. 61). The sensitization of HIV particles for innate recognition by M0 MDMs thus provides an additional antiviral mechanism that exploits a host cell subset that is typically not very actively involved in host defense. We now specifically discuss this point (lines 587-590).

5. Cytokines were measured on day 3 post-exposure, which may miss early innate immune responses. Measurement at earlier time points (e.g., 3–6 hours or 24 hours) would be more appropriate for capturing initial signaling events.

Reply: We thank the reviewer for this important comment that triggered us to generate new data that significantly improved the manuscript. To address the reviewer's point, we conducted a transcriptome analysis of MDMs 6 and 24 hours post challenge with HIV (see Fig. 6d-h, Sup. Fig 6, lines 459-499). The results reveal that collagen priming of HIV particles results in prominent transcriptional changes compared to cells challenged with suspension HIV. Deregulated genes include many antiviral factors. Overlap with known TLR2 and TLR8 signatures is partial, suggesting that collagen priming triggers a specialized TLR-mediated gene expression program.

6. References #44 and #85 do not provide information relevant to the use of HIV NL4.3 R5 viruses and should be revised or replaced with appropriate citations to state the identify of R5.

Reply: Thanks for pointing this out, these errors have been corrected.

Reviewer #2:

SUMMARY

This manuscript presents a compelling and well executed study that explores the biophysical and functional consequences of cell-free HIV-1 virion interaction with the three-dimensional extracellular matrix (ECM) modeling tissue-like environments, on HIV-1 pathophysiology. The authors convincingly demonstrate that ECM components assembled into 3D scaffolds of dense collagen (DC) or low-density collagen (LC) rapidly reduce virion infectivity. The effect is mediated by transient interactions of freely moving HIV-1 glycoprotein-bearing virions with collagen fibrils. Through a combination of elegant imaging approaches and appropriate controls they show that fibrils do not "stick" to virions or change their morphology. Rather, DC- and LC-experienced virions were found to form somewhat larger aggregates on target cells (TZM-bl model), and showed lower fusogenicity, and significantly reduced infectivity overall. The authors further demonstrated that transient interactions with collagen fibrils impacted HIV-1 Env glycoprotein conformation as evidenced by altered binding of epitope-specific anti-gp120 antibodies. Interestingly, the degree to which infectivity was reduced by virion interactions in 3D collagen matrices differed among a small panel of viruses bearing Env glycoproteins from different primary virus strains. Of note, the extent of infectivity impairment was less in primary CD4T cells in monocyte derived macrophages (MDM) for the model Env, NL4-3.R5.

The authors extended their study beyond direct effect on infectivity to address whether the "priming" of virions by collagen fibril interactions had functional consequences in MDM. They found that DC additionally rendered HIV-1 particles more visible to innate immune sensing via TLR2 and/or routing to the endoplasmic TLR8. The findings are novel and significant, as they reveal a model for which "non-infectious" cell free virions play an indirect role in promoting HIV-1 infection through sensitization of the innate immune system and chronic inflammation. This is especially impactful considering that the field's focal point has largely been on the compared efficiencies between cell-free and cell-cell infection, and not so much on the immune modulation imposed by cell free virus as a consequence of interactions with the ECM. Through robust mechanistic approaches, the study identifies collagen mediated conformational changes in Env as a trigger for TLR recognition and endosomal routing. The dual effect of limiting viral spread while inducing pro-inflammatory response, is highly relevant to tissue-level HIV pathogenesis and mucosal immunity. The study is further strengthened by

the inclusion of infectious molecular clones (IMCs) of primary HIV-1 strains and the appropriate choice of reporters and molecular approaches for the varied technical methods used. The relevance of the study is further strengthened by the extension to primary cells, i.e monocyte derived macrophages, which led to novel mechanistic insights about pathogen sensing that, while designed as an antiviral mechanism, ultimately may contribute to a pro-inflammatory state that enhances susceptibility to HIV-1 overall.

Major Strengths

- The study addresses a clear and biologically important question regarding HIV-host interactions in a tissue-like microenvironment based on the previous report of extracellular restriction of viral infectivity (ERVI)
- The experimental design is robust, with appropriate controls, biological replicates (except cytokine analyses, stated under areas of consideration below), and functional assays to support the conclusions.
- The manuscript is clearly written, presents detailed information on the methods used, which facilitates data interpretation.
- Use of primary human cells (MDMs) enhances the physiological relevance.
- The finding that abortive infections can elicit innate immune responses via Env and gRNA is both novel and mechanistically insightful.
- The study is likely to be of broad interest to a wide array of researchers including virologist, innate immunologists, and those in tissue immunobiology.

CONCLUSION

This is a well-conceived and technically rigorous study that contributes meaningfully to our understanding of how the extracellular matrix can modulate viral immunogenicity. The results are of substantial interest and the manuscript is appropriate for publication following minor clarifications or elaborations as noted above.

Reply: We would like to thank this referee for her/his particularly careful review of the manuscript and the many constructive comments. We were glad to learn that the quality and impact of our study was appreciated - addressing the points raised and implementing the requested changes helped to significantly improve the manuscript. In addition to the modifications made in response to comments by all reviewers, we also included new data to show that the collagen experience potentiates the physical association of TLR2 with HIV particles in an Env-dependent manner (new Fig. 7e,f).

Areas for Consideration / Minor Suggestions

The comments below do not take away from the overall assessment of the manuscript. However, addressing them would improve clarity in some areas of these very dense, very extensive data sets.

- The authors do not clarify if their TZM-bl infection protocol follows all details as described in Wei et al - was DEAE dextran included in the experiments conducted here? It can be assumed that it was not included as it may mask differences in virion infectivity, e.g. between different

strain; nevertheless, a clarification would be helpful, and if DEAE dextran was included, its potential influence should be discussed.

Reply: The reviewer is correct, the original study that we refer to uses DEAE dextran to enhance infection of TZM-bl cells while our infections were all conducted in the absence of DEAE dextran. This is now specified in the revised materials and methods (line 816).

- It would appear to be outside of the scope of the study, however ex vivo validation (e.g., tissue explants, lymphoid organ models), and/or a model that includes target cells of interest in the collagen matrices would be of interest and enhance the translational significance of the findings.

Reply: We fully agree with the reviewer that it would be great to validate our findings in an organoid or organotypic model system. From our viewpoint, tonsil explants or cultures would be ideal for this and are established in the lab. However, both systems are subject to infusion of cell culture media, creating unphysiological liquid flows. The difficulty arising from this is that it is impossible to know for both, virus particles added to the culture or produced in the culture as result of prior infection, whether they had the chance to undergo physical contact with ECM. For this, we would need an encapsulated lymphoid organ that produces virus in the inside and then transports particles to the periphery. Establishing such a microfluidics-supported organ-on-chip setting in a BSL-3 environment is a long-term goal of the lab, but we are nowhere near. The issue is now discussed in the revised manuscript (lines 580-582).

- The authors show the distinct effects ERVI has on the relative infectivity of the various primary HIV-1 env strains tested in TZM-bl cells. It would be informative to also show this for CD4+ T cells and MDM. Currently it is only shown for NL4-3.R5 that the extent of the relative infectivity loss is less in MDM (Figure 4b) than TZM-bl; and that the primary env strains are differentially “primed” by contact with DC to induce IL-6 release from MDM (Figure 7e).

Reply: As requested, we added new experimental data on MDM infection rates of the primary isolates for which we had investigated ECM priming and moved the parallel analysis of infection rates and IL-6 production to Fig. 4 where the sensitization for innate recognition is first described for NL4.3 R5. These isolates had been selected to represent high, intermediate and low sensitivity to the infectivity restriction exerted by contact with collagen as determined by TZM-bl infection. The results reveal that (i) similarly to the NL4.3 R5 strain, the extent of infectivity restriction of the primary strains tested is less pronounced with MDMs than with TZM-bl cells as targets and (ii) the induction of IL-6 is observed with primary HIV-1 strains, however with varying efficacy. In addition, they also demonstrate that the sensitivity of individual HIV-1 variants to the infectivity reduction strongly depends on the type of target cell used. As suggested by the reviewer, it is thus important to compare infection and cytokine production from the identical infection system. These results are now described and discussed in detail (lines 369-384).

- Use of the engineered NL4-3.R5 strain for experimentation with primary cells provides a well-defined model and is understandable given the complexity of the experiments (i.e. doing everything with several primary Envs isn't feasible). However, it would be helpful to include some discussion of the rationale and if the use of NL4-3.R5 may have any drawbacks.

Reply: The rationale for using the NL4.3 R5 virus was that we initially sought to assess the impact of CCR5 usage with a virus that varies minimally from the NL4.3 WT virus with which the phenomenon was identified. CCR5 usage of NL4.3 R5 is associated with only 7 point

mutations in *env* relative to NL4.3 WT. This consideration is now described in the revised text (line 310-311). To exclude that the induction of proinflammatory cytokines from MDMs is only observed with lab-adapted strains, several primary isolates have now been included (see fig. 4 g-h).

- The authors used rat tail derived and bovine skin derived collagen to generate the dense collagen (DC) and low-density collagen (LC) gels, and in Supp.Figure1b also compared it to human Type III collagen. However, they do not describe whether the rat and bovine derived collagens are essentially the same as tissue specific collagens in humans. I.e., is it of concern that the effect of non-human derive collagen on HIV-1 is measured, and not that of human-derived? The authors should discuss this to address potential concerns that the significance of their findings is lessened by the choice of collagen source. The brief mention (Line 509) doesn't adequately address this.

Reply: The reviewer raises an interesting point. Mammalian collagen alpha1 (I) proteins are highly similar, with e.g. 92.7% and 97.5% sequence identity of the *Rattus Norvegicus* and *Bos Taurus* orthologs to the *Homo Sapiens* ortholog. Our results using type III human placental collagen and Matrigel (mouse) confirm that collagen from a variety of species is in principle able to exert a restriction on the infectivity of HIV-1. We therefore conclude that this antiviral activity of collagen is not subject to major species barriers. We do however agree that further work will be required to assess this aspect in more detail and to identify the exact antiviral molecular determinants of collagens. This is now explained in more detail in the revised text (lines 574-580).

- In addition, Line 194-196, given n=1 each, it seems a bit of a stretch to conclude that the difference of topology of tested viral glycoproteins (Type I, II, and II) is the basis for the distinct sensitivity to EVRI – could it not just be a difference in aa sequence, irrespective of topology of viral glycoproteins?

Reply: We agree, this conclusion cannot really be drawn from this limited data set. We now emphasize that the observed sensitivities to ERVI represent intrinsic properties of these glycoproteins that may include their topology and that further research will be required to define the underlying determinants (lines 216-218 and 646-648).

Since MOST of the suggestions starting below with Figure1 are minor, the reviewer does not expect extensive responses but would appreciate acknowledgement that the suggestions/corrections were considered.

Reply: all comments were considered and respective changes made. For completeness and editorial review, we still provide a detailed reply to each point.

Figure 1:

- on Line 129 (Results) /Line 1241 (Figure legends), it is confusing that the legend to Fig 1c refers to relative infectivity “as determined in Fig 1b”, because in 1b, relative infectivity (TZMbl Firefly activity measured in RLU per RT activity units) is further normalized to % of supernatant control, while in 1c, the actual relative infectivity values are plotted over time (the latter makes sense since it shows that the relative infectivity for the supernatant control)

Reply: The description in the legend to Figure 1b, as well as other graphs throughout has been changed to “Normalized infectivity” to more adequately describe what is quantified and represented.

- Line 130: It is surprising that adding virions to DC and immediately removing them again (time $t=0$) leads to an immediate loss of relative infectivity. The same is not true for LC at time $t=0$. Do the authors see this as relevant?

Reply: We feel this likely reflects that physical interactions of particles with collagen fibers are more frequent in DC than in LC, which results in a longer time for the onset of infectivity reduction in LC. This is now explained in the revised text (lines 138-140).

- Figure 1e and 1f legends would be more clear if they read as: Line 1246: "One representative Western Blot analysis showing [...]"; Line 1248: "Quantification of Western blot analyses of $n=3$ independent experiments".

Reply: modified as suggested

- Line 150: p value $p=0.009$ given in the text is omitted in Figure 1g, and the relative infectivity unit label does not match that given in Figure 1c.

Reply: The appropriate p value was added to the text and the figure, and the unit label has been corrected.

- Figure 1i: the T/F virus strain name should be corrected to CH077.t.

Reply: modified as suggested

Suppl. Fig 1:

- Supp Figure 1c; Why were the pURT inputs not normalized at time of infection? And in legend (Line 1380), why is this graph described as showing a "correlation" between pURT and residual infectivity after virions exposure to DC? The x-axis isn't truly numerical, but categorical, and thus, it is hard to draw conclusions as to whether the extent of infectivity reduction is pURT input and/or env strain dependent. It would be more informative to show a true correlation plot and or use normalized pURT for the TZM-bl assay. Similarly, Line 182 only partially addresses this question. It is not clear to the reviewer if the relative sensitivity to ERVI of different lentivirus strains can really be compared given the different pURT inputs used, and the notion that Figure 1g and Supp Figure 1c suggest that the infectivity moi matters for the degree of reduction of infectivity.

Reply: We apologize if this was not sufficiently clear in the previous version. The experimental procedure for the culture of virus in suspension or collagen, and the subsequent infection of TZM-bl cells is a three-step process: 1- we determine how many infectious units our virus stocks contain (infection of TZM-bl reporter cells). 2- 1×10^5 blue cell units are embedded in collagen or seeded in suspension cultures per condition. 3- In the supernatants of these cultures, RT activity is quantified by the SG-PERT assay and used to normalize TZM-bl cell infections that serve to determine the relative infectivity in these supernatants. Reflecting intrinsic differences in their basal infectivity, different amounts of virus had to be used to determine their infectivity after culture in suspension or collagen. However, since within the comparison of different experimental conditions for each virus variant, identical amounts of RT units were for used for infection, we feel that these results allow us to conclude on the relative impact of suspension or collagen culture on their infectivity. This is now explained in greater detail in the revised manuscript (lines 199-202 + added paragraph to material and methods lines 823-828).

- Line 165 – word “Matrigel” missing?

Reply: has been corrected

Figure 2:

- The authors elegantly investigate if fibrils stuck to virions directly or via morphological changes affect infectivity. It is, however, not explained why they conduct this experiment with just fluorescently labelled LC (Line 210), and not (also) with DC. This would be helpful.

Reply: A protocol to perform fluorescent staining of loose collagen readily exists (see reference 38) but is difficult to apply to dense collagen. Collagens are provided in solution by the manufacturer, but this solution is far less viscous for loose than for dense rat tail collagen, which is provided at a much higher concentration. Given that the staining procedure for loose collagen involves a dialysis step that lasts already one week at 4°C to remove the unbound NHS-ester dye, we did not attempt to optimize this procedure for dense, rat tail collagen, which would require a much extended time frame for removal of unbound dye. This is now mentioned in the revised text (lines 235-237).

- While change in sensitivity can be extrapolated from the graph Figure 2a, it would be easier to follow the text if reported as fold-change (Line 209).

Reply: we added the fold-change to the text and figure.

- Contrast between fluorescently labelled collagen and embedded GFP labelled virions is clear in Fig 2c/d, however, “pCHIV-GFP” is not described in the Legend, (Lines 1264-1266) or in the Results Section.

Reply: Thanks for pointing this out, the appropriate virus is now cited (HIV-1 NL4.3 R5 Vpr.Int.GFP).

- Figure 2e convincingly shows that the lack of association between virions and collagen fibers, as well as lack of morphological aberrations to virions in suspension and the two models, however, there seems to be less virion recovery from the DC model (Figure 2f) that was not addressed anywhere in the manuscript (considered insignificant?).

Reply: This is indeed a consistent observation in our experiments. We think this again reflects the different densities of these matrices, where the higher density in DC slows down the release of virion into the supernatant. A discussion of this point has been added (lines 247-249).

- It would be helpful to mention infectivity enhancing peptide names in the Legend (Lines 1274-75). Does UT refer to “Untreated”? (Figure 2i) Without stating the PNFs in the Figure Legend, UT could be mistakenly interpretable as one. It would also help to spell out here that PNF stands for peptide nano fibrils.

Reply: The figure legends have been adjusted accordingly.

- Figure 2c / Line 1265 of the legend states that yellow arrows depict virions. However, in 2c

only one arrow seems associated with a GFP signal, making the shown image unconvincing without at least further discussion.

Reply: We thank the reviewer for noticing the poor contrast of the images, the micrograph has been replaced by one with an enhanced contrast making the GFP signal more visible.

• Line 1275 / Figure2i: the y-axis label says “infectivity [RLUs]” while the legends uses “Relative infectivity of PNF treated virions” as sub-heading; it could be more clear if a subtitle like “ effect of infectivity enhancers on virion infectivity” was chosen since relative infectivity doesn’t seem to have been calculated here, but rather actual RLU are plotted.

Reply: The figure legend has been updated accordingly.

Figure 3:

• It is not clearly stated in the Results if the image on Figure3c was acquired after the 2 hr incubation or after 16 hrs. (Lines 243-248). Maybe change to: “Prior incubation of virus particles for 16 h in suspension...”?

Reply: The text has been modified to clarify the timing of sample preparation and image acquisition.

• Figure3j: a decimal comma is used rather than a decimal point as in other figures.

Reply: has been corrected

• Line 255: the authors state that “The predominant action of ERVI therefore is not at the level of virus binding to target cells.” However, isn’t the difference in size of virion aggregates a difference in the way how virions are bound? Could it be possible that aggregate in itself had a negative impact on fusion?

Reply: This is an attractive hypothesis that we also pursued when we first observed these aggregates. However, LC also reduced virion infectivity without inducing any aggregation that is detectable at this resolution. We therefore conclude that aggregation is at least not a strict requirement for the effect. This is now discussed in lines 284-286)

Figure 4:

• Figure 4a/b: A clear rationale is given for using primary cells (CD4+ T cells and MDMs) and an R5 tropic virus and adds strength and relevance to the manuscript. Model is clear, but for reproducibility it may be helpful to include the number of RT units used to establish infection in the Legend (Lines 1305-06) or on the Results/Methods sections. Lines 807 and 817 state only “equivalent RT units” were used.

Reply: Virus input for these infections was normalized to contain 10^5 infectious units as determined on TZM-bl reporter cells. Parallel assessment of the RT activity of these virus stocks revealed that this corresponded to 3.25×10^{10} pURT per infection on average. Since these experiments were conducted with several independent virus stocks, the infectivity per RT activity ratio naturally varied slightly. Average and range is now provided in the revised text (lines 1003-1004).

• In Figure 4e, more PBMC donors for cytokine analyses would be more convincing; the

number analyzed may be due to resource limitation. Could the sample number be discussed to comment on how representative this is among donors?

Reply: We in fact conducted a full cytokine profiling with cells from 5 donors, but results for cells from three donors were presented in the supplement in the initial version of the manuscript. To avoid the impression that we present preliminary results, we now included the results from the cells from all five donors in revised Fig. 4. In addition, we present as new Fig. 4g analysis of IL-6 secretion as determined by in house ELISA from cells from twelve donors as independent confirmation that collagen-experienced HIV particles trigger proinflammatory cytokine production by MDMs in a highly reproducible manner.

Figure 6:

- In Figure 6b, it would be interesting in future (if that's the direction the work's taking), to see if there's also gene enrichment of the purported pathway and if small molecules could augment these pathways to desirable responses?

Reply: We conducted a transcriptome analysis of MDMs 6 and 24 hours post challenge with HIV (see Fig. 6d-h, Sup. Fig 6, lines 390-432). The results reveal that collagen priming of HIV particles results in prominent transcriptional changes compared to cells challenged with suspension HIV. Deregulated genes include many antiviral factors. Overlap with known TLR2 and TLR8 signatures is partial, suggesting that collagen priming triggers a specialized TLR-mediated gene expression program. The potential to therapeutically shape the transcriptional deregulation by ERVI by small molecules is now mentioned in the revised text (lines 627-630).

Figure

7

- Figure 7f: the figure legend should state clearly that/if here HIV-1 CH167 virions were generated in the presence of VprInt-GFP; it would be helpful to also point out WGA-647 as an MDM membrane dye in the figure legend, not just in the Methods.

Reply: The figure legend has been modified to clarify that Vpr.Int.GFP was indeed packaged into CH167 virions. We also now indicate that WGA-647 was used to stain the plasma membrane.

- Figure 7g, there's not enough information on the Legends about the x-axis. The reader assumes 2A and 7A to be other viruses since plotted next to ADA, but no other information to work with, also not explained in the Results Section. Could be made clearer.

Reply: We thank the reviewer for noticing this error, the adequate names are now cited to replace the 2A and 7A nomenclature (see revised Fig. 7, panel j).

Supp.

Figure

2

- The legend to SuppFigure2a should clearly indicate what cells are estimated to be infected (primary cells?). Also, the overlay of "predicted" and "measured" percent infection is hard to discern, especially for the DC plot where colors are nearly indistinguishable. Please edit for clarity. (beyond that this reviewer isn't well versed in the computational methods to judge the soundness of the conclusions drawn from the computational methods)

Reply: The figure legend has been adjusted to clarify which cell types are considered for the modeling. The colors have also been changed to display "predicted" and "measured" values with more clarity.

Supp. Figure 8 – very nice!

• On Line 122-124 (Introduction), the reference for TZM-bl cells should be added (currently listed as Ref. 92, Wei et al.)
Reply: The appropriate reference has been added.

Notes on Methods Section:

• On Line 533, the authors should acknowledge the contributors of the TZM-bl cells, not just list ARP as a source.

Reply: The appropriate contributors are now acknowledged.

• On Line 556, the names of two of the T/F viruses are misspelled: “CH077” should be referred to by its complete name as “CH077.t”, here and throughout (e.g. Figure 1i); there is an accidental extra space in “CH0 58.c” – should be “CH058.c”

Reply: has been corrected

• On Line 584 in the Plasmids section, the correct plasmid names are pCH077.t and pCH058.c as per table 1 in Ref. 88; the cloning vector is not part of the plasmid name; we suggest the authors use the established nomenclature. Also, it should be pCH077.t (there is also pCH077.c, which has a single CD8 escape mutation).

Reply: Thanks for making us aware of the adequate nomenclature, the plasmid names have been corrected.

• Similarly, the plasmids for the other TF and chronic IMC aren't usually listed with their cloning vectors as part of the plasmid name

Reply: has been corrected.

• Given that age and sex may influence HIV-1 pathology and/or innate immune response, it may be worth adding this demographic data for the peripheral blood donors used to derive primary cells, if at all accessible (Line 540).

Reply: This would indeed be interesting information but the ethics vote under which we obtain buffy coats does not allow such information to be disclosed.

• Isolated CD4+ T cells were stimulated using the CD3/CD28 beads or 3x3 activation (CD2/CD3/CD28). It's not stated why two different stimulation protocols were used, or the context in which each was applied (Lines 542-543). Were the data comparable from these two protocols?

Reply: Thanks for spotting this mistake - we used the 3x3 activation protocol (based on PHA and OKT3 supernatant coating of activation plates) to match the activation procedure used in our previous publication (Imle et al, 2019) and corrected the text accordingly.

• It would be beneficial to be more specific about the infection conditions of the TZM-bl cells.

On Line 632 it is stated that infection of TZMbl cells was “as described” by Wei et al –however, the standard TZM-bl assay includes DEAE Dextran and it isn’t stated anywhere if it was included in the authors’ assays; this may affect the data interpretation.

Reply: has been corrected.

- Although later stated in Figure 7c, it may be helpful to include the antibodies listed in the Methods “reagents & antibodies” Section under the RESULTS: Probing for HIV-1 Env epitope accessibility section (Lines 900-902), and/or add a note that the listed anti HIV gp120 antibodies (Line 606) are the bnAbs and nnAbs used to probe for Env conformation

Reply: We thank the reviewer for pointing this out, we now mention that the listed anti-HIV-1 gp120 antibodies were used for evaluation of gp120 epitope accessibility, and also mention it in the results part (lines 520-521).

- The mathematical modelling of the extent of ERVI effects is described in detail (Line 842fwd), but is rather complex for the reader unfamiliar with such approaches

Reply: The text was simplified and extended to increase accessibility and understanding.

- There’s an extra “and” on Line 688

Reply: has been corrected

Reviewer #3:

Here, Ahmed et al. investigate how polymerized type I and type III collagen reduce HIV infectivity in the context of model 3-D tissue extracellular matrices (ECM). The project builds on prior studies from this research team that identified what they called “environmental restrictions to virus infection” (ERVI) during 3-D cell culture studies of cell-free HIV spreading infection.

Noteworthy results include mechanistic insights into ERVI based on data demonstrating that interactions with either loose or dense collagen matrices cause reductions to the efficiency of virion-cell fusion in TZM-bl cells using the established Vpr.BlaM assay. Incubation with collagen matrices did not affect virion integrity but impacted viral glycoprotein epitope exposure, consistent with a model wherein collagen disrupts Envelope trimer conformations in ways that impact fusogenicity. In primary CD4+ T cells or macrophages, collagen effects on infectivity were less pronounced than for TZM-bl but, interestingly, the authors discovered that collagen exposure led the virus to induce inflammatory cytokine production in these cell types, tracked to pathogen receptor pathways regulated by TLR2 and TLR8 and dependent on viral Envelope and delivery of intact viral RNA.

Taken together, the study suggests at least two highly novel effects of collagen on HIV of potential significance to the field, impacting both virion infectivity and innate immune signaling. Both findings are of potential relevance to in vivo modes of HIV spreading infection, where the role of the ECM in viral infection remains poorly understood. On the other hand, that the collagen effects were reversible by other ECM-relevant peptides and that the fusion and

signaling effects in primary immune cells were fairly minor (2 to 3-fold) may make the in vivo significance of these effects hard to address in downstream studies.

Regarding rigor, overall this is an interesting but complex study with a wealth of orthogonal experimental approaches that largely support the authors' conclusions, with the following critiques and suggestions relevant to additional evidence or analysis:

Reply: We are grateful for this detailed assessment and very positive evaluation of our manuscript, its novelty and scientific rigor.

Major points:

1. Figures 1 vs. Figure 4. Based on the author's model for Envelope inactivation by collagen, it was surprising to see that the restrictive effects were stronger in TZM-bl cells relative to primary cells where receptor levels would be lower and immune signaling more relevant. Did not make sense to me- can these differences be better rationalized?

Reply: Thanks for pointing out that there is need for further clarification on this point. It is important to emphasize that our data clearly uncouple the impact of ECM on infection and immune signaling /innate immune recognition and that in the timing of our experiment, cytokine responses of MDMs may impact infection rates of bystander cells but do not affect the infection process of the initial target cell and thus do not directly amplify effects of ERVI on virion infectivity. However, TZM-bl cells are more susceptible than primary activated CD4+ T cells or MDMs on a per particle basis and it is indeed counterintuitive that the infectivity reduction by ERVI is less, rather than more, pronounced on the less effective infection process of primary cells. This could on the one hand reflect differences in the dynamic range of the detection method used to quantify infection in these cell types (Luciferase expression by TZM-bl cells vs intracellular p24 by FACS for primary cells). Alternatively, the specific lipid and protein environments of these various cell types may differentially impact on how the conformational changes induced by the collagen experience affect virion infectivity. This is now discussed in the revised manuscript (lines 654-662).

2. Figures 1c, 1g, 2i. Related to point #1, some of data from the more biophysical experiments was confusing. First, if collagen-Env interactions underpin the defect, then why is the rate of infectivity decay similar with or without collagen? This result did not seem consistent with an inactivating restriction (wherein rate of inactivation should be higher independent of time). Related, the authors propose a hit-and-run mechanism- it should be saturable (1g) but only at low levels of collagen and if hit-and-run is correct this would be expected to be highly time-dependent- can this be more rigorously tested? Finally, That PNF fully rescues the defect (2i) seemed to counter the argument that ERVI is hit-and-run or that collagen effects would necessarily be persistent in a more complex ECM model. However, this important observation was not further addressed/explained.

Reply: The reduction of infectivity we observe with time in fact reflects two independent processes: on the one hand, the infectivity of HIV particles kept in suspension is well known to steadily decrease with time (see reference 29). In addition, we describe here the effects of physical contact with ECM, which are implemented very rapidly. Once the ERVI-mediated reduction is implemented, the classical slow reduction of infectivity acts similarly on HIV particles irrespectively of their prior ERVI experience. Importantly, the first time point we analyzed (t=0) reflects a 10 min contact with collagen and we found it difficult to gain further time resolution. However, in our previous study (Imle et al, 2019), we demonstrated by using

probabilistic tracking of fluorescent reporter virions that within 5 min, 7.4% of all virus particles transiently interact with loose collagen fibers. It is thus probable that within one hour, most particles will have interacted with collagen fibers, explaining the initial steep drop in infectivity.

With respect to the PNFs, we would like to point out that the two peptides we use in this study are artificial peptides that do not occur naturally, and as such are not present in physiological ECM (EF-C is a HIV gp120 derived peptide, RM-8 is a derivative of IL-18). We employed these only as a tool to augment virion infectivity and to induce particle aggregation to mimic one of the effects we observed upon culturing HIV particles in 3D collagen. PNFs at high concentration as tool to assess if enhancing interactions of ERVI-experienced virions can rescue their infectivity. These experiments allowed us to conclude that ERVI significantly reduces virion infectivity but does not render them completely non-infectious and that particle aggregation is not the active principle by which collagen encounters sensitize HIV particles for innate immune recognition. These explanations are now added to the text (lines 420-430). We do not know if peptides with infectivity enhancing properties are present in complex ECM. If so, they do not prevent the antiviral activity of Matrigel (Fig. 1h) at this concentration and configuration. Naturally occurring PNFs with activity towards HIV have been identified in bodily fluids and their role in tissue remains to be explored, which is now mentioned in the revised text (lines 430-431).

3. Figures 5 and 6. It is really interesting that ERVI immune activation requires both native Envelope and viral genome (5d) but is not blocked by either T20 (fusion inhibitor) or EFV (RT inhibitor). The authors referred to this as a need for “abortive infection” but I would suggest caution here because I thought a traditional “abortive infection” was an RT-dependent post-fusion restriction (e.g., Doitsch et al. 2014). In this case genomes must be being detected in endosomes, but more details are needed here as to the the source of the signal and what is in these gRNA-minus particles. Related, in Figure 6, the two-step mechanism for immune activation is compelling. However, there is only a two-fold difference in TLR8 compartmentalization for ERVI virions that correlates to a 2 to 3-fold difference in activation. These effects seem minor (e.g., relative to LPS) so can they be more convincingly defined in terms of dose-dependence? I also wondered if these effects would be circumvented by PNF? If so, would this challenge in vivo relevance?

Reply: Thanks for pointing this out, in the sense of the discussion of how bystander CD4 T cell depletion occurs, the term “abortive infection” has indeed been prominently used to describe uptake and sensing of particles, driving cells into pyroptosis prior before viral expression can be detected. This is an important distinction to our scenario where HIV particles are uptaken into endosomes in a pathway that is generally non-productive in the sense that it cannot lead to productive infection of the target cell. We also did not observe any cytotoxicity associated with the uptake of collagen experienced HIV particles. We therefore now use the term non-productive infection throughout and explain how this differs from pyroptosis-inducing abortive infection (lines 624-627).

To generate lentiviral particles without viral genome, HEK293T cells were transfected using the lentiviral packaging vector pPax2 that encodes for the HIV-1 Gag and Pol polyproteins. Transfection of this plasmid alone is sufficient to trigger the release of particles that do not contain any viral genomic RNA. To assess for the role of Env and viral genomic RNA, an Env expression plasmid and/or the pWXPL-GFP lentiviral backbone that contains the viral genome including packaging signal was added during vector production. Challenge of MDMs with lentiviral particles of different composition revealed that both viral RNA and HIV Env glycoproteins were required for the ERVI induced sensing, while none of the lentiviral

pseudotypes produced in the absence of viral genomic RNA induced IL-6 release by MDMs. This is now explained with greater care in the text (lines 408-414).

We agree that the magnitude of the observed effects when looking at an individual cytokine is moderate and feel that the real functional impact of this phenomenon lies in the complex deregulation of the integrated cytokine milieu. In addition, the new transcriptome analysis (new Fig. 6) revealed that the effects of innate immune activation go far beyond cytokine production. It is also correct that significant amounts of particles are already internalized in TLR8+ endosomes without prior collagen experience and that the increased targeting to this compartment following collagen contact is in the range of two-fold for HIV-1 NL4.3 R5. We therefore proposed that immune sensitization likely also includes a yet to be determined mechanism by which collagen priming facilitates the recognition of viral gRNA once delivered into TLR8+ endosomes (lines 610-615). This is now corroborated by the analysis of primary HIV-1 strains that all induce increased levels of IL-6 following collagen contact, are all routed to TLR8+ endosomes but differ in the extent to which the frequency of colocalization with TLR8 is increased by tissue-like environments (Fig 7g,h, Sup. Fig. 7 e-f, lines 543-546). Moreover, we now include new data to show that prior collagen contacts increase the physical association of HIV particles with TLR2 in an Env dependent manner (Fig. 7e,f)

We also included a dose-response experiment which revealed that innate sensing of ECM-primed particles is not further boosted by increased amounts (Fig. 5, panel e) and tested if PNFs affect the sensing of ECM-primed particles (original manuscript: Sup. Fig. 6, panel e). In addition to the 3 donors we had tested in the original manuscript, we now included 3 more donors in our analysis. While the artificial aggregation of suspension derived virions does not lead to any appreciable induction of IL-6 release by MDMs, the incubation of DC-derived virus with PNFs slightly reduced the ERVI induced sensitization for some but not all donors, likely indicating that PNFs may indeed somewhat alleviate the ERVI induced sensitization of virions in a donor dependent manner, but the exact mechanism remains elusive. Regarding the in vivo relevance of the data obtained using the PNFs, it is important to note that none of the PNFs used in our study (EF-C, RM-8) are present in physiological conditions: these are artificial peptides that we used here as tools to induce particle aggregation and their effects in our experiments cannot be used to conclude on the physiological impact of naturally occurring PNFs on tissue-mediated intrinsic immunity. These conclusions are now detailed in the revised manuscript (lines 430-431) (also see reply to point 2).

A few minor comments:

1. Note that the experimental flow diagrams are often misleading, e.g., 1A, 2A, 3A, 4A the diagram makes it look like cells or viruses are being pooled in the final steps, could be improved for the reader.

Reply: We thank the reviewer for pointing this out, we modified the mentioned schematics accordingly.

2. Writing- overall good but there are several dense paragraphs- consider breaking up themes for easier readability.

Reply: Long and dense paragraphs were subdivided with separate headers to enhance readability.

3. Effects are largely convincing but need to be tempered

Reply: We carefully revised the text throughout to remove any potential overstatements.

Reviewer #4 (Remarks to the Author):

Reply: Thank you very much for your efforts in coupling the review of our manuscript with training of the next generation of scientists.

Reviewer #1 (Remarks to the Author):

My previous concern regarding the relevance of this model remains. The ex vivo models are available to validate this model. Importantly, authors minimally addressed most of my comments.

Reply: Reviewer1 dismisses the relevance of our model system without providing any specific reasons for this assessment. In our previous rebuttal letter, we provided a detailed reply to all concerns raised in the first round of review to explain how already included as well as newly added experimental data addressed these issues together with a large number of textual changes. Since no reference is made to our reply and revisions, it is difficult for us to assess which aspects of this revision were not satisfactory and why. As per editorial request, we address in this rebuttal letter again in detail the issue of additional controls to better contextualise the model (original comment 1 of Reviewer 1) and the concerns about 2D and 3D collagen matrices and whether this is altered by different types of collagen (original comment 2 of Reviewer 1). Edits made in response to the other comments are briefly listed below.

Original Comment 1. Lack of justification for model relevance: The manuscript does not adequately explain the rationale for using these two specific 3D collagen matrices or how they mimic tissue environments lacking key mucosal components, such as antimicrobial peptides and mucin. There are many different types of collagens. Would the study specific for these two products?

Reply: We fully agree with the assessment that the 3D collagen matrices we use do not recapitulate the complexity of a physiological extracellular matrix. This is in fact by design: For HIV infection studies, the complex tissue models that are available are very useful but do not allow to control and mechanistically dissect the relevance of individual parameters. Furthermore, it is impossible to link the functional characterization of a bulk population of virus particles from a cell culture supernatant to the individual contacts of virions with the tissue environment. Inspired by the seminal studies by Lämmermann and Sixt on immune cell motility, we therefore established 3D collagen cultures as reductionist model composed of minimal components to study the biophysical impact of a 3D environment (Imle et al., 2019). This rationale is now explained in more detail in the revised manuscript (lines 64-69). Our current manuscript describes that such environments impair infectivity and enhance innate immunogenicity of HIV particles. It is of course correct that these 3D collagen matrices are not equivalent to native tissue and lack additional ECM components, which is why we refer to them as “tissue-like”. Importantly however, we observed similar effects with several types of collagen as well as a complex ECM (human type III collagen and matrigel, see Fig. 1h). The effects we describe are thus conserved among a range of different collagens and complex ECM, and the biophysical properties of such ECM networks alone are sufficient to exert these antiviral effects. These conclusions are now emphasized more in the revised manuscript (lines 554-560). This does not preclude that, at anatomical sites with relevance to HIV infection, additional ECM components may exert further effects, which will be an interesting topic of future studies and is now discussed in more detail (lines 560-582).

Original Comment 2. It remains unclear whether collagen in a non-3D (soluble or 2D) form would also inhibit HIV, or whether the 3D structure itself is critical for the observed effects.

Reply: 2D collagen can indeed impair the infectivity of HIV particles, these results are shown as the “on top of collagen condition” in Fig. 1d. This effect is less pronounced than in 3D, where adhesive matrices of very different architecture exert potent antiviral effects (Fig. 1h, Sup. Fig. 1b), suggesting that a 3D architecture is not absolutely required for the antiviral effects, however contributes to the potency of the effect. To address the question if soluble collagen exerts antiviral effects, we incubated cell-free virions with soluble collagen. However, this reagent is provided dissolved in 0.02 N acetic acid, whose cytotoxicity on the target cells required for analyzing virion infectivity prevented this analysis. Importantly, this solvent is no longer present in our experiments with 3D matrices and the supernatant of these 3D cultures does not contain detectable antiviral activity (Fig. 1d (see “collagen supernatant + virus” condition)). We also digested already polymerized matrices using type I collagenase and incubated the digested matrices with virions. However, the remaining collagenase in these mixtures detached the reporter cells used for infectivity determination, again precluding the assessment of virion infectivity. This is now explained in the revised text (lines 141-143).

Additionally, we addressed all other original concerns regarding

- the identity of experimental controls (original comment 3) by adding the requested information to the text*
- differences in the magnitude of cytokine production from macrophages generated by different experimental protocols (original comment 4) by reference to specific literature describing this phenomenon*
- lack of characterization of early innate immune responses by adding new transcriptome analyses from primary macrophages at two time points post challenge with HIV*
- correctness of references to the HIV NL4.3 R5 by inserting the appropriate citation*

3 (new comment). The ex vivo models are available to validate this model.

Reply: A request for other ex vivo models is a new point not previously raised by this reviewer but the issue was mentioned by reviewer 2 in the first round of review.

Original comment Reviewer 2: It would appear to be outside of the scope of the study, however ex vivo validation (e.g., tissue explants, lymphoid organ models), and/or a model that includes target cells of interest in the collagen matrices would be of interest and enhance the translational significance of the findings.

Reply: We fully agree with the reviewer that it would be great to validate our findings in an organoid or organotypic model system. From our viewpoint, tonsil explants or cultures would be ideal for this and are established in the lab. However, both systems are subject to infusion of cell culture media, creating unphysiological liquid flows. The difficulty arising from this is that it is impossible to know for both, virus particles added to the culture or produced in the culture as result of prior infection, whether they had the chance to undergo physical contact with ECM. For this, we would need an encapsulated lymphoid organ that produces virus in the inside and then transports particles to the periphery. Establishing such a microfluidics-supported organ-on-chip setting in a BSL-3 environment is a long-term goal of the lab, but we are nowhere near.

This reply was satisfactory for reviewer 2: The authors now articulate the challenges of adapting organotypic systems for ECM studies under BSL-3 conditions, appropriately framing these as future directions in response to reviewer's inquisition on the relevance and translation of the study to in vivo events (see point vi of major revisions addressed in comments of reviewer 2 below).

Reviewer #2 (Remarks to the Author):

REVISED MANUSCRIPT EVALUATION REPORT: REVIEWER 2 (co-reviewed with Early Career Researchers)

Summary of Revisions:

The authors have submitted a scientifically sound rebuttal in response to prior reviewer feedback. All major and minor concerns have been comprehensively addressed resulting in a well-integrated, methodically rigorous and novel study. The additional experimental data, improved figure presentation, expanded methodological transparency and thoughtful discussion of scope and limitations have substantially elevated the manuscript. The newly included evidence of collagen-mediated enhancement of virion association with TLR2 in an Env-dependent manner (New Fig 7e-f), adds important mechanistic depth to the work.

Reply: We are glad to learn that the reviewer appreciates how we addressed his/her major and minor comments and is now fully satisfied with the manuscript.

Major Revisions Addressed:

i. Clarification of Infection Conditions. The authors confirmed that DEAE-Dextran was not used in the TZM-bl assay and clarified this in the revised Materials and Methods thereby strengthening the interpretation of strain-specific infectivity differences.

ii. Inclusion of Additional Experimental Data. New infection data in monocyte-derived macrophages (MDMs) using multiple primary Env strains now substantiate the earlier conclusions drawn from NL4-3.R5, with expanded donor validation.

iii. Mechanistic Expansion-TLR2 Association. Newly added analyses demonstrating that collagen experienced virions show enhanced physical association with TLR2 confirm a direct mechanistic link between ECM 'priming' and innate immune recognition.

iv. Clarification of Model Virus Rationale. The authors provided a clear rationale for using NL4-3.R5, addressing concerns regarding model relevance.

v. Collagen Source Discussion and Cross-Species Relevance. The revised text now discusses sequence homology across mammalian collagens and shows consistent restriction effects from human and murine collagens.

vi. Enhanced Discussion and Limitations. The authors now articulate the challenges of adapting organotypic systems for ECM studies under BSL-3 conditions, appropriately framing these as future directions in response to reviewer's inquisition on the relevance and translation of the study to in vivo events.

Evaluation of Added Data and Revisions:

i. Scientific Soundness
The new data sets are internally consistent, well-controlled, and directly address prior gaps.

ii. Data Presentation
Figure legends, axis labels, and nomenclature have been uniformly corrected.

iii. Textual Accuracy and Style
The manuscript reads clearly, with precise methodological detail and appropriate citations.

iv. Integration

v. Newly added data and clarifications blend seamlessly into the main narrative, strengthening logical flow.

Minor Editorial Points:

All previously noted typographical, labeling, and reference issues have been corrected. Figure legends now clearly describe reagents, strains, and analytical methods. The Materials and Methods section was updated to include all relevant antibody details, plasmid nomenclature and normalization procedures.

Scientific Significance and Impact.

This work provides a novel mechanistic framework linking extracellular matrix interactions with viral structural and immunological outcomes. The demonstration that transient engagement of HIV-1 virions with collagen fibrils simultaneously restricts infectivity and promotes innate immune activation is conceptually innovative and experimentally convincing. These findings expand understanding of how tissue microenvironments shape viral pathogenesis and immune sensing.

Reviewer #3 (Remarks to the Author):

This is a revised version of a manuscript by Fackler and colleagues characterizing the effects

of the extracellular matrix (ECM) on HIV infection. Incubation of virus in polymerized collagen was shown to negatively impact infectivity, and to influence how the virus causes cellular activation of innate immune signaling, detected by TLRs.

The authors were responsive to the prior review and have improved the manuscript and I only have a few remaining comments and questions, mostly regarding the effects of collagen on virion infectivity.

Reply:

Thank you for this positive assessment of the quality and quantity of our revisions. To clarify the few remaining comments, we added additional experimental data and made textual changes (see replies to the specific comments below).

1. Fusion mechanism- The data are convincing that there is a detrimental effect on virion infectivity but I still find the proposed mechanism puzzling and model underdeveloped. For example, Lines 132-135 discuss the “kinetics of infectivity impairment” and say the effects are “immediate” for LC (data in 1C). A rapid or “immediate” effect still doesn’t make sense to me for a saturable restriction.

Reply: We may not have made clear enough that the t=0 timepoint represents the time point at which both DC & LC are polymerized, which in fact represents 10 minutes of incubation time for DC and 45min for LC. The term “immediate” therefore is indeed misleading. We have modified the text accordingly and now depict more clearly the timing of the experiments in Fig. 1C. Our previous data demonstrated that inside of the collagen matrix, more than 7% of all virions physically interact with collagen fibers within 5 min (Imle et al., 2019). Together with the elevated probability for physical encounter of particles with fibers during the diffusion process at the surface of the 3D matrix, the time required for collagen polymerization is consistent with a scenario that these physical encounters trigger the observed drop in infectivity (also see response to point 3). This is now explained in more detail in the revised text (lines 133-135).

2. Also, more be said about the “saturable” results (1G, lines 150-157)- first, the effects are not fully saturated. Second, how can a physical restriction work in this fashion? E.g., if you were to pretreat the collagen with virus or soluble Envelope, wash, and then add virions- would the effect go away? Still need a more reasonable explanation here.

Reply: We agree that this would be a very informative experiment, however HIV-1 particles only very inefficiently enter into the 3D collagen matrix from the overlaying cell culture medium. Virions therefore have to be embedded into the 3D matrix during polymerization for studying the effect of 3D collagen, which precludes the suggested pretreatment-washout experiments. Alternatively, we made use of the of the 2D collagen setting, in which virions interact with a collagen-coated surface and their infectivity is reduced (Fig 1 d). In this experimental set-up, competition with non-infectious lentiviral particles that contain ERVI-sensitive Env reduces the ability of collagen to restrict the infectivity of HIV particles and this effect is dose-dependent for low levels of competing virus particles. This suggests the involvement of specific interaction sites in collagen that are rate limiting for the antiviral effect. However, the abrogation of antiviral effects by these low amounts of competing particles was only partial. Unfortunately, at higher

doses of competing virus, their effect is also detected at the level of the infection of reporter cells, presumably reflecting that they also compete with the infectious HIV particles for access to entry receptors at this step. While these results suggest an important role for specific physical or functional interaction sites in collagen that are involved in the antiviral activity of tissue-like environments, they do not allow us to conclude whether the presence of Env interaction sites in collagen is sufficient to explain the full range of antiviral activity. The data is now shown in Sup. Fig. 1b and discussed in the manuscript (lines 166-177).

3. I'm still puzzled why PNF was not included as a control for Figure 3's assays on Vpr-blam fusion considering that it fully rescued the defect.

Reply: Sorry, we did not understand correctly your previous comment to this point - this is of course a relevant and straightforward control. As expected, PNFs increase the fusogenicity of HIV-1 particles with prior collagen experience beyond that of particles kept in suspension. These new results are now included as Sup Fig 1f and are discussed in the manuscript (lines 302-305).

Very

minor:

1. Line 69- this may be a remaining instance of overstatement- "...can be controlled to complete the portfolio of experimental systems to study HIV spread.:

Reply: The statement has been modified accordingly.

2. Line 132- advert = adverse?

Reply: Corrected as suggested

3. Line 149- gp120 levels may have been similar but they do seem reduced for DC and LC- should be fair here in the text.

Reply: The statement was adjusted to explain that slight variations occurred but that gp120/p24 ratios were overall similar between all conditions.

4. Line 194- not sure I understand what is being said here- shouldn't RT be correlated with infectivity if that is what is being used to determine virion infectivity in earlier experiments?

Reply: Sorry if this was confusing. Identical amounts of infectious units were used for all virus. Due to intrinsic differences of the infectivity per RT unit ratio between different viruses and different virus preps, these virus stocks normalized for infectivity contain different amounts of RT, which is however not really relevant in this context. We simplified the statement.

5. Throughout- please specify time points in the results section for the reader- would help to interpret comparisons.

Reply: Time points have been added throughout the text.

6. Line 207- statement not necessarily correct unless dose titrations were done for all viruses?

Reply: The statement was adjusted.

7. Line 330: just a suggestion but this transition feels awkward.

Reply: Was corrected.

8. Line 408 may be a better sentence for the discussion.

Reply: The sentence was moved to the discussion as suggested.

Reviewer #4 (Remarks to the Author):

Reply: Thank you for your time and effort invested in reviewing our manuscript.

Reviewer #3 (Remarks to the Author):

In this second revision, the authors have addressed my remaining comments.

Reply: We thank the reviewer for helping to improve the manuscript and are glad to learn that our revisions were satisfactory.

Reviewer #1:

My primary concern is that the authors remain resistant to establishing the relevance and validity of their model. Their 3D collagen model itself has been published in Nature Communications previously, and in this manuscript the authors incorporate innate immune response components. However, I respectfully disagree with their position regarding the challenges of validating the model. A relatively simple air-liquid culture system was described in 2009 (Nature Protocols, PMID: 19197269) and has been widely adopted in the field, with more than 130 citations in PubMed. Our laboratory has also obtained highly reproducible data from multiple donors using this approach.

Reply: We fully agree with the reviewer that it would be great to validate our findings in an organoid or organotypic model system. As the reviewer suggests, tonsil explants (HLH) or reaggregating suspension cultures (HLAC) come to mind for this and are established in the lab. However, both systems are subject to infusion of cell culture media, creating unphysiological liquid flows. The difficulty arising from this is that it is impossible to know for both, virus particles added to the culture or produced in the culture as result of prior infection, whether they had the chance to undergo physical contact with ECM. Another major limitation comes from the fact that the amounts of virus particles produced from infected HLH cultures are too low to study innate immune signaling in MDM challenge experiments. Applying the tonsil system to our specific question is thus not feasible and potential results with this approach would be impossible to interpret. We are trying to solve this by two approaches in the lab: establishing an encapsulated organ-on-chip lymphoid organ that produces virus in the inside and then transports particles to the periphery and setting up protocols for the decellularization of tonsil explants and extraction of their ECM. Both these approaches constitute long-term goals for future studies of the lab, but we are nowhere near to having them running. These issues are now discussed in the revised manuscript (lines 554-557).